# CONVERGENCE ANALYSIS OF NESTEROV'S ACCELERATED GRADIENT DESCENT UNDER RELAXED ASSUMPTIONS

## ABSTRACT

We study convergence rates of Nesterov's Accelerated Gradient Descent (NAG) method for convex optimization in both deterministic and stochastic settings. We focus on a more general smoothness condition raised from several machine learning problems empirically and theoretically. We show the accelerated convergence rate of order $\mathcal{O}\left(1/T^2\right)$ in terms of the function value gap, given access to exact gradients of objective functions, matching the optimal rate for standard smooth convex optimization in (Nesterov, 1983). Under the relaxed affine-variance noise assumption for stochastic optimization, we establish the high-probability convergence rate of order $\tilde{\mathcal{O}}\left(\sqrt{\log\left(1/\delta\right)/T}\right)$ and this rate could improve to $\tilde{\mathcal{O}}\left(\log\left(1/\delta\right)/T^2\right)$ when the noise parameters are sufficiently small. Here, $T$ denotes the total number of iterations and $\delta$ is the probability margin. Up to logarithm factors, our probabilistic convergence rate reaches the same order of the expected rate obtained in (Ghadimi & Lan, 2016) where the assumptions of bounded variance noise and Lipschitz smoothness are required.

## 1 INTRODUCTION

In this paper, we consider the following classical unconstrained optimization problem,

$$\min_{\boldsymbol{x}\in\mathbb{R}^d} f(\boldsymbol{x}), \tag{1}$$

where the objective function $f(\boldsymbol{x})$ is convex and can be potentially stochastic, i.e.,

$$f(\boldsymbol{x}) = \mathbb{E}_{\mathbf{z}\sim\mathcal{D}}[f_{\mathbf{z}}(\boldsymbol{x};\mathbf{z})].$$

Here $\mathcal{D}$ is a probability distribution from which the random vector $\mathbf{z}$ is drawn.

Gradient-based algorithms (Robbins & Monro, 1951; Nesterov, 1983; 2013; Duchi et al., 2011) play an important role in solving (1). As usual, one typically focuses on the function value gap for convex objectives and the squared gradient norm for general non-convex ones.[1] In the deterministic setting with access to the exact gradient $\nabla f(\boldsymbol{x})$, Gradient Descent (GD) achieves a convergence rate of $\mathcal{O}(1/T)$ for smooth convex functions (Nesterov, 2013), whereas for smooth non-convex functions, the rate of the same order is obtained for the squared gradient norm. Here, $T$ is the total number of iterations. The convergence rate for smooth convex optimization can be improved to $\mathcal{O}(1/T^2)$ using Nesterov's Accelerated Gradient Descent (NAG), as established in the seminal work (Nesterov, 1983). Furthermore, this complexity bound is known to be optimal among gradient based algorithms (Nemirovskij & Yudin, 1983), without further assumptions.

For stochastic optimization where only the gradient estimator is accessible, Stochastic Gradient Descent (SGD) (Robbins & Monro, 1951) is commonly used. Lan (2012) provided an expected upper bound of order $\mathcal{O}\left(1/T + \sigma/\sqrt{T}\right)$ for convex objective functions and Ghadimi & Lan (2013)

---

[1] An extensive literature on minimizing structured non-convex functions focuses on the function value gap. Examples include work on Polyak-Łojasiewicz functions (Karimi et al., 2016), (strongly) quasar-convex functions (Hinder et al., 2020) and (strongly) quasiconvex functions (Grad et al., 2025). This is beyond the discussion of this paper.

obtained the bound of the same order for the non-convex case, both of them assuming bounded variance noise with noise parameter $\sigma$ and smooth objective functions. This bound is optimal in the non-convex setting since it matches the lower bound in (Arjevani et al., 2023). To study the acceleration behavior in the stochastic convex optimization, Lan (2012); Ghadimi & Lan (2016) explored (and generalized) stochastic NAG (SNAG) and obtained the expected convergence rate of order $\mathcal{O}\left(1/T^2 + \sigma/\sqrt{T}\right)$ for smooth objective functions, which in general cannot be improved in the same setting (Nemirovskij & Yudin, 1983; Lan, 2012).

Although much theoretical progress has been made on gradient-based algorithms, most of these analysis required Lipschitz smoothness condition (Ghadimi & Lan, 2013; 2016; Levy et al., 2018; Ward et al., 2020; Attia & Koren, 2023), i.e., $\exists L > 0$, such that

$$\|\nabla f\left(\boldsymbol{x}\right) - \nabla f\left(\boldsymbol{y}\right)\| \leq L \|\boldsymbol{x} - \boldsymbol{y}\|, \forall \boldsymbol{x}, \boldsymbol{y} \in \mathbb{R}^d,$$

or equivalently $\left\|\nabla^2 f\left(\boldsymbol{x}\right)\right\| \leq L, \forall \boldsymbol{x} \in \mathbb{R}^d$ for twice-differentiable functions. Recently, several researchers have found evidence that this condition is not satisfied by many important machine learning models (Chen et al., 2023), such as neural network models (Zhang et al., 2020b) and distributionally robust optimization (Jin et al., 2021). Based on empirical observations, Zhang et al. (2020b) proposed $(L_0, L_1)$-smoothness condition, allowing $\left\|\nabla^2 f\left(\boldsymbol{x}\right)\right\|$ to grow linearly with respect to $\|\nabla f\left(\boldsymbol{x}\right)\|$, and later Zhang et al. (2020a) further relaxed this condition, not requiring the second differentiability of the objective function, i.e., there exist $L_0, L_1 \geq 0$, for any $\boldsymbol{x}, \boldsymbol{y} \in \mathbb{R}^d$, such that $\|\boldsymbol{x} - \boldsymbol{y}\| \leq 1/L_1$,

$$\|\nabla f\left(\boldsymbol{x}\right) - \nabla f\left(\boldsymbol{y}\right)\| \leq \left(L_0 + L_1 \|\nabla f\left(\boldsymbol{x}\right)\|\right) \|\boldsymbol{x} - \boldsymbol{y}\|. \tag{2}$$

Based on this generalized smoothness condition, Yu et al. (2025) studied Randomized Stochastic Accelerated Gradient Descent (RSAG) proposed in (Ghadimi & Lan, 2016) and provided high-probability convergence rate of order $\tilde{\mathcal{O}}\left(1/T + \sigma/\sqrt{T}\right)$ for both convex and non-convex optimization (under sub-Gaussian relaxed affine-variance noise), which implies a gap between optimal rate obtained in the smooth convex optimization. Under a similar generalized smoothness condition, Li et al. (2024) showed accelerated convergence rate of order $\mathcal{O}\left(1/T^2\right)$ for deterministic NAG in convex optimization, and they also provided expected convergence rate of order $\mathcal{O}\left(1/T + \sigma/\sqrt{T}\right)$ for SGD in the non-convex stochastic optimization. To the best of our knowledge, it remains an open question whether SNAG can achieve an accelerated convergence rate of order $\tilde{\mathcal{O}}\left(1/T^2 + \sigma/\sqrt{T}\right)$ under the generalized smoothness condition for convex optimization. We believe that a proof for the stochastic setting presents certain challenges; in particular, the analysis for deterministic NAG by (Li et al., 2024) does not appear to be trivially extendable.

In this paper, we aim to close this gap, developing the accelerated convergence rate for SNAG under more generalized smoothness and relaxed affine-variance noises for stochastic convex optimization. Specifically, inspired by the theoretical examples in (Taheri & Thrampoulidis, 2023) and (Chen et al., 2023), we focus on the following more general and practical smoothness condition.

**Definition 1** $((L_0, L_1, L_2)$-smoothness). *Let* $L_i \geq 0, \forall 1 \leq i \leq 3$. $f(\cdot)$ *is* $(L_0, L_1, L_2)$-*smooth if and only if for any* $\boldsymbol{x}, \boldsymbol{y} \in \mathbb{R}^d$ *such that* $\|\boldsymbol{x} - \boldsymbol{y}\| \leq \min\left\{1/L_1, 1/L_2\right\}^2$,

$$\|\nabla f(\boldsymbol{x}) - \nabla f(\boldsymbol{y})\| \leq \left(L_0 + L_1 \|\nabla f(\boldsymbol{x})\|^p + L_2 \left(f(\boldsymbol{x}) - f^*\right)^q\right) \|\boldsymbol{x} - \boldsymbol{y}\|, \tag{3}$$

*where* $p \in [0, 2)$ *and* $q \geq 0$.

Obviously, Definition 1 covers a broader range of relaxed smoothness. Particularly, it is situated between two related notions: $(L_0, L_1, 0)$-smoothness, which is empirically verified (Zhang et al., 2020b) for neural networks training and is theoretically proved for phase retrieval from (Chen et al., 2023) and the appendix, and $(L_0, 0, L_2)$-smoothness, which is theoretically proven for specific shallow neural networks from (Taheri & Thrampoulidis, 2023) and the appendix.

Our analysis relies on a relaxed affine-variance noise condition, which will be formally defined in (5) (Hong & Lin, 2024; Yu et al., 2025). This condition was initially proposed by (Khaled & Richtárik,

---

[2] For the sake of rigor, we define $1/0 = +\infty$ throughout the paper.

2023) in the expected form given in (6), and many practical stochastic gradient settings, such as sub-sampling and compression schemes satisfy this noise model, but not bounded variance or the strong growth condition that the stochastic gradient $g(\boldsymbol{x})$ of $f$ at $\boldsymbol{x}$ satisfies, for some non-negative constants $B$,

$$\mathbb{E}\left[\|g(\boldsymbol{x}) - \nabla f(\boldsymbol{x})\|^2\right] \le B \|\nabla f(\boldsymbol{x})\|^2, \forall \boldsymbol{x} \in \mathbb{R}^d. \tag{4}$$

Closely related to our works are (Vaswani et al., 2019; Gupta et al., 2024). Under the strong growth condition, Vaswani et al. (2019) analyzed the Accelerated Coordinate Descent method (ACDM) (Nesterov, 2012), while Gupta et al. (2024) studied SNAG when $B \le 1$. Both works achieved the expected accelerated convergence rates in the (strongly) convex setting, but only under the standard smoothness condition.

We summarize our main contributions as follows.

(a) Motivated by several machine learning problems, we propose a more general smoothness condition defined in Definition 1.

(b) Under this new smoothness condition, we analyze NAG in the deterministic and convex setting, and we show the accelerated convergence rate of order $\mathcal{O}\left(1/T^2\right)$, matching the optimal rate in (Nesterov, 1983).

(c) For stochastic optimizations under this general smoothness, we focus on the sub-Gaussian version of relaxed affine-variance noise (Assumption 3), and we prove that SNAG converges at the rate of $\tilde{\mathcal{O}}\left(1/T^2 + \sqrt{(A + B + C)/T}\right)$ in high probability. This rate matches the optimal convergence rate for smooth convex optimization under bounded variance noise (Lan, 2012; Ghadimi & Lan, 2016). It could improve to $\tilde{\mathcal{O}}\left(1/T^2\right)$ if the noise parameters $A, B$ and $C$ are small enough.

(d) As a byproduct, we apply our analysis to standard smooth optimization under the expected relaxed affine-variance noises (Assumption 4), and we demonstrate that SNAG reaches the convergence rate of order $\mathcal{O}\left((1 + B)/T^2 + \sqrt{(A + C)/T}\right)$ in expectation.

The rest of this paper are organized as follows. We first briefly discuss some extra works related to NAG, generalized smoothness condition and the relaxed noise assumption. We then introduce some necessary assumptions and notations in Section 3. In Section 4, we provide the convergence results under $(L_0, L_1, L_2)$-smoothness, either in the deterministic setting or in the stochastic setting. In Section 5, we present the expected convergence rate of SNAG under the classic smoothness. In Section 6, we conduct numerical experiments and show the better performance of SNAG compared to SGD for the two-layer neural network and the phase retrieval model. In Section ??, we provide a proof sketch for high-probability convergence under the generalized smoothness. We also provide the convergence result for non-convex stochastic optimization under the generalized smoothness and relaxed noise assumptions in Section G. All the omitted proofs and lemmas are in the appendix.

## 2 RELATED WORK

We only briefly mention the most related works due to space and knowledge constraints.

**Accelerated Gradient Descent** NAG (Nesterov, 1983) was originally designed for smooth and convex optimizations in the deterministic setting, and it achieved the accelerated convergence rate of order $\mathcal{O}\left(1/T^2\right)$, compared to $\mathcal{O}\left(1/T\right)$ of GD. Numerous literature focused on the theoretical and practical convergence behavior of NAG and its variants (Nesterov, 2005; Beck & Teboulle, 2009). For example, Su et al. (2016) introduced a second-order ODE and accompanying tools for characterizing NAG. Lan (2012) generalized NAG for non-smooth and stochastic convex problems under certain conditions and provided optimal convergence rates under proper step sizes. Ghadimi & Lan (2016) proposed RSAG, and showed expected convergence rate of $\mathcal{O}\left(1/T + C/\sqrt{T}\right)$ in the non-convex case while $\mathcal{O}\left(1/T^2 + C/\sqrt{T}\right)$ in the convex case, both under bounded variance noises and smoothness. Li et al. (2024) obtained convergence rate of order $\mathcal{O}\left(1/T^2\right)$ for NAG under generalized smoothness and convexity, matching those for standard smooth convex optimizations. Their

analysis is limited to the non-stochastic case. Under mild noises in (4) and standard smoothness, Vaswani et al. (2019) proved that ACDM (Nesterov, 2012), which is a variant of SNAG, could reach expected accelerated convergence rates in both convex and strongly convex cases. Under the same setting, Gupta et al. (2024) proposed a new accelerated gradient method named AGNES and they proved that the algorithm could achieve acceleration, requiring fewer hyperparameters than ACDM. ~~They also demonstrated that SNAG could achieve acceleration rate when $B < 1$.~~ Furthermore, Hermant et al. (2025) showed the expected convergence rate of $\mathcal{O}\left((B+1)/T^2\right)$ and almost-sure rate of $o\left((B+1)/T^2\right)$ for ACDM in general convex optimization problems, and they derived fast convergence rates for ACDM in strongly convex optimization problems.

**Relaxed affine-variance noise and its variants**  Affine-variance noise (i.e., $A = 0$ in (6)) has attracted increasing attention as it can characterize gradient noises in many practical problems, such as machine learning with feature noise (Fuller, 2009; Khani & Liang, 2020), robust linear regression (Xu et al., 2008) and multilayer networks (Faw et al., 2022). Bottou et al. (2018) analyzed vanilla SGD and pointed out that there is no essential difference in the analysis between the bounded variance noise and the affine-variance noise under standard smoothness. For Adagrad-Norm, Faw et al. (2022) provided expected convergence rates of order $\tilde{\mathcal{O}}\left(1/\sqrt{T}\right)$ in the non-convex setting and this rate could reach $\tilde{\mathcal{O}}\left(1/T\right)$ when $B, C$ are of order $\mathcal{O}\left(1/\sqrt{T}\right)$. Under the same setting, Wang et al. (2023) further proposed a novel auxiliary function for analysis and obtained a tighter bound especially when $C = 0$. Attia & Koren (2023) derived high probability convergence for Adagrad-Norm in both convex and non-convex cases, under almost-sure version of affine-variance noises. Khaled & Richtárik (2023) proposed the relaxed affine-variance noise (see (6)), and they derived an expected convergence rate of order $\mathcal{O}\left(1/\sqrt{T}\right)$ for SGD in the non-convex and smooth setting. Hong & Lin (2024) considered sub-Gaussian version of the relaxed affine-variance noise, and they derived probabilistic convergence rates under $(L_0, L_1)$-smoothness. Yu et al. (2025) analyzed RSAG (covering SGD as a special case) in both convex and non-convex settings under $(L_0, L_1)$-smoothness.

**Generalized smoothness**  Motivated by practical observations, Zhang et al. (2020b) proposed $(L_0, L_1)$-smoothness for twice differentiable functions. They showed $\mathcal{O}\left(1/T\right)$ convergence rate for GD and $\mathcal{O}\left(1/\sqrt{T}\right)$ convergence rate for SGD in the non-convex setting, involving extra clipping mechanisms. Zhang et al. (2020a) improved the convergence analysis on problem-dependent parameters for clipped SGD under essentially the same smoothness. In the analysis of Adagrad-Norm under affine-variance noises, Faw et al. (2023) derived convergence bounds of order $\tilde{\mathcal{O}}\left(1/\sqrt{T}\right)$ in the non-convex case when $B < 1$. Wang et al. (2023) gave a counter-example showing the necessity of prior knowledge on problem parameters for learning rates in AdaGrad under $(L_0, L_1)$-smoothness. Via a notion of continuity, Guille-Escuret et al. (2021) demonstrated that the strong convexity and smoothness have a weakness resulting in a lack of robustness for tuning first-order algorithms, and they presented promising alternatives.

Refer to Table 1 and Table 2 for comparisons of the most relevant works.

## 3 PRELIMINARIES

We consider Problem (1) over the Euclidean space $\mathbb{R}^d$ with the $l_2$ norm, denoted as $\|\cdot\|$. We first introduce the following assumption.

**Assumption 1** (Below bounded). *There exists a minimizer $\boldsymbol{x}^* \in \mathbb{R}^d$ and the objective function is bounded from below, i.e.,*

$$f\left(\boldsymbol{x}^*\right) = f^* := \inf_{\boldsymbol{x} \in \mathbb{R}^d} f(\boldsymbol{x}) > -\infty.$$

In the stochastic setting, we make the following assumptions.

**Assumption 2** (Unbiased estimator). *The gradient oracle returns an unbiased estimator of $\nabla f(\boldsymbol{x})$, i.e., for all $\boldsymbol{x} \in \mathbb{R}^d$,*

$$\mathbb{E}_{\boldsymbol{z}}\left[\nabla f_{\boldsymbol{z}}(\boldsymbol{x}; \boldsymbol{z})\right] = \nabla f(\boldsymbol{x}).$$

**Assumption 3** (Relaxed affine-variance (sub-Gaussian form)). *The gradient oracle satisfies that for some constants $A, B, C \geq 0$,*

$$\mathbb{E}_{\mathbf{z}} \left[ \exp \left( \frac{\|\nabla f_{\mathbf{z}}(\boldsymbol{x}; \mathbf{z}) - \nabla f(\boldsymbol{x})\|^2}{A \left( f(\boldsymbol{x}) - f^* \right) + B \|\nabla f(\boldsymbol{x})\|^2 + C} \right) \right] \leq \exp(1), \forall \boldsymbol{x} \in \mathbb{R}^d. \tag{5}$$

**Assumption 4** (Relaxed affine-variance (expected form)). *The gradient oracle satisfies that for some constants $A, B, C \geq 0$,*

$$\mathbb{E}_{\mathbf{z}} \left[ \|\nabla f_{\mathbf{z}}(\boldsymbol{x}; \mathbf{z}) - \nabla f(\boldsymbol{x})\|^2 \right] \leq A \left( f(\boldsymbol{x}) - f^* \right) + B \|\nabla f(\boldsymbol{x})\|^2 + C, \forall \boldsymbol{x} \in \mathbb{R}^d. \tag{6}$$

Assumption 2 is a relevant assumption for studying many practical settings and is also commonly used in the analysis of stochastic optimization. Assumption 3 is weaker than the bounded noise in (Zhang et al., 2020b;a) and the almost-sure version of (relaxed) affine-variance noise in (Attia & Koren, 2023; Hong & Lin, 2024; Yu et al., 2025). Although~~While~~ Assumption 3 is stronger than its expected version in Assumption 4 as it controls all moments of the noise distribution, while Assumption 4 only controls its second moment (the variance), the former one could lead to high-probability convergence, which could ensure corresponding expected convergences. Assumption 4 was initially proposed by Khaled & Richtárik (2023) under the name expected smoothness. Its original, equivalent form is: $\mathbb{E}_{\mathbf{z}} \left[ \|\nabla f_{\mathbf{z}}(\boldsymbol{x}; \mathbf{z})\|^2 \right] \leq A \left( f(\boldsymbol{x}) - f^* \right) + \tilde{B} \|\nabla f(\boldsymbol{x})\|^2 + C, \forall \boldsymbol{x} \in \mathbb{R}^d$.

**Notations** We denote the set $\{1, \cdots, T\}$ as $[T]$. We use $\mathbb{E}_t[\cdot] \triangleq \mathbb{E}[\cdot | \mathbf{z}_1, \cdots, \mathbf{z}_{t-1}]$ to represent the conditional expectation, where $\mathbf{z}_i$ is the random sample in the $i$-th gradient oracle. The notation $a \sim \mathcal{O}(b)$ and $a \leq \mathcal{O}(b)$ refer to $c_1 b \leq a \leq c_2 b$ and $a \leq c_3 b$ with $c_1, c_2, c_3$ being positive constants, respectively. Also, we write $\tilde{\mathcal{O}}(b)$ for $\mathcal{O}(b \cdot \text{poly}(\log b))$. Throughout the paper, we define $0^0 = 1$.

## 4 CONVERGENCE OF NAG UNDER $(L_0, L_1, L_2)$-SMOOTHNESS

In this section, we assume that the objective function satisfies Definition 1. We present convergence results for the deterministic case in Section 4.1 and for the stochastic case in Section 4.2. The detail proofs for this section will be given in Section D and Section E of the appendix.

### 4.1 CONVERGENCE RESULTS FOR DETERMINISTIC OPTIMIZATION

We first present convergence rates of NAG in the deterministic case with a slight modification (see Algorithm 1). This modified NAG is proposed by (Li et al., 2024) where they obtained the optimal convergence rate under a general smoothness for convex non-stochastic optimizations. The only difference between Algorithm 1 and original NAG (Nesterov, 1983) is that the latter directly sets $A_t = B_t$. Such a modification could be used to control the gradient norms (or function value gaps) in the analysis.

---

**Algorithm 1** Nesterov's Accelerated Gradient Descent (NAG)

---

**Require:** Horizon $T$, $\boldsymbol{x}_0^{ag} = \boldsymbol{x}_0 \in \mathbb{R}^d$, step sizes $\beta, \{\lambda_t\}_{t \in [T]}$ and $A_0 = 1/\beta, B_0 = 0$.
1: **for** $t = 1, \cdots, T$ **do**
2: $\quad$ $B_t = B_{t-1} + \frac{1}{2} \left( 1 + \sqrt{4B_{t-1} + 1} \right)$;
3: $\quad$ $A_t = B_t + \frac{1}{\beta}$;
4: $\quad$ $\boldsymbol{x}_t^{md} = \frac{A_{t-1}}{A_t} \boldsymbol{x}_{t-1}^{ag} + \left( 1 - \frac{A_{t-1}}{A_t} \right) \boldsymbol{x}_{t-1}$;
5: $\quad$ $\boldsymbol{x}_t = \boldsymbol{x}_{t-1} - \lambda_t \nabla f(\boldsymbol{x}_t^{md})$;
6: $\quad$ $\boldsymbol{x}_t^{ag} = \boldsymbol{x}_t^{md} - \beta \nabla f(\boldsymbol{x}_t^{md})$.

---

To better understand the NAG method, we provide the following lemma summarized from (d'Aspremont et al., 2021; Li et al., 2024).

**Lemma 4.1.** *For all $0 \leq t \leq T$, we have*

1. $\frac{1}{4}t^2 \le B_t \le t^2$;

2. $(A_t - A_{t-1})^2 = (B_t - B_{t-1})^2 = B_t < A_t; (A_t - A_{t-1})^2 = (B_t - B_{t-1})^2 = B_t \le A_t$

3. $A_t - A_{t-1} = B_t - B_{t-1} \ge 1$. Thus, $\{A_t\}_{t\in[T]}$ and $\{B_t\}_{t\in[T]}$ are both monotonically increasing sequences.

The above lemma plays vital roles both in the induction argument for bounding the function value gap and in the final convergence analysis. Refer to Section H for the complete proof.

**Theorem 1.** *Let $T > 0$ and $f$ be an $(L_0, L_1, L_2)$-smooth convex function. Suppose that$\{x_t^{ag}\}_{t\in[T]}$ is a sequence generated by Algorithm 1 with step sizes $\beta, \lambda_t$ satisfying*

$$\beta = \frac{1}{\mathcal{L}_1}, \qquad \lambda_t = (A_t - A_{t-1})\beta, \tag{7}$$

*where $\mathcal{L}_1$ is a constant, depending on the smoothness parameters $\{L_i\}_{i\in[3]}, p, q$, with its explicit expression in (27). Then, under Assumption 1, we have[3]*

$$f(x_T^{ag}) - f^* \le \mathcal{O}\left(\frac{1}{T^2}\right). \tag{8}$$

Considering the definition of $\mathcal{L}_1$ in (27), $\beta$ could reduce to $1/2L$ when the objective function is $L$-smooth, which aligns with $\beta = 1/L$ in (Nesterov, 1983) up to a constant. Furthermore, Theorem 1 recovers the convergence rate of order $\mathcal{O}\left(1/T^2\right)$ obtained in (Nesterov, 1983) where the smoothness is required. This bound is optimal (Nemirovskij & Yudin, 1983) for smooth convex optimization when $d$ is large enough.

### 4.2 Convergence results for stochastic optimization

In this section, we provide a probabilistic convergence result for SNAG (see Algorithm 2) under the relaxed affine-variance noise assumption of its sub-Gaussian form. Compared to Algorithm 1, the only difference is that stochastic gradients, instead of accurate gradients, are accessible. Obviously, Lemma 4.1 still holds for the stochastic case.

---

**Algorithm 2** Stochastic Nesterov's Accelerated Gradient Descent (SNAG)

---

**Require:** Horizon $T$, $x_0^{ag} = x_0 \in \mathbb{R}^d$, step sizes $\beta, \{\lambda_t\}_{t\in[T]}$ and $A_0 = 1/\beta, B_0 = 0$.
1: **for** $t = 1, \cdots, T$ **do**
2: $\quad$ $B_t = B_{t-1} + \frac{1}{2}\left(1 + \sqrt{4B_{t-1} + 1}\right)$;
3: $\quad$ $A_t = B_t + \frac{1}{\beta}$;
4: $\quad$ $x_t^{md} = \frac{A_{t-1}}{A_t}x_{t-1}^{ag} + \left(1 - \frac{A_{t-1}}{A_t}\right)x_{t-1}$;
5: $\quad$ **Set** $g_t = \nabla f_{\mathbf{z}}\left(x_t^{md}; \mathbf{z}_t\right)$;
6: $\quad$ $x_t = x_{t-1} - \lambda_t g_t$;
7: $\quad$ $x_t^{ag} = x_t^{md} - \beta g_t$.

---

**Theorem 2.** *Let $T > 0$ and $\delta \in (0, \frac{1}{3})$. Suppose that $\{x_t^{ag}\}_{t\in[T]}$ is a sequence generated by Algorithm 2, $f$ is $(L_0, L_1, L_2)$-smooth and convex, and the step sizes $\beta, \lambda_t$ satisfy that*

$$\beta = \min\left\{\frac{1}{\mathcal{G}_1}, \frac{1}{\mathcal{G}_2 T^{\frac{6}{5}}}, \frac{1}{\mathcal{G}_3 T^{\frac{2}{3}}}, \frac{1}{\mathcal{M}T^{\frac{3}{2}}}, \frac{1}{\mathcal{M}^2 T}\right\}, \quad \lambda_t = \frac{1}{4}\beta\left(A_t - A_{t-1}\right), \tag{9}$$

*where $\mathcal{G}_1, \mathcal{G}_2, \mathcal{G}_3$ and $\mathcal{M}$ are polynomials of $\log\frac{T}{\delta}$, depending on the noise and smoothness parameters[4]. Under Assumptions 1, 2 and 3, with probability at least $1 - 3\delta$, we have[5]*

$$f(x_T^{ag}) - f^* \le \tilde{\mathcal{O}}\left(\frac{1}{T^2} + \sqrt{\frac{A + B + C}{T}}\right).$$

---

[3]We state the explicit convergence result in (42).

[4]The explicit expressions of these notations are presented in (43), (44), (45) and (46).

[5]Refer (73) for the explicit convergence result.

Theorem 2 provides accelerated convergence rates in high probability. Up to logarithm factors, this convergence rate matches the expected convergence rate in (Ghadimi & Lan, 2016), where they assumed bounded variance noise and standard smoothness, and it is unimprovable for smooth convex stochastic optimization (Lan, 2012).

Furthermore, the convergence rate in Theorem 2 could accelerate to $\tilde{\mathcal{O}}\left(1/T^2\right)$ if the noise parameters are sufficiently small, which matches the rate for the deterministic NAG in (Li et al., 2024) under a generalized $(L_0, L_1, 0)$-smoothness: $\left\|\nabla^2 f\left(\boldsymbol{x}\right)\right\| \leq l\left(\left\|\nabla f\left(\boldsymbol{x}\right)\right\|\right)$ with a sub-quadratic non-decreasing positive function $l$ up to logarithm factors. Note that Li et al. (2024) did not provide the analysis for NAG and we consider the $(L_0, L_1, L_2)$-smoothness. To extend to stochastic setting, we modify the step size slightly by a constant factor and $\beta \sum_{i=1}^{t} A_i \left\|\nabla f(\boldsymbol{x}_i^{md})\right\|^2$ appears in (61), which makes it feasible to bound $\left\|\boldsymbol{x}_{t+1}^{md} - \boldsymbol{x}_t^{ag}\right\|^2$ in stochastic optimization. Combining with several probabilistic lemmas, we could finish the proof. We refer to Section E for the complete proof. Our analysis for the above theorem, which relies on Assumption 3, does not apply under the weaker noise condition of Assumption 4 in the generalized smoothness.

## 5 CONVERGENCE OF NAG UNDER LIPSCHITZ SMOOTHNESS

We apply our analysis to smooth stochastic optimization and demonstrate that SNAG could reach the accelerated convergence rate in expectation under the relaxed affine-variance noises and standard smoothness.

**Theorem 3.** *Let $T > 0$, $f$ be $L$-smooth and convex. Suppose that $\{\boldsymbol{x}_t^{ag}\}_{t \in [T]}$ is a sequence generated by Algorithm 2 with step sizes*

$$\beta = \min\left\{\frac{1}{2L\left(1+B\right)}, \frac{1}{\mathcal{Q}^{\frac{1}{2}}T^{\frac{3}{2}}}, \frac{1}{\mathcal{Q}T}\right\}, \quad \lambda_t = \frac{\beta}{2\left(1+B\right)}\left(A_t - A_{t-1}\right), \tag{10}$$

*where $\mathcal{Q} = A\mathcal{F}_3 + C$ is a constant depending on the parameters of smoothness and noise with $\mathcal{F}_3$ defined in (78). Under Assumptions 1, 2 and 4, we have[6]*

$$\mathbb{E}\left[f\left(\boldsymbol{x}_T^{ag}\right) - f^*\right] \leq \mathcal{O}\left(\frac{1+B}{T^2} + \sqrt{\frac{A+C}{T}}\right). \tag{11}$$

The above theorem relaxes the bounded variance noise assumption in (Ghadimi & Lan, 2016) while providing the optimal expected convergence rate. Furthermore, Theorem 3 improves the convergence rate of order $\mathcal{O}\left(1/T + C/\sqrt{T}\right)$ for SGD and RSAG in (Yu et al., 2025) under the same assumption. Compared to Theorem 2, the suboptimal term $\mathcal{O}\left(\sqrt{B/T}\right)$ with respect to $B$ disappears in (11), which aligns with the expected result of $\mathcal{O}\left((B+1)/T^2\right)$ and almost-sure result of $o\left((B+1)/T^2\right)$ in (Hermant et al., 2025) where they focused on smooth stochastic optimization with noise satisfying (4).

## 6 NUMERICAL EXPERIMENT

In this section, we show the practical convergence behavior of SNAG (Algorithm 2) compared to stochastic AGD (Algorithm 3 discussed in the appendix) and SGD, i.e.,

$$\boldsymbol{x}_{t+1} = \boldsymbol{x}_t - \eta \nabla_{\boldsymbol{z}} f(\boldsymbol{x}_t; \boldsymbol{z}_t), \tag{12}$$

on the two-layer neural network (13) and phase retrieval model (14). We prove that both the two models satisfy the $(L_0, L_1, L_2)$-smoothness condition in Section B.

---

[6]The detail convergence result is presented in (80).

**Two-layer neural network**    Considering the following problem,

$$\min_{\boldsymbol{x} \in \mathbb{R}^{\tilde{d}}} F(\boldsymbol{x}) = \frac{1}{n} \sum_{i=1}^{n} f\left(y_i \Phi\left(\boldsymbol{x}, \boldsymbol{w}_i\right)\right), \tag{13}$$

where $\boldsymbol{w}_i \in \mathbb{R}^d$ is the data point and its associated label $y_i \in \{\pm 1\}$. The function $f(\cdot)$ is the exponential loss i.e., $f(t) = \exp(-t)$ and $\Phi(\cdot)$ is a two-layer neural network with $m$ neurons defined as

$$\Phi(\boldsymbol{x}, \boldsymbol{w}) = \sum_{j=1}^{m} a_j \sigma(\langle \boldsymbol{x}_j, \boldsymbol{w} \rangle),$$

Here $\sigma : \mathbb{R} \to \mathbb{R}$ is the activation function and $\boldsymbol{x}_j \in \mathbb{R}^d$ denotes the input weight vector of the $j$th hidden neuron. $\boldsymbol{x} \in \mathbb{R}^{\tilde{d}}$ represents the concatenation of these weights, i.e., $\boldsymbol{x} = [\boldsymbol{x}_1; \boldsymbol{x}_2; \cdots ; \boldsymbol{x}_m]$ where $\tilde{d} = md$. We assume that only $\boldsymbol{x}_j$ can be updated during training, while $\boldsymbol{a}_j \in \mathbb{R}$ are initialized randomly and kept fixed.

We conduct experiment on the specific shallow neural network with $m = 30$ hidden neurons, exponential loss $f(t) = \exp(-t)$ and smoothed-leaky-ReLU activation function, i.e.,

$$\sigma(t) = t\mathbb{I}(t \geq 0) + 0.2t\mathbb{I}(t < 0),$$

where $\mathbb{I}(\cdot)$ is the $0 - 1$ indicator function. We generate the data point $\boldsymbol{w}_i \in \mathbb{R}^d$, where dimension $d = 10$, coordinate-wise from Gaussian distribution $\mathcal{N}(0, 25)$ with its binary label $y_i \in \{\pm 1\}$ chosen randomly. The second layer weights are generated randomly from $a_j \in \left\{\pm \frac{1}{m}\right\}$ and kept fixed during training.

**Phase retrieval**    Phase retrieval is a classic model in the field of machine learning and signal processing (Drenth, 1994; Miao et al., 1999; Chen et al., 2023). In this setting, we are aimed to solve the following problem, i.e.,

$$\min_{\boldsymbol{x} \in \mathbb{R}^d} f(\boldsymbol{x}) := \frac{1}{2m} \sum_{i=1}^{m} \left(y_i - \left|\boldsymbol{a}_r^\top \boldsymbol{x}\right|^2\right)^2. \tag{14}$$

Here, $y_i$ represents the intensity measurements, i.e., $y_i = \left|\boldsymbol{a}_i^\top \boldsymbol{z}\right|, \forall i \in [m]$ with $\boldsymbol{a}_i \in \mathbb{R}^d$ being the fixed parameters and $\boldsymbol{z} \in \mathbb{R}^d$ being the true objects.

The data in our experiment are generated by $y_i = \left|\boldsymbol{a}_i^\top \boldsymbol{z}\right|^2 + \boldsymbol{\epsilon}_i, i \in [m]$, where each coordinate of both the measurement vector $\boldsymbol{a}_i \in \mathbb{R}^d$ and the true parameter $\boldsymbol{z}$ satisfy Gaussian distribution $\mathcal{N}(0, 0.5)$, and $\boldsymbol{\epsilon}_i \sim \mathcal{N}(0, 25)$ is the noise. Here, we set the number of samples $m = 1000$ and the dimension $d = 10$.

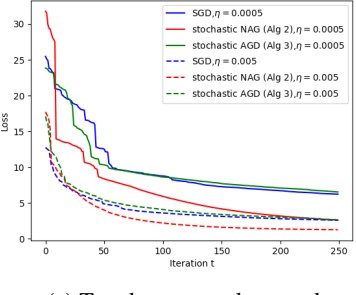
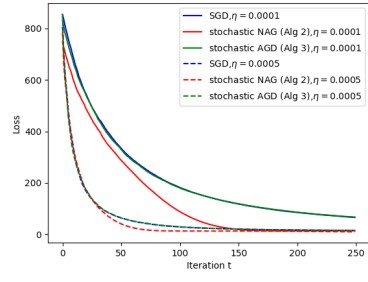

(a) Two-layer neural network          (b) Phase retrieval model

Figure 1: Experiment results. We run each algorithm 100 times and plot the average loss at each iteration.

**Experiment Setup**    We set $\beta = \eta$ in Algorithm 2 and $\lambda_t = \eta$ in Algorithm 3 where $\eta$ is also the step size of SGD. The stochastic gradient in each step is computed by samples randomly chosen with batch size 10. We start the training process with the initial vector satisfying $\mathcal{N}(1, 25)$.

**Results**   As Figure 1 shows, SGD and stochastic AGD (Algorithm 2) exhibit comparable performance under these two possibly non-convex setting, complementing their theoretical analysis. Stochastic NAG performs best among the three especially with small step size though we only prove its acceleration theoretically in the convex case.

# 7    CONCLUSION

In this paper, motivated by several machine learning problems, we propose a new general smoothness, which generalizes the global smoothness and $(L_0, L_1)$-smoothness. Under this condition, we analyze NAG method and obtain the accelerated convergence rate of order $\mathcal{O}\left(1/T^2\right)$ for convex optimizations with access to accurate gradients. For stochastic optimization, we obtain accelerated probabilistic convergence rates of order $\tilde{\mathcal{O}}\left(1/T^2 + \sqrt{(A + B + C)/T}\right)$ under sub-Gaussian relaxed affine-variance noises. Furthermore, we apply our analysis to smooth optimizations and obtain the result of order $\mathcal{O}\left((B + 1)/T^2 + \sqrt{(A + C)/T}\right)$ ~~the same convergence rates~~ in expectation under expected relaxed affine-variance noises. All the above derived convergence rates are optimal without further assumptions.

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

## DECLARATION OF LLM USAGE

We used a large language model (LLM) only for language polishing (grammar, clarity, and style). The model did not generate ideas, analyses, results, or citations. The authors are fully responsible for all content.

## A COMPARISONS OF PREVIOUS WORK WITH OURS

## B EXAMPLES SATISFYING THE $(L_0, L_1, L_2)$-SMOOTHNESS CONDITION

### B.1 TWO-LAYER NEURAL NETWORKS

Recall the two-layer neural network model in (13) and we have the following lemma from (Taheri & Thrampoulidis, 2023). We refer interested readers to see the proof in their paper.

Table 1: Related works under the generalized smoothness condition.

| | Alg. | Convexity | Noise | Smoothness | Conv. type | Conv. rate | Extra cond. for gradient |
|---|---|---|---|---|---|---|---|
| Zhang et al. (2020b) | SGD | non-convex | bounded (a.s.) | $(L_0, L_1)$ | $\mathbb{E}$ | $\frac{1+C}{\sqrt{T}}$ | ✓ |
| Li et al. (2024) | SGD | non-convex | bounded variance | generalized $(L_0, L_1)$ | $\mathbb{E}$ | $\frac{1+\sqrt{C}}{\sqrt{T}}$ | |
| Li et al. (2024) | NAG | convex | - | generalized $(L_0, L_1)$ | - | $\frac{1}{T^2}$ | |
| Yu et al. (2025) | SGD or RSAG | non-convex | relaxed affine (a.s.) | $(L_0, L_1)$ | **w.h.p** | $\frac{1}{T} + \frac{\sqrt{A}+\sqrt{C}}{\sqrt{T}}$ | |
| | | convex | | | | | |
| Thm 1 | NAG | convex | - | $(L_0, L_1, L_2)$ | - | $\frac{1}{T^2}$ | |
| Thm 2 | SNAG | convex | relaxed affine (a.s.) | $(L_0, L_1, L_2)$ | **w.h.p** | $\frac{1}{T^2} + \sqrt{\frac{A+B+C}{T}}$ | |
| Thm 3 | SNAG | convex | relaxed affine | smooth | $\mathbb{E}$ | $\frac{B+1}{T^2} + \sqrt{\frac{A+C}{T}}$ | |

[1] Indeed, Li et al. (2024) provided the probabilistic results for SGD while the dependence of the probability margin is the polynomial of $1/\delta$. In order to distinguish them from other high-probability results with dependence of $\log \frac{T}{\delta}$, we consider them as the expected results.

[2] "Alg.", "con." and "cond." are the shorthand of the words "algorithm", "convergence" and "condition".

[3] We ignore the dependence on the noise parameters order and the logarithm factors of the horizon $T$ in this table.

Table 2: Previous works related to NAG.

| | Algorithm | Convexity | Noise | Smoothness | Conv. type | Conv. rate |
|---|---|---|---|---|---|---|
| Nesterov (1983) | NAG | convex | - | Lipschitz | - | $\frac{1}{T^2}$ |
| Ghadimi & Lan (2016) | RSAG | non-convex | bounded variance | Lipschitz | $\mathbb{E}$ | $\frac{1}{T} + \sqrt{\frac{C}{T}}$ |
| | | convex | | | | $\frac{1}{T^2} + \sqrt{\frac{C}{T}}$ |
| Vaswani et al. (2019) | SNAG | convex | strongly growth | Lipschitz | $\mathbb{E}$ | $\frac{B+1}{T^2}$ |
| Li et al. (2024) | NAG | convex | - | generalized $(L_0, L_1)$ | - | $\frac{1}{T^2}$ |
| Hermant et al. (2025) | SNAG | convex | strongly growth | Lipschitz | a.s. | $\frac{B+1}{T^2}$ |
| Thm 1 | NAG | convex | - | $(L_0, L_1, L_2)$ | - | $\frac{1}{T^2}$ |
| Thm 2 | SNAG | convex | relaxed affine (a.s.) | $(L_0, L_1, L_2)$ | **w.h.p** | $\frac{1}{T^2} + \sqrt{\frac{A+B+C}{T}}$ |
| Thm 3 | SNAG | convex | relaxed affine | smooth | $\mathbb{E}$ | $\frac{B+1}{T^2} + \sqrt{\frac{A+C}{T}}$ |

[1] As discussed in Section 2, Vaswani et al. (2019); Hermant et al. (2025) analyzed ACDM, which is a variant of SNAG. However, ACDM is equivalent to SNAG with the specific step size setting in the convex case.

[2] "Con" and "cond" are the shorthand of the words "convergence" and "condition".

[3] We ignore the dependence on the noise parameters order and the logarithm factors of the horizon $T$ in this table.

**Lemma B.1** (Lemma 5 in (Taheri & Thrampoulidis, 2023)). *Let $F$ be in* (13) *and $\Phi$ be a two layer neural network with the activation function satisfying that there exist $L, \alpha, l > 0$, such that*

$$|\sigma''(t)| \leq L, \quad \alpha \leq \sigma'(t) \leq l, \quad \forall t \in \mathbb{R}.$$

*Then, $F$ is self-bounded of gradient and Hessian with constants $h = \frac{lR}{\sqrt{m}}, H = \frac{LR^2}{m^2} + \frac{l^2 R^2}{m}$, i.e.,*

$$\|\nabla F(\boldsymbol{x})\| \leq hF(\boldsymbol{x}), \quad \|\nabla^2 F(\boldsymbol{x})\| \leq HF(\boldsymbol{x}),$$

*where $R = \max_{i \in [n]} \|\boldsymbol{w}_i\| R = \max_{i \in [n]} \|\boldsymbol{x}_i\|$.*

In the next lemma, we denote $F^*$ is the minimum of $F(\boldsymbol{x})$ in (13), i.e., $F(\boldsymbol{x}) \geq F^*, \forall \boldsymbol{x} \in \mathbb{R}^{\tilde{d}}$.

**Lemma B.2.** *Under the condition of Lemma B.1, $F(\boldsymbol{x})$ in* (13) *is $(L_0, 0, L_2)$-smooth, where $L_0$ and $L_2$ are non-negative constants such that $L_2 = \max\{h, He\}, L_0 = L_2 F^*$ $L_2 \log L_2 \geq h \log H, L_0 = L_2 F^*$.*

*Proof.* For any $\|\boldsymbol{x} - \boldsymbol{y}\| \le 1/L_2$, define $\gamma(t) = t(\boldsymbol{y} - \boldsymbol{x}) + \boldsymbol{x}, t \in [0, 1]$. Then, for any $\mu \in [0, 1]$ we have,

$$
\begin{aligned}
F(\gamma(\mu)) &= \int_0^\mu \langle \nabla F(\gamma(t)), \boldsymbol{y} - \boldsymbol{x} \rangle \, dt + F(\boldsymbol{x}) \\
&\le \int_0^\mu \|\nabla F(\gamma(t))\| \cdot \|\boldsymbol{y} - \boldsymbol{x}\| \, dt + F(\boldsymbol{x}) \\
&\le h \|\boldsymbol{y} - \boldsymbol{x}\| \int_0^\mu F(\gamma(t)) dt + F(\boldsymbol{x}),
\end{aligned}
\tag{15}
$$

where the first inequality holds since Cauchy-Schwarz inequality and the second inequality follows from Lemma B.1. By Gronwall's inequality, we have

$$
F(\gamma(\mu)) \le F(\boldsymbol{x}) \cdot \exp(\mu h \|\boldsymbol{y} - \boldsymbol{x}\|), \quad \mu \in [0, 1].
\tag{16}
$$

Moreover, we have

$$
\nabla F(\boldsymbol{y}) - \nabla F(\boldsymbol{x}) = \nabla F(\gamma(1)) - \nabla F(\gamma(0)) = \int_0^1 \nabla^2 F(\gamma(t))(\boldsymbol{y} - \boldsymbol{x}) dt,
\tag{17}
$$

which implies,

$$
\begin{aligned}
\|\nabla F(\boldsymbol{y}) - \nabla F(\boldsymbol{x})\| &= \left\| \int_0^1 \nabla^2 F(\gamma(t))(\boldsymbol{y} - \boldsymbol{x}) dt \right\| \\
&\le \int_0^1 \|\nabla^2 F(\gamma(t))\| \|\boldsymbol{y} - \boldsymbol{x}\| \, dt \\
&\le H \|\boldsymbol{y} - \boldsymbol{x}\| \int_0^1 F(\gamma(t)) dt \\
&\le H \|\boldsymbol{y} - \boldsymbol{x}\| \int_0^1 F(\boldsymbol{x}) \cdot \exp(th \|\boldsymbol{y} - \boldsymbol{x}\|) dt.
\end{aligned}
\tag{18}
$$

Since $\|\boldsymbol{y} - \boldsymbol{x}\| \le \frac{1}{L_2}$, we have

$$
\begin{aligned}
\|\nabla F(\boldsymbol{y}) - \nabla F(\boldsymbol{x})\| &\le H F(\boldsymbol{x}) \exp(\frac{h}{L_2}) \|\boldsymbol{y} - \boldsymbol{x}\| \\
&= \left( H \exp(\frac{h}{L_2}) F^* + H \exp(\frac{h}{L_2}) (F(\boldsymbol{x}) - F^*) \right) \|\boldsymbol{y} - \boldsymbol{x}\|.
\end{aligned}
\tag{19}
$$

By the constraints that $L_2 = \max\{h, He\} \cancel{L_2 \log L_2 \ge h \log H}$, we have

$$
L_2 \ge H \exp(\frac{h}{L_2}).
$$

Combining with the fact that $F^*$ is positive for the exponential loss, we have

$$
\|\nabla F(\boldsymbol{y}) - \nabla F(\boldsymbol{x})\| \le (L_0 + L_2 (F(\boldsymbol{x}) - F^*)) \|\boldsymbol{y} - \boldsymbol{x}\|.
\tag{20}
$$

$\square$

## B.2 PHASE RETRIEVAL MODEL

We then provide the proof that the phase retrieval model in (14) satisfying $(L_0, L_1, L_2)$-smoothness condition. The following lemma is presented in (Chen et al., 2023).

**Lemma B.3.** *The function $f(\boldsymbol{x})$ in (14) belongs to $\mathcal{L}_{sym}^* \left( \frac{2}{3} \right)$, i.e., for any $\boldsymbol{x}, \boldsymbol{y} \in \mathbb{R}^d$,*

$$
\|\nabla f(\boldsymbol{x}) - \nabla f(\boldsymbol{y})\| \le \left( L_0' + L_1' \|\nabla f(\boldsymbol{x})\|^{\frac{2}{3}} + L_2' \|\boldsymbol{x} - \boldsymbol{y}\|^2 \right) \|\boldsymbol{x} - \boldsymbol{y}\|,
\tag{21}
$$

*where $L_0', L_1', L_2'$ are non-negative constants.*

Thus, we could derive Lemma B.4.

**Lemma B.4.** *Suppose that $f(\boldsymbol{x})$ is the phase retrieval model defined in* (21). *Then, $f(\boldsymbol{x})$ is $(L_0, L_1, 0)$-smooth, where $L_0 = L_0' + L_2'/L_1^2$ and $L_1 = L_1'$.*

*Proof.* By (21), for any $\boldsymbol{x}, \boldsymbol{y} \in \mathbb{R}^d$ such that $\|\boldsymbol{x} - \boldsymbol{y}\| \leq \frac{1}{L_1}$, we have

$$
\|\nabla f(\boldsymbol{x}) - \nabla f(\boldsymbol{y})\| \leq \left( L_0' + L_1' \|\nabla f(\boldsymbol{x})\|^{\frac{2}{3}} + L_2'/L_1^2 \right) \|\boldsymbol{x} - \boldsymbol{y}\|
$$

$$
= \left( L_0 + L_1 \|\nabla f(\boldsymbol{x})\|^{\frac{2}{3}} \right) \|\boldsymbol{x} - \boldsymbol{y}\|.
$$

$\square$

## C  COMPLEMENTARY LEMMAS

The following lemma characterizes the relationship between the gradient and the function value gap under the smoothness condition. Refer to (Attia & Koren, 2023) for a proof.

**Lemma C.1.** *Let $f(\cdot): \mathbb{R}^d \to \mathbb{R}$ be an L-smooth function with minimum $f^*$. Then, we have*

$$
\|\nabla f(\boldsymbol{x}_t)\|^2 \leq 2L \left( f(\boldsymbol{x}_t) - f^* \right).
$$

Lemma C.2 and Lemma C.3 are the key to the analysis for $(L_0, L_1, L_2)$-smooth functions.

**Lemma C.2.** *If $f(\cdot): \mathbb{R}^d \to \mathbb{R}$ is $(L_0, L_1, L_2)$-smooth with minimum $f^*$, then for any $\boldsymbol{x}, \boldsymbol{y} \in \mathbb{R}^d$ such that $\|\boldsymbol{x} - \boldsymbol{y}\| \leq \min\{1/L_1, 1/L_2\}$, we have*

$$
f(\boldsymbol{y}) \leq f(\boldsymbol{x}) + \langle \nabla f(\boldsymbol{x}), \boldsymbol{y} - \boldsymbol{x} \rangle + \frac{L_0 + L_1 \|\nabla f(\boldsymbol{x})\|^p + L_2 \left( f(\boldsymbol{x}) - f(\boldsymbol{x}^*) \right)^q}{2} \|\boldsymbol{x} - \boldsymbol{y}\|^2.
$$

*Proof.*

$$
f(\boldsymbol{y}) - f(\boldsymbol{x}) - \langle \boldsymbol{y} - \boldsymbol{x}, \nabla f(\boldsymbol{x}) \rangle = \int_0^1 \langle \nabla f(\theta \boldsymbol{y} + (1 - \theta) \boldsymbol{x}) - \nabla f(\boldsymbol{x}), \boldsymbol{y} - \boldsymbol{x} \rangle \, d\theta
$$

$$
\leq \int_0^1 \|\nabla f(\theta \boldsymbol{y} + (1 - \theta) \boldsymbol{x}) - \nabla f(\boldsymbol{x})\| \cdot \|\boldsymbol{x} - \boldsymbol{y}\| \, d\theta
$$

$$
\leq \int_0^1 \theta \cdot \left( L_0 + L_1 \|\nabla f(\boldsymbol{x})\|^p + L_2 \left( f(\boldsymbol{x}) - f^* \right)^q \right) \|\boldsymbol{x} - \boldsymbol{y}\|^2 \, d\theta
$$

$$
= \frac{L_0 + L_1 \|\nabla f(\boldsymbol{x})\|^p + L_2 \left( f(\boldsymbol{x}) - f^* \right)^q}{2} \|\boldsymbol{x} - \boldsymbol{y}\|^2,
$$

where the first inequality holds since Cauchy-Schwarz inequality and the second inequality follows from the definition of $(L_0, L_1, L_2)$-smoothness. $\square$

**Lemma C.3.** *If $f(\cdot): \mathbb{R}^d \to \mathbb{R}$ is $(L_0, L_1, L_2)$-smooth with minimum $f^*$, then we have*

$$
\|\nabla f(\boldsymbol{x}_t)\|^2 \leq 4L_0 \triangle_t + (2 - p) \left( (2p)^{\frac{p}{2}} 2L_1 \triangle_t \right)^{\frac{2}{2-p}} + 4L_2 \triangle_t^{q+1} + 8 \left( L_1 + L_2 \right)^2 \triangle_t^2, \quad (22)
$$

*where $\triangle_t = f(\boldsymbol{x}_t) - f^*$.*

*Proof.* Let $\boldsymbol{x} = \boldsymbol{x}_t - \frac{1}{L_0 + L_1(\|\nabla f(\boldsymbol{x}_t)\| + \|\nabla f(\boldsymbol{x}_t)\|^p) + L_2(\|\nabla f(\boldsymbol{x}_t)\| + \triangle_t^q)} \nabla f(\boldsymbol{x}_t)$. It is easy to verify $\|\boldsymbol{x} - \boldsymbol{x}_t\| \leq \min\{1/L_1, 1/L_2\}$. By Lemma C.2, we have

$$
\begin{aligned}
f(\boldsymbol{x}) \leq & f(\boldsymbol{x}_t) + \langle \nabla f(\boldsymbol{x}_t), \boldsymbol{x} - \boldsymbol{x}_t \rangle + \frac{L_0 + L_1 \|\nabla f(\boldsymbol{x}_t)\|^p + L_2 \triangle_t^q}{2} \|\boldsymbol{x} - \boldsymbol{x}_t\|^2 \\
= & f(\boldsymbol{x}_t) - \frac{\|\nabla f(\boldsymbol{x}_t)\|^2}{L_0 + L_1(\|\nabla f(\boldsymbol{x}_t)\| + \|\nabla f(\boldsymbol{x}_t)\|^p) + L_2(\|\nabla f(\boldsymbol{x}_t)\| + \triangle_t^q)} \\
& + \frac{L_0 + L_1 \|\nabla f(\boldsymbol{x}_t)\|^p + L_2 \triangle_t^q}{2(L_0 + L_1(\|\nabla f(\boldsymbol{x}_t)\| + \|\nabla f(\boldsymbol{x}_t)\|^p) + L_2(\|\nabla f(\boldsymbol{x}_t)\| + \triangle_t^q))^2} \cdot \|\nabla f(\boldsymbol{x}_t)\|^2 \\
\leq & f(\boldsymbol{x}_t) - \frac{\|\nabla f(\boldsymbol{x}_t)\|^2}{L_0 + L_1(\|\nabla f(\boldsymbol{x}_t)\| + \|\nabla f(\boldsymbol{x}_t)\|^p) + L_2(\|\nabla f(\boldsymbol{x}_t)\| + \triangle_t^q)} \\
& + \frac{\|\nabla f(\boldsymbol{x}_t)\|^2}{2(L_0 + L_1(\|\nabla f(\boldsymbol{x}_t)\| + \|\nabla f(\boldsymbol{x}_t)\|^p) + L_2(\|\nabla f(\boldsymbol{x}_t)\| + \triangle_t^q))},
\end{aligned}
$$

which implies

$$
\frac{\|\nabla f(\boldsymbol{x}_t)\|^2}{2(L_0 + L_1(\|\nabla f(\boldsymbol{x}_t)\| + \|\nabla f(\boldsymbol{x}_t)\|^p) + L_2(\|\nabla f(\boldsymbol{x}_t)\| + \triangle_t^q))} \leq f(\boldsymbol{x}_t) - f(\boldsymbol{x}) \leq \triangle_t.
$$

Re-arranging the above inequality, we obtain that

$$
\|\nabla f(\boldsymbol{x}_t)\|^2 \leq 2L_0 \triangle_t + 2L_1 \|\nabla f(\boldsymbol{x}_t)\|^p \triangle_t + 2L_2 \triangle_t^{q+1} + 2(L_1 + L_2) \|\nabla f(\boldsymbol{x}_t)\| \cdot \triangle_t.
$$

When $p > 0$, ~~Then,~~ applying Young's inequality, we have

$$
\begin{aligned}
\|\nabla f(\boldsymbol{x}_t)\|^2 \leq & 2L_0 \triangle_t + \frac{1}{4} \|\nabla f(\boldsymbol{x}_t)\|^2 + \left(1 - \frac{p}{2}\right) \left((2p)^{\frac{p}{2}} 2L_1 \triangle_t\right)^{\frac{2}{2-p}} \\
& + 2L_2 \triangle_t^{q+1} + \frac{1}{4} \|\nabla f(\boldsymbol{x}_t)\|^2 + 4(L_1 + L_2)^2 \triangle_t^2.
\end{aligned} \tag{23}
$$

Note that (23) still holds when $p = 0$ since $0^0 = 1$. Hence,

$$
\|\nabla f(\boldsymbol{x}_t)\|^2 \leq 4L_0 \triangle_t + (2 - p) \left((2p)^{\frac{p}{2}} 2L_1 \triangle_t\right)^{\frac{2}{2-p}} + 4L_2 \triangle_t^{q+1} + 8(L_1 + L_2)^2 \triangle_t^2.
$$

$\square$

**Corollary 1.** *Let $f(\cdot)$ be an $(L_0, L_1, L_2)$-smooth function with minimum $f^*$. If $f(\boldsymbol{x}_t) - f^* \leq G$, then we have*

$$
\|\nabla f(\boldsymbol{x}_t)\|^2 \leq g(G),
$$

*where $g(\cdot) : \mathbb{R} \to \mathbb{R}$ is defined as*

$$
g(\mu) = 4L_0 \mu + (2 - p) \left((2p)^{\frac{p}{2}} 2L_1 \mu\right)^{\frac{2}{2-p}} + 4L_2 \mu^{q+1} + 8(L_1 + L_2)^2 \mu^2. \tag{24}
$$

The following lemma plays crucial role in our probabilistic analysis. Refer to (Li & Orabona, 2020) for a proof.

**Lemma C.4** (Lemma 1 in (Li & Orabona, 2020)). *Assume that $\{Z_t\}_{t \in [T]}$ is a martingale difference sequence with respect to $\gamma_1, \gamma_2, \cdots, \gamma_T$ and $\mathbb{E}_t[\exp(Z_t^2/\sigma_t^2)] \leq \exp(1)$ for all $1 \leq t \leq T$, where $\sigma_t$ is a sequence of measurable random variables with respect to $\gamma_1, \gamma_2, \cdots, \gamma_{t-1}$. Then, for any fixed $\lambda > 0$ and $\delta \in (0, 1)$, with probability at least $1 - \delta$, we have*

$$
\sum_{t=1}^{T} Z_t \leq \frac{3\lambda}{4} \sum_{t=1}^{T} \sigma_t^2 + \frac{1}{\lambda} \log \frac{1}{\delta}.
$$

# D PROOF OF THEOREM 1

We first present the explicit expressions of $\mathcal{C}_1, \mathcal{F}_1, \mathcal{L}_1$,

$$\mathcal{C}_1 = \triangle_0^{ag} + \frac{1}{2} \left\| \boldsymbol{x}_0 - \boldsymbol{x}^* \right\|^2, \tag{25}$$

$$\mathcal{F}_1 = \mathcal{C}_1 + \frac{1}{L_1 + L_2} \sqrt{g(\mathcal{C}_1)} + \frac{L_0 + L_1 \left( g(\mathcal{C}_1) \right)^{\frac{p}{2}} + L_2 \mathcal{C}_1^q}{2 \left( L_1 + L_2 \right)^2}, \tag{26}$$

$$\mathcal{L}_1 = 2 \left( L_0 + L_1 \left( (g(\mathcal{F}_1))^{\frac{1}{2}} + (g(\mathcal{F}_1))^{\frac{p}{2}} \right) + L_2 \left( (g(\mathcal{F}_1))^{\frac{1}{2}} + \mathcal{F}_1^q \right) + 8\mathcal{C}_1^2 \left( L_1 + L_2 \right)^4 \right), \tag{27}$$

where $g(\cdot) : \mathbb{R} \to \mathbb{R}$ is a function defined in (24). Then, we will provide some useful lemmas. Lemma D.1 follows from the analysis in (Ghadimi & Lan, 2016) and is derived from the iteration steps in Algorithm 1.

**Lemma D.1.** *Let $\left\{ \boldsymbol{x}_k^{md} \right\}_{k \in [T]}$ and $\left\{ \boldsymbol{x}_k^{ag} \right\}_{k \in [T]}$ be the two sequences generated by Algorithm 1. Then we have for all $k \in [T]$,*

$$\left\| \boldsymbol{x}_k^{md} - \boldsymbol{x}_{k-1}^{ag} \right\|^2 \le \frac{1}{A_k \cdot A_{k-1}} \sum_{i=1}^{k-1} \frac{A_i^2 \cdot (\lambda_i - \beta)^2}{A_i - A_{i-1}} \left\| \nabla f(\boldsymbol{x}_i^{md}) \right\|^2.$$

*Proof.* From Algorithm 1, we have

$$\boldsymbol{x}_k^{ag} - \boldsymbol{x}_k = \boldsymbol{x}_k^{md} - \beta \nabla f(\boldsymbol{x}_k^{md}) - \boldsymbol{x}_{k-1} + \lambda_k \nabla f(\boldsymbol{x}_k^{md})$$

$$= \frac{A_{k-1}}{A_k} \left( \boldsymbol{x}_{k-1}^{ag} - \boldsymbol{x}_{k-1} \right) + \left( \lambda_k - \beta \right) \nabla f(\boldsymbol{x}_k^{md}).$$

Since $\boldsymbol{x}_0^{ag} = \boldsymbol{x}_0$, we obtain that

$$\boldsymbol{x}_k^{ag} - \boldsymbol{x}_k = \frac{1}{A_k} \sum_{i=1}^{k} A_i \left( \lambda_i - \beta \right) \nabla f(\boldsymbol{x}_i^{md}),$$

which implies

$$\left\| \boldsymbol{x}_k^{ag} - \boldsymbol{x}_k \right\| \le \frac{1}{A_k} \sum_{i=1}^{k} A_i \left| \lambda_i - \beta \right| \cdot \left\| \nabla f(\boldsymbol{x}_i^{md}) \right\|. \tag{28}$$

Applying the iteration step in Algorithm 1 again, we have

$$\boldsymbol{x}_k^{md} - \boldsymbol{x}_{k-1}^{ag} = \left( 1 - \frac{A_{k-1}}{A_k} \right) \left( \boldsymbol{x}_{k-1} - \boldsymbol{x}_{k-1}^{ag} \right).$$

Combining with (28), we have

$$\left\| \boldsymbol{x}_k^{md} - \boldsymbol{x}_{k-1}^{ag} \right\| \le \left( 1 - \frac{A_{k-1}}{A_k} \right) \cdot \frac{1}{A_{k-1}} \sum_{i=1}^{k-1} A_i \left| \lambda_i - \beta \right| \cdot \left\| \nabla f(\boldsymbol{x}_i^{md}) \right\|. \tag{29}$$

Using the fact that

$$\sum_{i=1}^{k-1} A_i \cdot \left( 1 - \frac{A_{i-1}}{A_i} \right) = A_{k-1} - A_0,$$

we have

$$\left\| \boldsymbol{x}_k^{md} - \boldsymbol{x}_{k-1}^{ag} \right\|^2$$

$$\le \left( 1 - \frac{A_{k-1}}{A_k} \right)^2 \cdot \frac{1}{A_{k-1}^2} \left[ \sum_{i=1}^{k-1} A_i \left( 1 - \frac{A_{i-1}}{A_i} \right) \cdot \frac{|\lambda_i - \beta|}{1 - \frac{A_{i-1}}{A_i}} \left\| \nabla f(\boldsymbol{x}_i^{md}) \right\| \right]^2$$

$$\le \left( 1 - \frac{A_{k-1}}{A_k} \right)^2 \cdot \frac{A_{k-1} - A_0}{A_{k-1}^2} \sum_{i=1}^{k-1} A_i \left( 1 - \frac{A_{i-1}}{A_i} \right) \frac{A_i^2 \cdot (\lambda_i - \beta)^2}{(A_i - A_{i-1})^2} \left\| \nabla f(\boldsymbol{x}_i^{md}) \right\|^2$$

$$\le \frac{1}{A_k \cdot A_{k-1}} \sum_{i=1}^{k-1} \frac{A_i^2 \cdot (\lambda_i - \beta)^2}{A_i - A_{i-1}} \left\| \nabla f(\boldsymbol{x}_i^{md}) \right\|^2,$$

where the second inequality follows from Jensen's inequality and the last inequality holds since Lemma 4.1. $\qquad\square$

In the next Lemma, we assume the function value gap is bounded. Under this assumption, the analysis for $(L_0, L_1, L_2)$-smooth and $L$ smooth objective functions becomes similar. With reference to (Nesterov, 1983; Ghadimi & Lan, 2016; d'Aspremont et al., 2021), we provide the following analysis with a step size specific to the novel smoothness condition.

**Lemma D.2.** *Suppose that* $f\left(\boldsymbol{x}_l^{md}\right) - f^* \leq \mathcal{F}_1, \forall l \in [t]$. *Then, under the conditions of Theorem 1, we have*

$$\triangle_t^{ag} \leq \frac{\mathcal{C}_1}{A_t \beta},$$

*where $\mathcal{C}_1$ is a constant related to the initial point and is defined in* (25).

*Proof.* By Corollary 1 and the assumption that $\triangle_l^{md} \leq \mathcal{F}_1, \forall l \in [t]$, we have $\left\|\nabla f(\boldsymbol{x}_l^{md})\right\|^2 \leq g(\mathcal{F}_1), \forall l \in [t]$. Therefore,

$$\left\|\boldsymbol{x}_l^{ag} - \boldsymbol{x}_l^{md}\right\| = \beta \left\|\nabla f(\boldsymbol{x}_l^{md})\right\| \leq \beta \sqrt{g(\mathcal{F}_1)} \leq \min\left\{1/L_1, 1/L_2\right\},$$

where the last inequality holds since $\beta = \frac{1}{\mathcal{L}_1}$ with $\mathcal{L}_1$ defined in (27). By Lemma C.2, we have

$$f\left(\boldsymbol{x}_l^{ag}\right) \leq f\left(\boldsymbol{x}_l^{md}\right) + \left\langle\nabla f(\boldsymbol{x}_l^{md}), \boldsymbol{x}_l^{ag} - \boldsymbol{x}_l^{md}\right\rangle + \frac{L_0 + L_1\left(g\left(\mathcal{F}_1\right)\right)^{\frac{p}{2}} + L_2 \mathcal{F}_1^q}{2}\left\|\boldsymbol{x}_l^{ag} - \boldsymbol{x}_l^{md}\right\|^2$$

$$= f\left(\boldsymbol{x}_l^{md}\right) - \beta\left\|\nabla f(\boldsymbol{x}_l^{md})\right\|^2 + \frac{L_0 + L_1\left(g\left(\mathcal{F}_1\right)\right)^{\frac{p}{2}} + L_2 \mathcal{F}_1^q}{2}\beta^2\left\|\nabla f(\boldsymbol{x}_l^{md})\right\|^2. \quad (30)$$

By the convexity and the iteration step in Algorithm 1, we have

$$f\left(\boldsymbol{x}_l^{md}\right) - \left[\frac{A_{l-1}}{A_l} \cdot f\left(\boldsymbol{x}_{l-1}^{ag}\right) + \left(1 - \frac{A_{l-1}}{A_l}\right) \cdot f^*\right]$$

$$= \left(1 - \frac{A_{l-1}}{A_l}\right) \cdot \left[f\left(\boldsymbol{x}_l^{md}\right) - f^*\right] + \frac{A_{l-1}}{A_l} \cdot \left[f\left(\boldsymbol{x}_l^{md}\right) - f\left(\boldsymbol{x}_{l-1}^{ag}\right)\right]$$

$$\leq \left(1 - \frac{A_{l-1}}{A_l}\right) \cdot \left\langle\nabla f(\boldsymbol{x}_l^{md}), \boldsymbol{x}_l^{md} - \boldsymbol{x}^*\right\rangle + \frac{A_{l-1}}{A_l} \cdot \left\langle\nabla f(\boldsymbol{x}_l^{md}), \boldsymbol{x}_l^{md} - \boldsymbol{x}_{l-1}^{ag}\right\rangle$$

$$= \left\langle\nabla f(\boldsymbol{x}_l^{md}), \left(1 - \frac{A_{l-1}}{A_l}\right) \cdot \left(\boldsymbol{x}_l^{md} - \boldsymbol{x}^*\right) + \frac{A_{l-1}}{A_l} \cdot \left(\boldsymbol{x}_l^{md} - \boldsymbol{x}_{l-1}^{ag}\right)\right\rangle$$

$$= \left(1 - \frac{A_{l-1}}{A_l}\right) \cdot \left\langle\nabla f(\boldsymbol{x}_l^{md}), \boldsymbol{x}_{l-1} - \boldsymbol{x}^*\right\rangle. \quad (31)$$

Combining (30) and (31), we obtain that

$$f\left(\boldsymbol{x}_l^{ag}\right) \leq \frac{A_{l-1}}{A_l} \cdot f\left(\boldsymbol{x}_{l-1}^{ag}\right) + \left(1 - \frac{A_{l-1}}{A_l}\right) \cdot f^* + \left(1 - \frac{A_{l-1}}{A_l}\right) \cdot \left\langle\nabla f(\boldsymbol{x}_l^{md}), \boldsymbol{x}_{l-1} - \boldsymbol{x}^*\right\rangle$$

$$- \beta\left\|\nabla f(\boldsymbol{x}_l^{md})\right\|^2 + \frac{L_0 + L_1\left(g\left(\mathcal{F}_1\right)\right)^{\frac{p}{2}} + L_2 \mathcal{F}_1^q}{2}\beta^2\left\|\nabla f(\boldsymbol{x}_l^{md})\right\|^2. \quad (32)$$

Also,

$$\left\|\boldsymbol{x}_{l-1} - \boldsymbol{x}^*\right\|^2 - 2\lambda_l\left\langle\nabla f(\boldsymbol{x}_l^{md}), \boldsymbol{x}_{l-1} - \boldsymbol{x}^*\right\rangle + \lambda_l^2\left\|\nabla f(\boldsymbol{x}_l^{md})\right\|^2$$

$$= \left\|\boldsymbol{x}_{l-1} - \lambda_l\nabla f(\boldsymbol{x}_l^{md}) - \boldsymbol{x}^*\right\|^2 = \left\|\boldsymbol{x}_l - \boldsymbol{x}^*\right\|^2.$$

Hence,

$$\left\langle\nabla f(\boldsymbol{x}_l^{md}), \boldsymbol{x}_{l-1} - \boldsymbol{x}^*\right\rangle = \frac{1}{2\lambda_l}\left[\left\|\boldsymbol{x}_{l-1} - \boldsymbol{x}^*\right\|^2 - \left\|\boldsymbol{x}_l - \boldsymbol{x}^*\right\|^2\right] + \frac{\lambda_l}{2}\left\|\nabla f(\boldsymbol{x}_l^{md})\right\|^2. \quad (33)$$

Substituting (33) into (32), we have

$$f\left(\boldsymbol{x}_l^{ag}\right) \leq \frac{A_{l-1}}{A_l} \cdot f\left(\boldsymbol{x}_{l-1}^{ag}\right) + \left(1 - \frac{A_{l-1}}{A_l}\right) \cdot f^* + \frac{A_l - A_{l-1}}{2A_l \cdot \lambda_l}\left[\|\boldsymbol{x}_{l-1} - \boldsymbol{x}^*\|^2 - \|\boldsymbol{x}_l - \boldsymbol{x}^*\|^2\right]$$

$$- \beta\left(1 - \frac{\left(L_0 + L_1\left(g\left(\mathcal{F}_1\right)\right)^{\frac{p}{2}} + L_2\mathcal{F}_1^q\right)\beta}{2} - \frac{\lambda_l\left(A_l - A_{l-1}\right)}{2\beta A_l}\right)\left\|\nabla f(\boldsymbol{x}_l^{md})\right\|^2. \quad (34)$$

By the constraint of $\lambda_l$ in (7) and applying Lemma 4.1, we obtain that

$$\lambda_l \cdot \frac{A_l - A_{l-1}}{A_l} = \beta \cdot \frac{(A_l - A_{l-1})^2}{A_l} = \beta\frac{(B_l - B_{l-1})^2}{B_l + 1/\beta} = \beta\frac{B_l}{B_l + 1/\beta} < \beta.$$

Also, recalling the constraint of $\beta$ in (7), we have ~~and~~

$$\left(L_0 + L_1\left(g\left(\mathcal{F}_1\right)\right)^{\frac{p}{2}} + L_2\mathcal{F}_1^q\right)\beta \leq \frac{1}{2}.$$

Therefore,

$$1 - \frac{\left(L_0 + L_1\left(g\left(\mathcal{F}_1\right)\right)^{\frac{p}{2}} + L_2\mathcal{F}_1^q\right)\beta}{2} - \frac{\lambda_l\left(A_l - A_{l-1}\right)}{2\beta A_l} \geq \frac{1}{4}.$$

Combining with (34) and reorganizing the terms, we have

$$\triangle_l^{ag} \leq -\frac{1}{4}\beta\left\|\nabla f(\boldsymbol{x}_l^{md})\right\|^2 + \frac{A_{l-1}}{A_l} \cdot \triangle_{l-1}^{ag} + \frac{A_l - A_{l-1}}{2A_l \cdot (A_l - A_{l-1})\beta}\left[\|\boldsymbol{x}_{l-1} - \boldsymbol{x}^*\|^2 - \|\boldsymbol{x}_l - \boldsymbol{x}^*\|^2\right]$$

$$= -\frac{1}{4}\beta\left\|\nabla f(\boldsymbol{x}_l^{md})\right\|^2 + \frac{A_{l-1}}{A_l} \cdot \triangle_{l-1}^{ag} + \frac{1}{2\beta A_l}\left[\|\boldsymbol{x}_{l-1} - \boldsymbol{x}^*\|^2 - \|\boldsymbol{x}_l - \boldsymbol{x}^*\|^2\right]. \quad (35)$$

With both sides of the above inequality multiplying $A_l$, we have

$$A_l\triangle_l^{ag} + \frac{1}{2\beta}\|\boldsymbol{x}_l - \boldsymbol{x}^*\|^2 \leq A_{l-1}\triangle_{l-1}^{ag} + \frac{1}{2\beta}\|\boldsymbol{x}_{l-1} - \boldsymbol{x}^*\|^2 - \frac{1}{4}\beta A_l\left\|\nabla f(\boldsymbol{x}_l^{md})\right\|^2. \quad (36)$$

Summing up over $l \in [t]$ and re-arranging the inequality, we obtain that

$$\triangle_t^{ag} \leq \frac{A_0}{A_t}\triangle_0^{ag} + \frac{1}{2\beta A_t}\|\boldsymbol{x}_0 - \boldsymbol{x}^*\|^2 = \frac{1}{A_t\beta}\triangle_0^{ag} + \frac{1}{2\beta A_t}\|\boldsymbol{x}_0 - \boldsymbol{x}^*\|^2$$

$$= \frac{1}{A_t\beta}\left(\triangle_0^{ag} + \frac{1}{2}\|\boldsymbol{x}_0 - \boldsymbol{x}^*\|^2\right). \quad (37)$$

$\square$

Based on the proof for Lemma D.2, we will prove the bound of $f\left(\boldsymbol{x}_t^{md}\right) - f^*$ for all $t \in [T]$, using an induction argument.

**Lemma D.3.** *Under the conditions of Theorem 1, we have*

$$f\left(\boldsymbol{x}_t^{md}\right) - f^* \leq \mathcal{F}_1, \quad \forall t \in [T],$$

*where $\mathcal{F}_1$ is defined in (26).*

*Proof.* It is apparent that $f\left(\boldsymbol{x}_1^{md}\right) - f^* = f\left(\boldsymbol{x}_0^{ag}\right) - f^* \leq \mathcal{F}_1$ since $\boldsymbol{x}_0^{ag} = \boldsymbol{x}_0$. Suppose that for some $t \in [T]$,

$$f\left(\boldsymbol{x}_l^{md}\right) - f^* \leq \mathcal{F}_1, \forall l \in [t].$$

Next, we will bound $f\left(\boldsymbol{x}_{t+1}^{md}\right) - f^*$. By Lemma D.1, we have

$$\left\|\boldsymbol{x}_{t+1}^{md} - \boldsymbol{x}_t^{ag}\right\|^2 \leq \frac{1}{A_{t+1} \cdot A_t}\sum_{i=1}^{t}\frac{A_i^2 \cdot (\lambda_i - \beta)^2}{A_i - A_{i-1}}\left\|\nabla f(\boldsymbol{x}_i^{md})\right\|^2$$

$$\leq \sum_{i=1}^{t}\frac{(A_i - A_{i-1} - 1)^2}{A_i - A_{i-1}}\beta^2\left\|\nabla f(\boldsymbol{x}_i^{md})\right\|^2$$

$$\leq \beta^2\sum_{i=1}^{t}(A_i - A_{i-1})\left\|\nabla f(\boldsymbol{x}_i^{md})\right\|^2,$$

where the second and the third inequalities follow from Lemma 4.1. Applying Lemma 4.1 again and using the fact that $A_i = B_i + 1/\beta, \forall i \in [T]$, we have

$$\left\| \boldsymbol{x}_{t+1}^{md} - \boldsymbol{x}_t^{ag} \right\|^2 \leq \beta^2 \sum_{i=1}^{t} \sqrt{A_i} \left\| \nabla f(\boldsymbol{x}_i^{md}) \right\|^2 \leq \beta^{\frac{5}{2}} \sum_{i=1}^{t} A_i \left\| \nabla f(\boldsymbol{x}_i^{md}) \right\|^2 . \tag{38}$$

Since the assumption that $f\left(\boldsymbol{x}_l^{md}\right) - f^* \leq \mathcal{F}_1, \forall l \in [t]$, (36) holds here for all $l \in [t]$. Therefore, summing up (36) over $l \in [t]$, we have

$$\frac{1}{4}\beta \sum_{i=1}^{t} A_i \left\| \nabla f(\boldsymbol{x}_i^{md}) \right\|^2 \leq A_0 \triangle_0^{ag} + \frac{1}{2\beta} \left\| \boldsymbol{x}_0 - \boldsymbol{x}^* \right\|^2 . \tag{39}$$

Combining (38) and (39) and the constraint of $\beta$, we obtain that

$$\left\| \boldsymbol{x}_{t+1}^{md} - \boldsymbol{x}_t^{ag} \right\|^2 \leq \sqrt{\beta} \cdot 4 \left( \triangle_0^{ag} + \frac{1}{2} \left\| \boldsymbol{x}_0 - \boldsymbol{x}^* \right\|^2 \right) \leq \frac{1}{(L_1 + L_2)^2} .$$

Applying Lemma C.2 again, we have

$$f\left(\boldsymbol{x}_{t+1}^{md}\right)$$

$$\leq f\left(\boldsymbol{x}_t^{ag}\right) + \left\langle \nabla f(\boldsymbol{x}_t^{ag}), \boldsymbol{x}_{t+1}^{md} - \boldsymbol{x}_t^{ag} \right\rangle + \frac{L_0 + L_1 \left\| \nabla f(\boldsymbol{x}_t^{ag}) \right\|^p + L_2 \left(\triangle_t^{ag}\right)^q}{2} \left\| \boldsymbol{x}_{t+1}^{md} - \boldsymbol{x}_t^{ag} \right\|^2$$

$$\leq f\left(\boldsymbol{x}_t^{ag}\right) + \left\| \nabla f(\boldsymbol{x}_t^{ag}) \right\| \cdot \left\| \boldsymbol{x}_{t+1}^{md} - \boldsymbol{x}_t^{ag} \right\| + \frac{L_0 + L_1 \left\| \nabla f(\boldsymbol{x}_t^{ag}) \right\|^p + L_2 \left(\triangle_t^{ag}\right)^q}{2} \left\| \boldsymbol{x}_{t+1}^{md} - \boldsymbol{x}_t^{ag} \right\|^2$$

$$\leq f\left(\boldsymbol{x}_t^{ag}\right) + \frac{1}{L_1 + L_2} \left\| \nabla f(\boldsymbol{x}_t^{ag}) \right\| + \frac{L_0 + L_1 \left\| \nabla f(\boldsymbol{x}_t^{ag}) \right\|^p + L_2 \left(\triangle_t^{ag}\right)^q}{2 \left(L_1 + L_2\right)^2} , \tag{40}$$

where the second inequality holds since Cauchy-Schwarz inequality. Further, considering the assumption that $f\left(\boldsymbol{x}_l^{md}\right) - f^* \leq \mathcal{F}_1, \forall l \in [t]$, (37) holds here. Noting that $A_t \beta = \left(B_t + \frac{1}{\beta}\right) \cdot \beta \geq 1$, we could deduce

$$\triangle_t^{ag} \leq \triangle_0^{ag} + \frac{1}{2} \left\| \boldsymbol{x}_0 - \boldsymbol{x}^* \right\|^2 = \mathcal{C}_1 . \tag{41}$$

Plugging (41) into (40), subtracting $f^*$ from both sides and applying Corollary 1, we obtain that

$$\triangle_{t+1}^{md} \leq \mathcal{C}_1 + \frac{1}{L_1 + L_2} \sqrt{g(\mathcal{C}_1)} + \frac{L_0 + L_1 \left(g(\mathcal{C}_1)\right)^{\frac{p}{2}} + L_2 \mathcal{C}_1^q}{2 \left(L_1 + L_2\right)^2} = \mathcal{F}_1 ,$$

where $g(\cdot)$ is the function defined in (24). Therefore, the induction is complete and we obtain the desired result. $\qquad \square$

Now we are ready to obtain the main convergence result.

*Proof of Theorem 1.* Noting that $\triangle_t^{md} \leq \mathcal{F}_1, \forall t \in [T]$ proved in Lemma D.3, we could apply Lemma D.2 and obtain that

$$\triangle_T^{ag} \leq \frac{1}{A_T \beta} \left( \triangle_0^{ag} + \frac{1}{2} \left\| \boldsymbol{x}_0 - \boldsymbol{x}^* \right\|^2 \right) = \frac{\mathcal{C}_1 \cdot \mathcal{L}_1}{A_T} .$$

Applying Lemma 4.1, we obtain that

$$\triangle_T^{ag} \leq \frac{4\mathcal{C}_1 \cdot \left( 2 \left( L_0 + L_1 \left( (g(\mathcal{F}_1))^{\frac{1}{2}} + (g(\mathcal{F}_1))^{\frac{p}{2}} \right) + L_2 \left( (g(\mathcal{F}_1))^{\frac{1}{2}} + \mathcal{F}_1^q \right) + 8\mathcal{C}_1^2 \left( L_1 + L_2 \right)^4 \right) \right)}{T^2} . \tag{42}$$

$\qquad \square$

# E   PROOF OF THEOREM 2

We first introduce some notations used in Theorem 2, i.e.,

$$\mathcal{M} = \sqrt{A\mathcal{F}_2 + Bg\left(\mathcal{F}_2\right) + C}, \qquad\qquad \mathcal{G}_1 = \max\left\{\mathcal{G}_{1,1}, \mathcal{G}_{1,2}, \mathcal{G}_{1,3}, \mathcal{G}_{1,4}\right\},$$

$$\mathcal{G}_2 = (595)^{\frac{2}{5}}\left(L_1 + L_2\right)^{\frac{4}{5}}\mathcal{M}^{\frac{4}{5}}\left(\log\frac{T\mathrm{e}}{\delta}\right)^{\frac{2}{5}}, \quad \mathcal{G}_3 = (595)^{\frac{2}{3}}\left(L_1 + L_2\right)^{\frac{4}{3}}\mathcal{M}^{\frac{4}{3}}\left(\log\frac{T\mathrm{e}}{\delta}\right)^{\frac{2}{3}}, \quad (43)$$

where

$$\mathcal{G}_{1,1} = L_1\left(\sqrt{g\left(\mathcal{F}_2\right)} + \mathcal{M}\sqrt{\log\frac{T\mathrm{e}}{\delta}}\right), \qquad \mathcal{G}_{1,2} = L_2\left(\sqrt{g\left(\mathcal{F}_2\right)} + \mathcal{M}\sqrt{\log\frac{T\mathrm{e}}{\delta}}\right),$$

$$\mathcal{G}_{1,3} = 4\left(L_0 + L_1\left(g\left(\mathcal{F}_2\right)\right)^{\frac{p}{2}} + L_2\mathcal{F}_2^q\right), \qquad \mathcal{G}_{1,4} = 4624\left(L_1 + L_2\right)^4\mathcal{C}_2^2. \qquad (44)$$

Furthermore,

$$\mathcal{F}_2 = \mathcal{H} + \frac{\sqrt{g(\mathcal{H})}}{L_1 + L_2} + \frac{L_0 + L_1\left(g(\mathcal{H})\right)^{\frac{p}{2}} + L_2\mathcal{H}^q}{2\left(L_1 + L_2\right)^2}, \qquad (45)$$

with the notations

$$\mathcal{C}_2 = \triangle_0^{ag} + 2\left\|\boldsymbol{x}_0 - \boldsymbol{x}^*\right\|^2, \quad \mathcal{H} = 2\mathcal{C}_2 + 17\log\frac{T\mathrm{e}}{\delta}. \qquad (46)$$

In what follows, we will present several high-probability lemmas for the probabilistic analysis.

## E.1   PRELIMINARIES

The following lemma bound the noise norm under Assumption 3.

**Lemma E.1.** *Given $T \geq 1$, suppose that for any $t \in [T]$, $\boldsymbol{v}_t = \nabla f_{\mathbf{z}}\left(\boldsymbol{x}_t; \mathbf{z}_t\right) - \nabla f\left(\boldsymbol{x}_t\right)$ satisfies Assumption 3. Then, for any given $\delta \in (0, 1)$, it holds that with probability at least $1 - \delta$,*

$$\left\|\boldsymbol{v}_t\right\|^2 \leq \left(A\left(f(\boldsymbol{x}_t) - f^*\right) + B\left\|\nabla f\left(\boldsymbol{x}_t\right)\right\|^2 + C\right)\log\frac{T\mathrm{e}}{\delta}, \quad \forall t \in [T]. \qquad (47)$$

*Proof.* Denote $\zeta_t = \frac{\left\|\boldsymbol{v}_t\right\|^2}{A(f(\boldsymbol{x}_t) - f^*) + B\left\|\nabla f(\boldsymbol{x}_t)\right\|^2 + C}$, $\forall t \in [T]$, where $T$ is fixed. By the definition of the noise model, we have

$$\mathbb{E}_t\left[\exp\left(\zeta_t\right)\right] \leq \mathrm{e}, \quad \text{thus,} \quad \mathbb{E}\left[\exp\left(\zeta_t\right)\right] \leq \mathrm{e}.$$

By Markov's inequality, for any $\beta \in \mathbb{R}$,

$$\mathbb{P}\left(\max_{t\in[T]}\zeta_t \geq \beta\right) = \mathbb{P}\left(\exp\left(\max_{t\in[T]}\zeta_t\right) \geq \mathrm{e}^\beta\right)$$

$$\leq \mathrm{e}^{-\beta}\mathbb{E}\left[\exp\left(\max_{t\in[T]}\zeta_t\right)\right] \leq \mathrm{e}^{-\beta}\mathbb{E}\left[\sum_{t=1}^T \exp\left(\zeta_t\right)\right] \leq \mathrm{e}^{-\beta}T\mathrm{e}.$$

Therefore, with probability at least $1 - \delta$, we have

$$\left\|\boldsymbol{v}_t\right\|^2 \leq \left(A\left(f(\boldsymbol{x}_t) - f^*\right) + B\left\|\nabla f\left(\boldsymbol{x}_t\right)\right\|^2 + C\right)\log\frac{T\mathrm{e}}{\delta}, \quad \forall t \in [T].$$

$\square$

Next, we will establish a probabilistic upper bound for summation of the two martingale difference sequences based on the noise assumption and Lemma C.4.

**Lemma E.2.** *Given $T \geq 1$ and $\delta \in (0, 1)$, if Assumptions 1, 2 and 3 hold, then with probability at least $1 - \delta$, for all $l \in [T]$, we have*

$$\sum_{k=1}^{l} -A_k \left\langle \boldsymbol{\xi}_k, \nabla f(\boldsymbol{x}_k^{md}) \right\rangle \leq \frac{1}{4A_T \mathcal{M}^2} \sum_{k=1}^{l} A_k^2 \left\| \nabla f(\boldsymbol{x}_k^{md}) \right\|^2 \mathcal{M}_k^2 + 3A_T \mathcal{M}^2 \log \frac{T}{\delta}, \qquad (48)$$

*where*

$$\mathcal{M}_t = \sqrt{A \triangle_t^{md} + Bg\left(\triangle_t^{md}\right) + C}, \qquad (49)$$

*$\mathcal{M}$ is defined in (43) and $g\left(\cdot\right)$ is a function defined in (24).*

*Proof.* Let $X_k = -A_k \left\langle \boldsymbol{\xi}_k, \nabla f(\boldsymbol{x}_k^{md}) \right\rangle$. Note that $\boldsymbol{x}_k^{md}$ and $\boldsymbol{x}_{k-1}$ are random variables dependent on $\mathbf{z}_1, \cdots, \mathbf{z}_{k-1}$ and $\boldsymbol{\xi}_k$ is dependent on $\mathbf{z}_1, \cdots, \mathbf{z}_k$. It is apparent that $X_k$ is the martingale difference sequence since

$$\mathbb{E}_k \left[X_k\right] = -A_k \left\langle \mathbb{E}_k \left[\boldsymbol{\xi}_k\right], \nabla f(\boldsymbol{x}_k^{md}) \right\rangle = 0.$$

Also, by Assumption 3 and applying Cauchy-Schwarz inequality, we have

$$\mathbb{E}_k \left[ \exp \left( \frac{X_k^2}{A_k^2 \left\| \nabla f(\boldsymbol{x}_k^{md}) \right\|^2 \left( A\triangle_k^{md} + B \left\| \nabla f\left(\boldsymbol{x}_k^{md}\right) \right\|^2 + C \right)} \right) \right]$$

$$\leq \mathbb{E}_k \left[ \exp \left( \frac{A_k^2 \left\| \boldsymbol{\xi}_k \right\|^2 \left\| \nabla f(\boldsymbol{x}_k^{md}) \right\|^2}{A_k^2 \left\| \nabla f(\boldsymbol{x}_k^{md}) \right\|^2 \left( A\triangle_k^{md} + B \left\| \nabla f\left(\boldsymbol{x}_k^{md}\right) \right\|^2 + C \right)} \right) \right] \leq \exp(1) \qquad (50)$$

Thus, given any $l \in [T]$, applying Lemma C.4, we have that for any $\lambda > 0$, with probability at least $1 - \delta$,

$$\sum_{k=1}^{l} X_k \leq \frac{3\lambda}{4} \sum_{k=1}^{l} A_k^2 \left\| \nabla f(\boldsymbol{x}_k^{md}) \right\|^2 \left( A\triangle_k^{md} + B \left\| \nabla f\left(\boldsymbol{x}_k^{md}\right) \right\|^2 + C \right) + \frac{1}{\lambda} \log \frac{1}{\delta}$$

$$\leq \frac{3\lambda}{4} \sum_{k=1}^{l} A_k^2 \left\| \nabla f(\boldsymbol{x}_k^{md}) \right\|^2 \mathcal{M}_k^2 + \frac{1}{\lambda} \log \frac{1}{\delta}, \qquad (51)$$

where the second inequality follows from Lemma C.3. For any fixed $\lambda$, we can rescale over $\delta$ and have that with probability at least $1 - \delta$, for all $l \in [T]$,

$$\sum_{k=1}^{l} X_k \leq \frac{3\lambda}{4} \sum_{k=1}^{l} A_k^2 \left\| \nabla f(\boldsymbol{x}_k^{md}) \right\|^2 \mathcal{M}_k^2 + \frac{1}{\lambda} \log \frac{T}{\delta}.$$

Let $\lambda = \frac{1}{3A_T \mathcal{M}^2}$, and we obtain the desired result. $\qquad \square$

**Lemma E.3.** *Given $T \geq 1$ and $\delta \in (0, 1)$, if Assumptions 1, 2 and 3 hold. Then, with probability at least $1 - \delta$, for all $l \in [T]$, we have*

$$\sum_{k=1}^{l} (A_k - A_{k-1}) \left\langle \boldsymbol{\xi}_k, \boldsymbol{x}^* - \boldsymbol{x}_{k-1} \right\rangle \leq \frac{3 \log \frac{T}{\delta}}{2\mathcal{P}(\mathcal{F}_2)} \sum_{k=1}^{l} A_k \left\| \boldsymbol{x}^* - \boldsymbol{x}_{k-1} \right\|^2 \mathcal{M}_k^2 + \frac{\mathcal{P}(\mathcal{F}_2)}{2}, \qquad (52)$$

*where $\mathcal{M}_k$ is defined in (49) and $\mathcal{P}(\mathcal{F}_2)$ is defined in (62).*

*Proof.* Let $Y_k = (A_k - A_{k-1}) \left\langle \boldsymbol{\xi}_k, \boldsymbol{x}^* - \boldsymbol{x}_{k-1} \right\rangle$. Note that $\boldsymbol{x}_k^{md}$ and $\boldsymbol{x}_{k-1}$ are random variables dependent on $\mathbf{z}_1, \cdots, \mathbf{z}_{k-1}$ and $\boldsymbol{\xi}_k$ is dependent on $\mathbf{z}_1, \cdots, \mathbf{z}_k$. It is apparent that $Y_k$ is the martingale difference sequence since

$$\mathbb{E}_k \left[Y_k\right] = (A_k - A_{k-1}) \left\langle \mathbb{E}_k \left[\boldsymbol{\xi}_k\right], \boldsymbol{x}^* - \boldsymbol{x}_{k-1} \right\rangle = 0.$$

Also, by Assumption 3 and applying Cauchy-Schwarz inequality, we have

$$
\mathbb{E}_k \left[ \exp \left( \frac{Y_k^2}{A_k \left\| \boldsymbol{x}^* - \boldsymbol{x}_{k-1} \right\|^2 \left( A \triangle_k^{md} + B \left\| \nabla f \left( \boldsymbol{x}_k^{md} \right) \right\|^2 + C \right)} \right) \right]
$$

$$
\leq \mathbb{E}_k \left[ \exp \left( \frac{(A_k - A_{k-1})^2 \left\| \boldsymbol{\xi}_k \right\|^2 \left\| \boldsymbol{x}^* - \boldsymbol{x}_{k-1} \right\|^2}{A_k \left\| \boldsymbol{x}^* - \boldsymbol{x}_{k-1} \right\|^2 \left( A \triangle_k^{md} + B \left\| \nabla f \left( \boldsymbol{x}_k^{md} \right) \right\|^2 + C \right)} \right) \right] \leq \exp(1), \qquad (53)
$$

where the last inequality follows from Lemma 4.1. Thus, given any $l \in [T]$, applying Lemma C.4, we have that for any $\lambda > 0$, with probability at least $1 - \delta$,

$$
\sum_{k=1}^l Y_k \leq \frac{3\lambda}{4} \sum_{k=1}^l A_k \left\| \boldsymbol{x}^* - \boldsymbol{x}_{k-1} \right\|^2 \left( A \triangle_k^{md} + B \left\| \nabla f \left( \boldsymbol{x}_k^{md} \right) \right\|^2 + C \right) + \frac{1}{\lambda} \log \frac{1}{\delta}
$$

$$
\leq \frac{3\lambda}{4} \sum_{k=1}^l A_k \left\| \boldsymbol{x}^* - \boldsymbol{x}_{k-1} \right\|^2 \mathcal{M}_k^2 + \frac{1}{\lambda} \log \frac{1}{\delta}, \qquad (54)
$$

where the second inequality follows from Lemma C.3 and the definition of $\mathcal{M}_k$ in (49). For any fixed $\lambda$, we can rescale over $\delta$ and have that with probability at least $1 - \delta$, for all $l \in [T]$,

$$
\sum_{k=1}^l Y_k \leq \frac{3\lambda}{4} \sum_{k=1}^l A_k \left\| \boldsymbol{x}^* - \boldsymbol{x}_{k-1} \right\|^2 \mathcal{M}_k^2 + \frac{1}{\lambda} \log \frac{T}{\delta}.
$$

Let $\lambda = \frac{2 \log \frac{T}{\delta}}{\mathcal{P}(\mathcal{F}_2)}$, and we obtain the desired result. $\qquad \square$

We provide the following lemma for Algorithm 2, which is similar to Lemma D.1 in the deterministic case.

**Lemma E.4.** *Let $\left\{ \boldsymbol{x}_k^{md} \right\}_{k \in [T]}$ and $\left\{ \boldsymbol{x}_k^{ag} \right\}_{k \in [T]}$ be the two sequences generated by Algorithm 2. Then we have that for all $k \in [T]$,*

$$
\left\| \boldsymbol{x}_k^{md} - \boldsymbol{x}_{k-1}^{ag} \right\|^2 \leq \frac{1}{A_k \cdot A_{k-1}} \sum_{i=1}^{k-1} \frac{A_i^2 \cdot (\lambda_i - \beta)^2}{A_i - A_{i-1}} \left\| \boldsymbol{g}_i \right\|^2.
$$

*Proof.* Lemma E.4 can be seen as a corollary of Lemma D.1. As long as we replace the accurate gradient $\nabla f(\boldsymbol{x}_k^{md})$ in Lemma D.1 with the stochastic gradient $\boldsymbol{g}_t$, the proof is finished. $\qquad \square$

### E.2 CONVERGENCE ANALYSIS

In the next two lemmas, we assume that $\triangle_l^{md}$ is bounded in the first $t$ iterations and derive the iteration sequence based on the above analysis, in preparation for the induction argument in Lemma E.7.

**Lemma E.5.** *Suppose that $f \left( \boldsymbol{x}_l^{md} \right) - f^* \leq \mathcal{F}_2, \forall l \in [t]$. Then, under (47), for all $l \in [t]$, the conditions of Theorem 2, we have that for all $l \in [t]$, given $\delta \in (0, 1)$, with probability at least $1 - \delta$*

$$
A_l \triangle_l^{ag} + \frac{2}{\beta} \left\| \boldsymbol{x}_l - \boldsymbol{x}^* \right\|^2 \leq A_{l-1} \triangle_{l-1}^{ag} + \frac{2}{\beta} \left\| \boldsymbol{x}_{l-1} - \boldsymbol{x}^* \right\|^2 - \frac{1}{2} \beta A_l \left\| \nabla f(\boldsymbol{x}_l^{md}) \right\|^2 + \frac{1}{2} \beta A_l \left\| \boldsymbol{\xi}_l \right\|^2
$$

$$
+ \left\langle \boldsymbol{\xi}_l, -\beta A_l \nabla f(\boldsymbol{x}_l^{md}) + (A_l - A_{l-1}) (\boldsymbol{x}^* - \boldsymbol{x}_{l-1}) \right\rangle. \qquad (55)
$$

*Proof.* Suppose that (47) in Lemma E.1 always happen, then we deduce (55) always holds. Since (47) holds with probability at least $1 - \delta$, it follows that (55) happens with probability at least $1 - \delta$. With the assumption that $\triangle_l^{md} \leq \mathcal{F}_2, \forall l \in [t]$ and applying Corollary 1, we have $\left\| \nabla f(\boldsymbol{x}_l^{md}) \right\| \leq \sqrt{g(\mathcal{F}_2)}, \forall l \in [t]$. Therefore,

$$
\left\| \boldsymbol{x}_l^{ag} - \boldsymbol{x}_l^{md} \right\| = \beta \left\| \nabla f(\boldsymbol{x}_l^{md}) + \boldsymbol{\xi}_l \right\| \leq \beta \left( \left\| \nabla f(\boldsymbol{x}_l^{md}) \right\| + \left\| \boldsymbol{\xi}_l \right\| \right)
$$

$$
\leq \beta \left( \sqrt{g(\mathcal{F}_2)} + \mathcal{M} \sqrt{\log \frac{T\mathrm{e}}{\delta}} \right) \leq \min \left\{ 1/L_1, 1/L_2 \right\},
$$

where the first inequality follows from the triangle inequality and the second inequality holds since (47). The last inequality holds since $\beta \le 1/\mathcal{G}_{1,1}$ and $\beta \le 1/\mathcal{G}_{1,2}$ with $\mathcal{G}_{1,1}, \mathcal{G}_{1,2}$ defined in (44). By Lemma C.2, we have

$$
\begin{aligned}
f\left(\boldsymbol{x}_l^{ag}\right) \le & f\left(\boldsymbol{x}_l^{md}\right) + \left\langle \nabla f(\boldsymbol{x}_l^{md}), \boldsymbol{x}_l^{ag} - \boldsymbol{x}_l^{md} \right\rangle + \frac{L_0 + L_1 \left(g(\mathcal{F}_2)\right)^{\frac{p}{2}} + L_2 \mathcal{F}_2^q}{2} \left\| \boldsymbol{x}_l^{ag} - \boldsymbol{x}_l^{md} \right\|^2 \\
= & f\left(\boldsymbol{x}_l^{md}\right) - \beta \left\| \nabla f(\boldsymbol{x}_l^{md}) \right\|^2 - \beta \left\langle \nabla f(\boldsymbol{x}_l^{md}), \boldsymbol{\xi}_l \right\rangle \\
& + \frac{L_0 + L_1 \left(g(\mathcal{F}_2)\right)^{\frac{p}{2}} + L_2 \mathcal{F}_2^q}{2} \beta^2 \left\| \nabla f(\boldsymbol{x}_l^{md}) + \boldsymbol{\xi}_l \right\|^2 .
\end{aligned}
\tag{56}
$$

Note that (31) is derived from the convexity of $f$ and the iteration step

$$
\boldsymbol{x}_t^{md} = \frac{A_{t-1}}{A_t} \boldsymbol{x}_{t-1}^{ag} + \left(1 - \frac{A_{t-1}}{A_t}\right) \boldsymbol{x}_{t-1},
$$

which is the same in Algorithm 1 and Algorithm 2. Thus, (31) holds here. Combining (31) and (56),

$$
\begin{aligned}
& f\left(\boldsymbol{x}_l^{ag}\right) \\
\le & \frac{A_{l-1}}{A_l} f\left(\boldsymbol{x}_{l-1}^{ag}\right) + \left(1 - \frac{A_{l-1}}{A_l}\right) f^* + \left(1 - \frac{A_{l-1}}{A_l}\right) \left\langle \nabla f(\boldsymbol{x}_l^{md}), \boldsymbol{x}_{l-1} - \boldsymbol{x}^* \right\rangle \\
& - \beta \left\| \nabla f(\boldsymbol{x}_l^{md}) \right\|^2 - \beta \left\langle \nabla f(\boldsymbol{x}_l^{md}), \boldsymbol{\xi}_l \right\rangle + \frac{L_0 + L_1 \left(g(\mathcal{F}_2)\right)^{\frac{p}{2}} + L_2 \mathcal{F}_2^q}{2} \beta^2 \left\| \nabla f(\boldsymbol{x}_l^{md}) + \boldsymbol{\xi}_l \right\|^2 .
\end{aligned}
\tag{57}
$$

Also, by the iteration step, we have

$$
\begin{aligned}
& \left\| \boldsymbol{x}_{l-1} - \boldsymbol{x}^* \right\|^2 - 2\lambda_l \left\langle \nabla f(\boldsymbol{x}_l^{md}) + \boldsymbol{\xi}_l, \boldsymbol{x}_{l-1} - \boldsymbol{x}^* \right\rangle + \lambda_l^2 \left\| \nabla f(\boldsymbol{x}_l^{md}) + \boldsymbol{\xi}_l \right\|^2 \\
= & \left\| \boldsymbol{x}_{l-1} - \lambda_l \left( \nabla f(\boldsymbol{x}_l^{md}) + \boldsymbol{\xi}_l \right) - \boldsymbol{x}^* \right\|^2 = \left\| \boldsymbol{x}_l - \boldsymbol{x}^* \right\|^2 .
\end{aligned}
$$

Hence,

$$
\left\langle \nabla f(\boldsymbol{x}_l^{md}) + \boldsymbol{\xi}_k, \boldsymbol{x}_{l-1} - \boldsymbol{x}^* \right\rangle = \frac{1}{2\lambda_l} \left[ \left\| \boldsymbol{x}_{l-1} - \boldsymbol{x}^* \right\|^2 - \left\| \boldsymbol{x}_l - \boldsymbol{x}^* \right\|^2 \right] + \frac{\lambda_l}{2} \left\| \nabla f(\boldsymbol{x}_l^{md}) + \boldsymbol{\xi}_l \right\|^2 .
\tag{58}
$$

Combining with the fact that

$$
\left\| \nabla f(\boldsymbol{x}_l^{md}) + \boldsymbol{\xi}_l \right\|^2 = \left\| \nabla f(\boldsymbol{x}_l^{md}) \right\|^2 + 2 \left\langle \boldsymbol{\xi}_l, \nabla f(\boldsymbol{x}_l^{md}) \right\rangle + \left\| \boldsymbol{\xi}_l \right\|^2 \le 2 \left\| \nabla f(\boldsymbol{x}_l^{md}) \right\|^2 + 2 \left\| \boldsymbol{\xi}_l \right\|^2 ,
\tag{59}
$$

we have

$$
\begin{aligned}
f\left(\boldsymbol{x}_l^{ag}\right) \le & \frac{A_{l-1}}{A_l} f\left(\boldsymbol{x}_{l-1}^{ag}\right) + \left(1 - \frac{A_{l-1}}{A_l}\right) f^* + \frac{A_l - A_{l-1}}{2A_l \cdot \lambda_l} \left[ \left\| \boldsymbol{x}_{l-1} - \boldsymbol{x}^* \right\|^2 - \left\| \boldsymbol{x}_l - \boldsymbol{x}^* \right\|^2 \right] \\
& - \beta \left( 1 - \left( L_0 + L_1 \left(g(\mathcal{F}_2)\right)^{\frac{p}{2}} + L_2 \mathcal{F}_2^q \right) \beta - \frac{\lambda_l \left( A_l - A_{l-1} \right)}{\beta A_l} \right) \left\| \nabla f(\boldsymbol{x}_l^{md}) \right\|^2 \\
& + \left( \left( L_0 + L_1 \left(g(\mathcal{F}_2)\right)^{\frac{p}{2}} + L_2 \mathcal{F}_2^q \right) \beta^2 + \frac{\lambda_l \left( A_l - A_{l-1} \right)}{A_l} \right) \left\| \boldsymbol{\xi}_l \right\|^2 \\
& + \left\langle \boldsymbol{\xi}_l, -\beta \nabla f(\boldsymbol{x}_l^{md}) + \frac{A_l - A_{l-1}}{A_l} \left( \boldsymbol{x}^* - \boldsymbol{x}_{l-1} \right) \right\rangle .
\end{aligned}
\tag{60}
$$

Since the setting of $\lambda_l$ in (9), we have

$$
\frac{A_l - A_{l-1}}{2A_l \cdot \lambda_l} = \frac{2}{A_l \cdot \beta},
$$

and

$$
\frac{A_l - A_{l-1}}{A_l} \lambda_l = \frac{A_l - A_{l-1}}{4A_l} \cdot \beta \left( A_l - A_{l-1} \right) \le \frac{\beta}{4},
$$

where the inequality follows from Lemma 4.1. Combining with the constraint that $\beta \leq 1/\mathcal{G}_{1,3}$ Thus, we have

$$
f\left(\boldsymbol{x}_l^{ag}\right) \leq \frac{A_{l-1}}{A_l} f\left(\boldsymbol{x}_{l-1}^{ag}\right) + \left(1 - \frac{A_{l-1}}{A_l}\right) f^* + \frac{2}{A_l \cdot \beta} \left[\|\boldsymbol{x}_{l-1} - \boldsymbol{x}^*\|^2 - \|\boldsymbol{x}_l - \boldsymbol{x}^*\|^2\right]
$$

$$
- \frac{1}{2}\beta \left\|\nabla f(\boldsymbol{x}_l^{md})\right\|^2 + \frac{1}{2}\beta \|\boldsymbol{\xi}_l\|^2 + \left\langle \boldsymbol{\xi}_l, -\beta\nabla f(\boldsymbol{x}_l^{md}) + \frac{A_l - A_{l-1}}{A_l}\left(\boldsymbol{x}^* - \boldsymbol{x}_{l-1}\right)\right\rangle.
$$

Multiplying $A_l$ on both sides and re-arranging the inequality, we obtain the desired result. $\qquad \square$

**Lemma E.6.** *Under the condition of Lemma E.5, let* (47), (48) *and* (52). *Then for any $\delta \in (0, 1/3)$, it holds that with probability at least $1 - 3\delta$,*

$$
A_l \triangle_l^{ag} + \frac{2}{\beta}\|\boldsymbol{x}_l - \boldsymbol{x}^*\|^2 + \frac{1}{4}\beta \sum_{i=1}^l A_i \left\|\nabla f(\boldsymbol{x}_i^{md})\right\|^2 \leq \mathcal{P}(\mathcal{F}_2), \forall 0 \leq l \leq t, \tag{61}
$$

*where*

$$
\mathcal{P}(\mathcal{F}_2) = \frac{2\mathcal{C}_2}{\beta} + \frac{17}{2}TA_T\beta\mathcal{M}^2 \log \frac{Te}{\delta}, \tag{62}
$$

*and $\mathcal{C}_2$ is defined in* (46).

*Proof.* It is apparent that

$$
A_0 \triangle_0^{ag} + \frac{2}{\beta}\|\boldsymbol{x}_0 - \boldsymbol{x}^*\|^2 \leq \mathcal{P}(\mathcal{F}_2).
$$

Suppose that for some $k \in [t-1]$,

$$
A_l \triangle_l^{ag} + \frac{2}{\beta}\|\boldsymbol{x}_l - \boldsymbol{x}^*\|^2 + \frac{1}{4}\beta \sum_{i=1}^l A_i \left\|\nabla f(\boldsymbol{x}_i^{md})\right\|^2 \leq \mathcal{P}(\mathcal{F}_2), \forall 0 \leq l \leq k. \tag{63}
$$

In what follows, we will bound

$$
A_{k+1} \triangle_{k+1}^{ag} + \frac{2}{\beta}\|\boldsymbol{x}_{k+1} - \boldsymbol{x}^*\|^2 + \frac{1}{4}\beta \sum_{i=1}^{k+1} A_i \left\|\nabla f(\boldsymbol{x}_i^{md})\right\|^2.
$$

Note that $f\left(\boldsymbol{x}_l^{md}\right) - f^* \leq \mathcal{F}_2, \forall l \in [t]$, according to Lemma E.5, (55) and $\mathcal{M}_l \leq \mathcal{M}$ hold here for all $l \in [k+1]$. Thus, summing up (55) over $l \in [k+1]$, we have

$$
A_{k+1} \triangle_{k+1}^{ag} + \frac{2}{\beta}\|\boldsymbol{x}_{k+1} - \boldsymbol{x}^*\|^2 \leq A_0 \triangle_0^{ag} + \frac{2}{\beta}\|\boldsymbol{x}_0 - \boldsymbol{x}^*\|^2 - \frac{1}{2}\beta \sum_{i=1}^{k+1} A_i \left\|\nabla f(\boldsymbol{x}_i^{md})\right\|^2
$$

$$
+ \frac{1}{2}\beta \sum_{i=1}^{k+1} A_i \|\boldsymbol{\xi}_i\|^2 - \beta \sum_{i=1}^{k+1} A_i \left\langle \boldsymbol{\xi}_i, \nabla f(\boldsymbol{x}_i^{md})\right\rangle + \sum_{i=1}^{k+1} (A_i - A_{i-1})\left\langle \boldsymbol{\xi}_i, \boldsymbol{x}^* - \boldsymbol{x}_{i-1}\right\rangle. \tag{64}
$$

Applying (48) and letting $l = k+1$, we have

$$
-\beta \sum_{i=1}^{k+1} A_i \left\langle \boldsymbol{\xi}_i, \nabla f(\boldsymbol{x}_i^{md})\right\rangle \leq \frac{1}{4A_T\mathcal{M}^2}\beta \sum_{i=1}^{k+1} A_i^2 \left\|\nabla f(\boldsymbol{x}_i^{md})\right\|^2 \mathcal{M}_i^2 + 3A_T\beta\mathcal{M}^2 \log \frac{T}{\delta}
$$

$$
\leq \frac{1}{4}\beta \sum_{i=1}^{k+1} A_i \left\|\nabla f(\boldsymbol{x}_i^{md})\right\|^2 + 3A_T\beta\mathcal{M}^2 \log \frac{T}{\delta}, \tag{65}
$$

where the second inequality follows from $\mathcal{M}_i \leq \mathcal{M}$ and $A_i \leq A_T$ for all $i \in [k+1]$. Similarly, applying (52), we obtain that

$$
\sum_{i=1}^{k+1} (A_i - A_{i-1})\left\langle \boldsymbol{\xi}_i, \boldsymbol{x}^* - \boldsymbol{x}_{i-1}\right\rangle \leq \frac{3\log \frac{T}{\delta}}{2\mathcal{P}(\mathcal{F}_2)} \sum_{i=1}^{k+1} A_i \|\boldsymbol{x}^* - \boldsymbol{x}_{i-1}\|^2 \mathcal{M}_i^2 + \frac{\mathcal{P}(\mathcal{F}_2)}{2}
$$

$$
\leq \frac{3}{4}\beta \log \frac{T}{\delta} \sum_{i=1}^{k+1} A_i\mathcal{M}_i^2 + \frac{\mathcal{P}(\mathcal{F}_2)}{2}
$$

$$
\leq \frac{3}{4}\beta T \cdot A_T\mathcal{M}^2 \log \frac{T}{\delta} + \frac{\mathcal{P}(\mathcal{F}_2)}{2}, \tag{66}
$$

where the second inequality holds since

$$\|\boldsymbol{x}_i - \boldsymbol{x}^*\|^2 \le \frac{1}{2}\beta \cdot \mathcal{P}(\mathcal{F}_2), \quad \forall 0 \le i \le k,$$

derived from (63), and the last inequality follows from $\mathcal{M}_i \le \mathcal{M}$ and $A_i \le A_T$ for all $i \in [k+1]$. Combining (64), (65) and (66), we have

$$A_{k+1}\triangle_{k+1}^{ag} + \frac{2}{\beta}\|\boldsymbol{x}_{k+1} - \boldsymbol{x}^*\|^2$$

$$\le A_0\triangle_0^{ag} + \frac{2}{\beta}\|\boldsymbol{x}_0 - \boldsymbol{x}^*\|^2 - \frac{1}{2}\beta\sum_{i=1}^{k+1} A_i \left\|\nabla f(\boldsymbol{x}_i^{md})\right\|^2 + \frac{1}{2}\beta\sum_{i=1}^{k+1} A_i \|\boldsymbol{\xi}_i\|^2$$

$$+ \frac{1}{4}\beta\sum_{i=1}^{k+1} A_i \left\|\nabla f(\boldsymbol{x}_i^{md})\right\|^2 + 3A_T\beta\mathcal{M}^2 \log\frac{T}{\delta} + \frac{3}{4}\beta T \cdot A_T\mathcal{M}^2 \log\frac{T}{\delta} + \frac{\mathcal{P}(\mathcal{F}_2)}{2}.$$

Applying (48) with the assumption that $\triangle_l^{md} \le \mathcal{F}_2, \forall l \in [t]$,

$$\|\boldsymbol{\xi}_l\|^2 \le \mathcal{M}^2 \log\frac{T\mathrm{e}}{\delta}.$$

Combining the above inequalities, we obtain that

$$A_{k+1}\triangle_{k+1}^{ag} + \frac{2}{\beta}\|\boldsymbol{x}_{k+1} - \boldsymbol{x}^*\|^2$$

$$\le A_0\triangle_0^{ag} + \frac{2}{\beta}\|\boldsymbol{x}_0 - \boldsymbol{x}^*\|^2 - \frac{1}{4}\beta\sum_{i=1}^{k+1} A_i \left\|\nabla f(\boldsymbol{x}_i^{md})\right\|^2$$

$$+ \left(\frac{1}{2}\log\frac{T\mathrm{e}}{\delta} + \frac{3}{4}\log\frac{T}{\delta}\right)T \cdot A_T\beta\mathcal{M}^2 + 3A_T\beta\mathcal{M}^2 \log\frac{T}{\delta} + \frac{\mathcal{P}(\mathcal{F}_2)}{2}$$

$$\le A_0\triangle_0^{ag} + \frac{2}{\beta}\|\boldsymbol{x}_0 - \boldsymbol{x}^*\|^2 - \frac{1}{4}\beta\sum_{i=1}^{k+1} A_i \left\|\nabla f(\boldsymbol{x}_i^{md})\right\|^2 + \frac{17}{4}TA_T\beta\mathcal{M}^2 \log\frac{T\mathrm{e}}{\delta} + \frac{\mathcal{P}(\mathcal{F}_2)}{2}.$$

Hence, we could deduce that

$$A_{k+1}\triangle_{k+1}^{ag} + \frac{2}{\beta}\|\boldsymbol{x}_{k+1} - \boldsymbol{x}^*\|^2 + \frac{1}{4}\beta\sum_{i=1}^{k+1} A_i \left\|\nabla f(\boldsymbol{x}_i^{md})\right\|^2 \le \mathcal{P}(\mathcal{F}_2), \tag{67}$$

since

$$\mathcal{P}(\mathcal{F}_2) = \frac{2}{\beta}\left(\triangle_0^{ag} + 2\|\boldsymbol{x}_0 - \boldsymbol{x}^*\|^2\right) + \frac{17}{2}TA_T\beta\mathcal{M}^2 \log\frac{T\mathrm{e}}{\delta}.$$

$\square$

Based on previous lemmas, we will provide the upper bound of $\triangle_t^{md}$ for all $t \in [T]$.

**Lemma E.7.** *Under the condition of Theorem 2, let (47), (48) and (52). Then~~for any given $\delta \in (0, 1/3)$ we have that with probability at least $1 - 3\delta$,~~*

$$f\left(\boldsymbol{x}_t^{md}\right) - f^* \le \mathcal{F}_2, \forall t \in [T], \tag{68}$$

*where $\mathcal{F}_2$ is defined in (45).*

*Proof.* It is apparent that $f\left(\boldsymbol{x}_1^{md}\right) - f^* = f\left(\boldsymbol{x}_0^{ag}\right) - f^* \le \mathcal{F}_2$. Suppose that for some $t \in [T]$,

$$f\left(\boldsymbol{x}_l^{md}\right) - f^* \le \mathcal{F}_2, \forall l \in [t].$$

Then, by Lemma E.6, (61) holds. Next, we will bound $f\left(\boldsymbol{x}_{t+1}^{md}\right) - f^*$. By Lemma E.4, we have

$$\left\|\boldsymbol{x}_{t+1}^{md} - \boldsymbol{x}_t^{ag}\right\|^2 \le \frac{1}{A_{t+1} \cdot A_t}\sum_{i=1}^t \frac{A_i^2 \cdot (\lambda_i - \beta)^2}{A_i - A_{i-1}}\left\|\nabla f(\boldsymbol{x}_i^{md}) + \boldsymbol{\xi}_i\right\|^2$$

$$\le 2\sum_{i=1}^t \frac{\lambda_i^2 + \beta^2}{A_i - A_{i-1}}\left\|\nabla f(\boldsymbol{x}_i^{md}) + \boldsymbol{\xi}_i\right\|^2, \tag{69}$$

where the second inequality holds since $(a - b)^2 \le 2a^2 + 2b^2$ and $A_i \le A_t \le A_{t+1}, \forall i \in [t]$. Also, since $\lambda_t = \frac{1}{4}\beta (A_t - A_{t-1})$ for all $t \in [T]$, we have

$$\left\| \boldsymbol{x}_{t+1}^{md} - \boldsymbol{x}_t^{ag} \right\|^2 \le 2\beta^2 \sum_{i=1}^{t} \frac{\frac{1}{16}(A_i - A_{i-1})^2 + 1}{A_i - A_{i-1}} \left\| \nabla f(\boldsymbol{x}_i^{md}) + \boldsymbol{\xi}_i \right\|^2$$

$$\le \frac{17}{8}\beta^2 \sum_{i=1}^{t} (A_i - A_{i-1}) \left\| \nabla f(\boldsymbol{x}_i^{md}) + \boldsymbol{\xi}_i \right\|^2$$

$$\le \frac{17}{4}\beta^2 \sum_{i=1}^{t} (A_i - A_{i-1}) \left( \left\| \nabla f(\boldsymbol{x}_i^{md}) \right\|^2 + \left\| \boldsymbol{\xi}_i \right\|^2 \right),$$

where the second inequality follows from Lemma 4.1 and the last inequality holds since $\left\| \boldsymbol{a} + \boldsymbol{b} \right\|^2 \le 2 \left\| \boldsymbol{a} \right\|^2 + 2 \left\| \boldsymbol{b} \right\|^2$. Applying Lemma 4.1 and using the fact that $\sqrt{\beta A_i} = \sqrt{\beta B_i + 1} \ge 1, \forall i \in [T]$, we have

$$\left\| \boldsymbol{x}_{t+1}^{md} - \boldsymbol{x}_t^{ag} \right\|^2 \le \frac{17}{4}\beta^2 \sum_{i=1}^{t} \sqrt{A_i} \left( \left\| \nabla f(\boldsymbol{x}_i^{md}) \right\|^2 + \left\| \boldsymbol{\xi}_i \right\|^2 \right)$$

$$\le \frac{17}{4}\beta^{\frac{5}{2}} \sum_{i=1}^{t} A_i \left( \left\| \nabla f(\boldsymbol{x}_i^{md}) \right\|^2 + \left\| \boldsymbol{\xi}_i \right\|^2 \right). \tag{70}$$

Since the assumption that $f(\boldsymbol{x}_l^{md}) - f^* \le \mathcal{F}_2, \forall l \in [t]$, by (61), we have

$$\beta \sum_{i=1}^{t} A_i \left\| \nabla f(\boldsymbol{x}_i^{md}) \right\|^2 \le 4\mathcal{P}(\mathcal{F}_2).$$

Combining with (70), (47) and recalling the expression of $\mathcal{P}(\mathcal{F}_2)$ in (62), we obtain that

$$\left\| \boldsymbol{x}_{t+1}^{md} - \boldsymbol{x}_t^{ag} \right\|^2 \le 17\beta^{\frac{3}{2}}\mathcal{P}(\mathcal{F}_2) + \frac{17}{4}\beta^{\frac{5}{2}}T A_T \mathcal{M}^2 \log\frac{Te}{\delta}$$

$$= 34\sqrt{\beta} \cdot \mathcal{C}_2 + \frac{289}{2}\beta^{\frac{5}{2}}T A_T \mathcal{M}^2 \log\frac{Te}{\delta} + \frac{17}{4}\beta^{\frac{5}{2}}T A_T \mathcal{M}^2 \log\frac{Te}{\delta}$$

$$= 34\sqrt{\beta} \cdot \mathcal{C}_2 + \frac{595}{4}\beta^{\frac{5}{2}}T A_T \mathcal{M}^2 \log\frac{Te}{\delta}.$$

Combining with Lemma 4.1 and the setting that $A_T = B_T + 1/\beta$, we have

$$\left\| \boldsymbol{x}_{t+1}^{md} - \boldsymbol{x}_t^{ag} \right\|^2 \le 34\sqrt{\beta} \cdot \mathcal{C}_2 + \frac{595}{4}\beta^{\frac{5}{2}}T^3 \mathcal{M}^2 \log\frac{Te}{\delta} + \frac{595}{4}\beta^{\frac{3}{2}}T \mathcal{M}^2 \log\frac{Te}{\delta}.$$

Since $\beta \le \min\left\{ 1/\mathcal{G}_{1,4}, 1/\left( \mathcal{G}_2 T^{\frac{6}{5}} \right), 1/\left( \mathcal{G}_3 T^{\frac{2}{3}} \right) \right\}$, where $\mathcal{G}_{1,4}, \mathcal{G}_2, \mathcal{G}_3$ are defined in (43), (44),

$$\left\| \boldsymbol{x}_{t+1}^{md} - \boldsymbol{x}_t^{ag} \right\|^2 \le \frac{1}{(L_1 + L_2)^2}.$$

Hence, applying Lemma C.2 and Cauchy-Schwarz inequality, we have

$$f\left(\boldsymbol{x}_{t+1}^{md}\right)$$

$$\le f\left(\boldsymbol{x}_t^{ag}\right) + \left\langle \nabla f(\boldsymbol{x}_t^{ag}), \boldsymbol{x}_{t+1}^{md} - \boldsymbol{x}_t^{ag} \right\rangle + \frac{L_0 + L_1 \left\| \nabla f(\boldsymbol{x}_t^{ag}) \right\|^p + L_2 (\triangle_t^{ag})^q}{2} \left\| \boldsymbol{x}_{t+1}^{md} - \boldsymbol{x}_t^{ag} \right\|^2$$

$$\le f\left(\boldsymbol{x}_t^{ag}\right) + \left\| \nabla f(\boldsymbol{x}_t^{ag}) \right\| \cdot \left\| \boldsymbol{x}_{t+1}^{md} - \boldsymbol{x}_t^{ag} \right\| + \frac{L_0 + L_1 \left\| \nabla f(\boldsymbol{x}_t^{ag}) \right\|^p + L_2 (\triangle_t^{ag})^q}{2} \left\| \boldsymbol{x}_{t+1}^{md} - \boldsymbol{x}_t^{ag} \right\|^2$$

$$\le f\left(\boldsymbol{x}_t^{ag}\right) + \frac{1}{L_1 + L_2} \left\| \nabla f(\boldsymbol{x}_t^{ag}) \right\| + \frac{L_0 + L_1 \left\| \nabla f(\boldsymbol{x}_t^{ag}) \right\|^p + L_2 (\triangle_t^{ag})^q}{2 (L_1 + L_2)^2}. \tag{71}$$

Since the assumption that $\triangle_l^{md} \le \mathcal{F}_2, \forall l \in [t]$, by Lemma E.6, we have

$$\triangle_t^{ag} \le \frac{\mathcal{P}(\mathcal{F}_2)}{A_t} \le \beta \cdot \mathcal{P}(\mathcal{F}_2), \tag{72}$$

where the second inequality holds since $A_t \geq 1/\beta$. Plugging (62) into (72), we obtain that

$$\triangle_t^{ag} \leq 2\mathcal{C}_2 + \frac{17}{2}T^3\beta^2\mathcal{M}^2 \log \frac{T\mathrm{e}}{\delta} + \frac{17}{2}T\beta\mathcal{M}^2 \log \frac{T\mathrm{e}}{\delta} \leq 2\mathcal{C}_2 + 17\log \frac{T\mathrm{e}}{\delta} = \mathcal{H},$$

where the last inequality follow from

$$\beta \leq \frac{1}{\mathcal{M}T^{\frac{3}{2}}}, \quad \text{and} \quad \beta \leq \frac{1}{\mathcal{M}^2 T}.$$

Note that $\mathcal{H}$ is independent on $\mathcal{F}_2$. By Corollary 1, we have $\|\nabla f(\boldsymbol{x}_t^{ag})\| \leq \sqrt{g(\mathcal{H})}$. Combining with (71) and subtracting $f^*$ from both sides, we obtain that

$$\triangle_{t+1}^{md} \leq \mathcal{H} + \frac{\sqrt{g(\mathcal{H})}}{L_1 + L_2} + \frac{L_0 + L_1 \left(g(\mathcal{H})\right)^{\frac{p}{2}} + L_2\mathcal{H}^q}{2\left(L_1 + L_2\right)^2} = \mathcal{F}_2.$$

Now we finish the induction and obtain the desired result. $\qquad\square$

With the above lemmas, we are ready to prove the final convergence result.

*Proof of Theorem 2.* In what follows, we assume (47), (48) and (52) always hold, and under these conditions we prove the desired error bounds. Using Lemmas E.1, E.2 and E.3, (47), (48) and (52) hold with probability at least $1 - 3\delta$. Thus, the desired error bounds also hold with probability at least $1 - 3\delta$.

By Lemma E.7, (68) holds. Based on Lemma E.6, we obtain that

$$\triangle_T^{ag} \leq \frac{\mathcal{P}(\mathcal{F}_2)}{A_T} \leq \frac{8\mathcal{C}_2}{T^2\beta} + \frac{17}{2}T\beta\mathcal{M}^2 \log \frac{T\mathrm{e}}{\delta}.$$

Since the constraints of $\beta$ in (9), we have

$$\triangle_T^{ag} \leq \frac{8\mathcal{C}_2}{T^2}\left(L_1 + L_2\right)\left(\sqrt{g\left(\mathcal{F}_2\right)} + \mathcal{M}\sqrt{\log \frac{T\mathrm{e}}{\delta}}\right)$$

$$+ \frac{32\mathcal{C}_2}{T^2}\left(L_0 + L_1\left(g\left(\mathcal{F}_2\right)\right)^{\frac{p}{2}} + L_2\mathcal{F}_2^q + 1156\left(L_1 + L_2\right)^4\mathcal{C}_2^2\right)$$

$$+ \frac{8\mathcal{C}_2}{T^{\frac{4}{5}}}(595)^{\frac{2}{5}}\left(L_1 + L_2\right)^{\frac{4}{5}}\mathcal{M}^{\frac{4}{5}}\left(\log \frac{T\mathrm{e}}{\delta}\right)^{\frac{2}{5}} + \frac{8\mathcal{C}_2}{T^{\frac{4}{3}}}(595)^{\frac{2}{3}}\left(L_1 + L_2\right)^{\frac{4}{3}}\mathcal{M}^{\frac{4}{3}}\left(\log \frac{T\mathrm{e}}{\delta}\right)^{\frac{2}{3}}$$

$$+ \frac{8\mathcal{C}_2\mathcal{M}^2}{T} + \frac{\mathcal{M}}{\sqrt{T}}\left(\frac{17}{2}\log \frac{T\mathrm{e}}{\delta} + 8\mathcal{C}_2\right). \tag{73}$$

$\qquad\square$

# F  PROOF OF THEOREM 3

We first provide the following lemma as a key to the induction argument in Lemma F.2.

**Lemma F.1.** *Under the conditions of Theorem 3, for all $t \in [T]$, it holds that*

$$\mathbb{E}\left[A_t\triangle_t^{ag}\right] + \frac{1 + B}{\beta}\mathbb{E}\left[\|\boldsymbol{x}_t - \boldsymbol{x}^*\|^2\right]$$

$$\leq \frac{\mathcal{C}_3}{\beta} - \frac{1}{2}\beta\sum_{l=1}^{t}A_l\mathbb{E}\left[\left\|\nabla f(\boldsymbol{x}_l^{md})\right\|^2\right] + \frac{1}{2\left(1 + B\right)}\beta\sum_{l=1}^{t}A_l\mathbb{E}\left[A\triangle_l^{md} + C\right],$$

*where*

$$\mathcal{C}_3 = \triangle_0^{ag} + (1 + B)\|\boldsymbol{x}_0 - \boldsymbol{x}^*\|^2. \tag{74}$$

*Proof.* By the descent lemma for Lipschitz smooth functions and the iteration step in Algorithm 2,

$$f\left(\boldsymbol{x}_l^{ag}\right) \leq f\left(\boldsymbol{x}_l^{md}\right) + \left\langle \nabla f(\boldsymbol{x}_l^{md}), \boldsymbol{x}_l^{ag} - \boldsymbol{x}_l^{md} \right\rangle + \frac{L}{2} \left\|\boldsymbol{x}_l^{ag} - \boldsymbol{x}_l^{md}\right\|^2$$

$$= f\left(\boldsymbol{x}_l^{md}\right) - \beta \left\|\nabla f(\boldsymbol{x}_l^{md})\right\|^2 - \beta \left\langle \nabla f(\boldsymbol{x}_l^{md}), \boldsymbol{\xi}_l \right\rangle + \frac{L}{2}\beta^2 \left\|\nabla f(\boldsymbol{x}_l^{md}) + \boldsymbol{\xi}_l\right\|^2.$$

Note that (31), (58) and (59) still holds here as they are independent of the smoothness condition. Thus,

$$f\left(\boldsymbol{x}_l^{ag}\right)$$

$$\leq \frac{A_{l-1}}{A_l} f\left(\boldsymbol{x}_{l-1}^{ag}\right) + \left(1 - \frac{A_{l-1}}{A_l}\right) f^* + \left(1 - \frac{A_{l-1}}{A_l}\right) \left\langle \nabla f(\boldsymbol{x}_l^{md}), \boldsymbol{x}_{l-1} - \boldsymbol{x}^* \right\rangle$$

$$- \beta \left\|\nabla f(\boldsymbol{x}_l^{md})\right\|^2 - \beta \left\langle \nabla f(\boldsymbol{x}_l^{md}), \boldsymbol{\xi}_l \right\rangle + \frac{L}{2}\beta^2 \left\|\nabla f(\boldsymbol{x}_l^{md}) + \boldsymbol{\xi}_l\right\|^2$$

$$\leq \frac{A_{l-1}}{A_l} f\left(\boldsymbol{x}_{l-1}^{ag}\right) + \left(1 - \frac{A_{l-1}}{A_l}\right) f^* + \frac{A_l - A_{l-1}}{2A_l \cdot \lambda_l} \left[\|\boldsymbol{x}_{l-1} - \boldsymbol{x}^*\|^2 - \|\boldsymbol{x}_l - \boldsymbol{x}^*\|^2\right]$$

$$- \beta \left(1 - \frac{L\beta}{2} - \frac{\lambda_l (A_l - A_{l-1})}{2\beta A_l}\right) \left\|\nabla f(\boldsymbol{x}_l^{md})\right\|^2 + \left(\frac{L\beta^2}{2} + \frac{\lambda_l (A_l - A_{l-1})}{2A_l}\right) \|\boldsymbol{\xi}_l\|^2$$

$$+ \left\langle \boldsymbol{\xi}_l, \left(-\beta + L\beta^2 + \frac{\lambda_l (A_l - A_{l-1})}{A_l}\right) \nabla f(\boldsymbol{x}_l^{md}) \right\rangle + \left\langle \boldsymbol{\xi}_l, \frac{A_l - A_{l-1}}{A_l} \left(\boldsymbol{x}^* - \boldsymbol{x}_{l-1}\right) \right\rangle. \quad (75)$$

By Assumption 4, we obtain that for all $l \in [T]$,

$$\mathbb{E}\left[\|\boldsymbol{\xi}_l\|^2\right] = \mathbb{E}\left[\mathbb{E}_l\left[\|\boldsymbol{\xi}_l\|^2\right]\right] \leq \mathbb{E}\left[A\triangle_l^{md} + B \left\|\nabla f(\boldsymbol{x}_l^{md})\right\|^2 + C\right]. \quad (76)$$

With multiplying $A_l$ and taking expectation on both sides of (75), we have

$$\mathbb{E}\left[A_l \triangle_l^{ag}\right] \leq \mathbb{E}\left[A_{l-1} \triangle_{l-1}^{ag}\right] + \frac{A_l - A_{l-1}}{2\lambda_l} \mathbb{E}\left[\|\boldsymbol{x}_{l-1} - \boldsymbol{x}^*\|^2 - \|\boldsymbol{x}_l - \boldsymbol{x}^*\|^2\right]$$

$$- \beta A_l \left(1 - \frac{L\beta}{2} - \frac{\lambda_l (A_l - A_{l-1})}{2\beta A_l}\right) \mathbb{E}\left[\left\|\nabla f(\boldsymbol{x}_l^{md})\right\|^2\right]$$

$$+ \beta A_l \left(\frac{L\beta}{2} + \frac{\lambda (A_l - A_{l-1})}{2\beta A_l}\right) \mathbb{E}\left[\|\boldsymbol{\xi}_l\|^2\right]$$

$$\leq \mathbb{E}\left[A_{l-1} \triangle_{l-1}^{ag}\right] + \frac{A_l - A_{l-1}}{2\lambda_l} \mathbb{E}\left[\|\boldsymbol{x}_{l-1} - \boldsymbol{x}^*\|^2 - \|\boldsymbol{x}_l - \boldsymbol{x}^*\|^2\right]$$

$$- \beta A_l \left(1 - \frac{L\beta}{2} - \frac{\lambda_l (A_l - A_{l-1})}{2\beta A_l}\right) \mathbb{E}\left[\left\|\nabla f(\boldsymbol{x}_l^{md})\right\|^2\right]$$

$$+ \beta A_l \left(\frac{L\beta}{2} + \frac{\lambda_l (A_l - A_{l-1})}{2\beta A_l}\right) \mathbb{E}\left[A\triangle_l^{md} + B \left\|\nabla f(\boldsymbol{x}_l^{md})\right\|^2 + C\right]$$

$$= \mathbb{E}\left[A_{l-1} \triangle_{l-1}^{ag}\right] + \frac{A_l - A_{l-1}}{2\lambda_l} \mathbb{E}\left[\|\boldsymbol{x}_{l-1} - \boldsymbol{x}^*\|^2 - \|\boldsymbol{x}_l - \boldsymbol{x}^*\|^2\right]$$

$$- \beta A_l \left(1 - (1 + B) \left(\frac{L\beta}{2} + \frac{\lambda_l (A_l - A_{l-1})}{2\beta A_l}\right)\right) \mathbb{E}\left[\left\|\nabla f(\boldsymbol{x}_l^{md})\right\|^2\right]$$

$$+ \beta A_l \left(\frac{L\beta}{2} + \frac{\lambda_l (A_l - A_{l-1})}{2\beta A_l}\right) \mathbb{E}\left[A\triangle_l^{md} + C\right], \quad (77)$$

where the second inequality follows from (76). Since $\lambda_k = \frac{\beta}{2(1+B)} (A_k - A_{k-1})$ $\lambda_k = \frac{1}{2}\beta (A_k - A_{k-1})$, we have

$$\frac{A_l - A_{l-1}}{2\lambda_l} = \frac{1 + B}{\beta},$$

and

$$\frac{\lambda_l (A_l - A_{l-1})}{2\beta A_l} = \frac{(A_l - A_{l-1})^2}{4A_l (1 + B)} \leq \frac{1}{4(1 + B)},$$

where the inequality follows from Lemma 4.1. Combining with (77), we have

$$\mathbb{E}\left[A_l \triangle_l^{ag}\right] \leq \mathbb{E}\left[A_{l-1}\triangle_{l-1}^{ag}\right] + \frac{1+B}{\beta}\mathbb{E}\left[\|\boldsymbol{x}_{l-1}-\boldsymbol{x}^*\|^2 - \|\boldsymbol{x}_l-\boldsymbol{x}^*\|^2\right]$$

$$- \beta A_l\left(1-(1+B)\left(\frac{L\beta}{2}+\frac{1}{4(1+B)}\right)\right)\mathbb{E}\left[\left\|\nabla f(\boldsymbol{x}_l^{md})\right\|^2\right]$$

$$+ \beta A_l\left(\frac{L\beta}{2}+\frac{1}{4(1+B)}\right)\mathbb{E}\left[A\triangle_l^{md}+C\right]$$

$$\leq \mathbb{E}\left[A_{l-1}\triangle_{l-1}^{ag}\right] + \frac{1+B}{\beta}\mathbb{E}\left[\|\boldsymbol{x}_{l-1}-\boldsymbol{x}^*\|^2 - \|\boldsymbol{x}_l-\boldsymbol{x}^*\|^2\right]$$

$$- \frac{1}{2}\beta A_l\mathbb{E}\left[\left\|\nabla f(\boldsymbol{x}_l^{md})\right\|^2\right] + \frac{1}{2(1+B)}\beta A_l\mathbb{E}\left[A\triangle_l^{md}+C\right],$$

where the last inequality follows from $\beta \leq \frac{1}{2L(1+B)}$. Re-arranging the above inequality and summing up over $l \in [t]$, we obtain that

$$\mathbb{E}\left[A_t\triangle_t^{ag}\right] + \frac{1+B}{\beta}\mathbb{E}\left[\|\boldsymbol{x}_t-\boldsymbol{x}^*\|^2\right]$$

$$\leq A_0\triangle_0^{ag} + \frac{1+B}{\beta}\|\boldsymbol{x}_0-\boldsymbol{x}^*\|^2 - \frac{1}{2}\beta\sum_{l=1}^{t}A_l\mathbb{E}\left[\left\|\nabla f(\boldsymbol{x}_l^{md})\right\|^2\right]$$

$$+ \frac{1}{2(1+B)}\beta\sum_{l=1}^{t}A_l\mathbb{E}\left[A\triangle_l^{md}+C\right]$$

$$= \frac{\mathcal{C}_3}{\beta} - \frac{1}{2}\beta\sum_{l=1}^{t}A_l\mathbb{E}\left[\left\|\nabla f(\boldsymbol{x}_l^{md})\right\|^2\right] + \frac{1}{2(1+B)}\beta\sum_{l=1}^{t}A_l\mathbb{E}\left[A\triangle_l^{md}+C\right],$$

where the last line holds since $A_0 = 1/\beta$. $\qquad\square$

Similar to Lemma E.7, we will bound the function value gap in expectation by induction.

**Lemma F.2.** *Under the condition of Theorem 3, we have*

$$\mathbb{E}\left[f\left(\boldsymbol{x}_t^{md}\right)-f^*\right] \leq \mathcal{F}_3, \forall t \in [T],$$

*where*

$$\mathcal{F}_3 = \left(2+5\sqrt{2L(1+B)}\right)\mathcal{C}_3 + 1 + 10\sqrt{2L}, \tag{78}$$

*with $\mathcal{C}_3$ defined in (74).*

*Proof.* We will prove this lemma by induction. Obviously, we have $\mathbb{E}\left[f\left(\boldsymbol{x}_1^{md}\right)-f^*\right] = f\left(\boldsymbol{x}_0^{ag}\right) - f^* \leq \mathcal{F}_3$. Suppose that for some $t \in [T]$,

$$\mathbb{E}\left[f\left(\boldsymbol{x}_l^{md}\right)-f^*\right] \leq \mathcal{F}_3, \forall l \in [t].$$

Next, we will bound $\mathbb{E}\left[f\left(\boldsymbol{x}_{t+1}^{md}\right)-f^*\right]$. Since (69) is independent of the smoothness condition, it still holds here.

$$\left\|\boldsymbol{x}_{t+1}^{md}-\boldsymbol{x}_t^{ag}\right\|^2 \leq 2\sum_{i=1}^{t}\frac{\lambda_i^2+\beta^2}{A_i-A_{i-1}}\left\|\nabla f(\boldsymbol{x}_i^{md})+\boldsymbol{\xi}_i\right\|^2$$

$$\leq 2\sum_{i=1}^{t}\frac{\frac{1}{4}(A_i-A_{i-1})^2\beta^2+\beta^2}{A_i-A_{i-1}}\left\|\nabla f(\boldsymbol{x}_i^{md})+\boldsymbol{\xi}_i\right\|^2$$

$$\leq \frac{5}{2}\beta^2\sum_{i=1}^{t}(A_i-A_{i-1})\left\|\nabla f(\boldsymbol{x}_i^{md})+\boldsymbol{\xi}_i\right\|^2$$

$$\leq 5\beta^2\sum_{i=1}^{t}(A_i-A_{i-1})\left(\left\|\nabla f(\boldsymbol{x}_i^{md})\right\|^2+\|\boldsymbol{\xi}_i\|^2\right),$$

where the second inequality holds since the constraint of $\lambda_i$ in (44), the third inequality follows from Lemma 4.1 and the last inequality holds since $\|\boldsymbol{a} + \boldsymbol{b}\|^2 \leq 2 \left( \|\boldsymbol{a}\|^2 + \|\boldsymbol{b}\|^2 \right)$. Applying Lemma 4.1 again and using the fact that $\sqrt{\beta A_t} = \sqrt{\beta \left( B_t + 1/\beta \right)} \geq 1, \forall t \in [T]$, we have

$$\left\| \boldsymbol{x}_{t+1}^{md} - \boldsymbol{x}_t^{ag} \right\|^2 \leq 5\beta^2 \sum_{i=1}^t \sqrt{A_i} \left( \left\| \nabla f(\boldsymbol{x}_i^{md}) \right\|^2 + \|\boldsymbol{\xi}_i\|^2 \right) \leq 5\beta^{\frac{5}{2}} \sum_{i=1}^t A_i \left( \left\| \nabla f(\boldsymbol{x}_i^{md}) \right\|^2 + \|\boldsymbol{\xi}_i\|^2 \right). \tag{79}$$

Taking expectation on both sides of the above inequality and combining with (76), we obtain that

$$\mathbb{E} \left[ \left\| \boldsymbol{x}_{t+1}^{md} - \boldsymbol{x}_t^{ag} \right\|^2 \right] \leq 5\beta^{\frac{5}{2}} \sum_{i=1}^t A_i \left( \mathbb{E} \left[ \left\| \nabla f(\boldsymbol{x}_i^{md}) \right\|^2 \right] + \mathbb{E} \left[ A \triangle_i^{md} + B \left\| \nabla f(\boldsymbol{x}_i^{md}) \right\|^2 + C \right] \right)$$

$$= 5\beta^{\frac{5}{2}} \left( 1 + B \right) \sum_{i=1}^t A_i \mathbb{E} \left[ \left\| \nabla \Psi(x_i^{md}) \right\|^2 \right] + 5\beta^{\frac{5}{2}} \sum_{i=1}^t A_i \mathbb{E} \left[ A \triangle_i^{md} + C \right]$$

$$\leq 10\beta^{\frac{1}{2}} \left( 1 + B \right) \mathcal{C}_3 + 10\beta^{\frac{5}{2}} \sum_{i=1}^t A_i \mathbb{E} \left[ A \triangle_i^{md} + C \right]$$

$$\leq 10\beta^{\frac{1}{2}} \left( 1 + B \right) \mathcal{C}_3 + 10\beta^{\frac{5}{2}} \cdot T \cdot A_T \left( A \mathcal{F}_3 + C \right)$$

$$\leq 10\beta^{\frac{1}{2}} \left( 1 + B \right) \mathcal{C}_3 + 10\beta^{\frac{5}{2}} T^3 \left( A \mathcal{F}_3 + C \right) + 10\beta^{\frac{3}{2}} T \left( A \mathcal{F}_3 + C \right),$$

where the second inequality follows from Lemma F.1, the third inequality holds since $A_i \leq A_T, \forall i \in [t]$ and the assumption that $\mathbb{E} \left[ \triangle_i^{md} \right] \leq \mathcal{F}_3, \forall i \in [t]$, and the last inequality follows from Lemma 4.1 with $A_t = B_t + 1/\beta, \forall t \in [T]$. Since the constraints of $\beta$ in (10), we have

$$\mathbb{E} \left[ \left\| \boldsymbol{x}_{t+1}^{md} - \boldsymbol{x}_t^{ag} \right\|^2 \right] \leq \frac{5\sqrt{2 \left( 1 + B \right)}}{\sqrt{L}} \mathcal{C}_3 + \frac{10\sqrt{2}}{\sqrt{L \left( 1 + B \right)}}.$$

Applying the descent lemma again, we obtain that

$$f \left( \boldsymbol{x}_{t+1}^{md} \right) \leq f \left( \boldsymbol{x}_t^{ag} \right) + \left\langle \nabla f(\boldsymbol{x}_t^{ag}), \boldsymbol{x}_{t+1}^{md} - \boldsymbol{x}_t^{ag} \right\rangle + \frac{L}{2} \left\| \boldsymbol{x}_{t+1}^{md} - \boldsymbol{x}_t^{ag} \right\|^2$$

$$\leq f \left( \boldsymbol{x}_t^{ag} \right) + \left\| \nabla f(\boldsymbol{x}_t^{ag}) \right\| \cdot \left\| \boldsymbol{x}_{t+1}^{md} - \boldsymbol{x}_t^{ag} \right\| + \frac{L}{2} \left\| \boldsymbol{x}_{t+1}^{md} - \boldsymbol{x}_t^{ag} \right\|^2$$

$$\leq f \left( \boldsymbol{x}_t^{ag} \right) + \frac{1}{2L} \left\| \nabla f(\boldsymbol{x}_t^{ag}) \right\|^2 + \frac{L}{2} \left\| \boldsymbol{x}_{t+1}^{md} - \boldsymbol{x}_t^{ag} \right\|^2 + \frac{L}{2} \left\| \boldsymbol{x}_{t+1}^{md} - \boldsymbol{x}_t^{ag} \right\|^2$$

$$\leq f \left( \boldsymbol{x}_t^{ag} \right) + \left( f \left( \boldsymbol{x}_t^{ag} \right) - f^* \right) + L \cdot \left\| \boldsymbol{x}_{t+1}^{md} - \boldsymbol{x}_t^{ag} \right\|^2,$$

where we apply Cauchy-Schwarz inequality in the second inequality and apply Young's inequality in the third line. The last inequality follows from Lemma C.1. Subtracting $f^*$ from both sides and taking expectation, we have

$$\mathbb{E} \left[ \triangle_{t+1}^{md} \right] \leq 2\mathbb{E} \left[ \triangle_t^{ag} \right] + L \cdot \mathbb{E} \left[ \left\| \boldsymbol{x}_{t+1}^{md} - \boldsymbol{x}_t^{ag} \right\|^2 \right].$$

With the assumption that $f \left( \boldsymbol{x}_l^{md} \right) - f^* \leq \mathcal{F}_3, \forall l \in [t]$ and applying Lemma F.1, we obtain that

$$\mathbb{E} \left[ \triangle_t^{ag} \right] \leq \frac{1}{A_t \beta} \mathcal{C}_3 + \frac{1}{2A_t \left( 1 + B \right)} \beta \cdot \sum_{i=1}^t A_i \mathbb{E} \left[ A \triangle_l^{md} + C \right]$$

$$\leq \mathcal{C}_3 + \frac{1}{2 \left( 1 + B \right)} \beta T \left( A \mathcal{F}_3 + C \right) \leq \mathcal{C}_3 + \frac{1}{2},$$

where the second inequality holds since $A_t \geq 1/\beta$ and $A_i \leq A_t, \forall \in [t]$, and the last line follows from the definition of $\beta$. Therefore, we have

$$\mathbb{E} \left[ \triangle_{t+1}^{md} \right] \leq \left( 2 + 5\sqrt{2L \left( 1 + B \right)} \right) \mathcal{C}_3 + 1 + 10\sqrt{2L} = \mathcal{F}_3.$$

Now we finish the induction and obtain the desired result. $\qquad\square$

Based on Lemma F.1 and Lemma F.2, we could obtain the final convergence rate.

*Proof of Theorem 3.* By Lemma F.2, we have $\mathbb{E}\left[f\left(\boldsymbol{x}_t^{md}\right) - f^*\right] \leq \mathcal{F}_3, \forall t \in [T]$. Then, combining Lemma F.1, Assumption 4 and the fact that $A_t \leq A_T, \forall t \in [T]$, we obtain that

$$
\begin{aligned}
\mathbb{E}\left[f\left(\boldsymbol{x}_T^{ag}\right) - f^*\right] \leq & \frac{1}{A_T\beta}\mathcal{C}_3 + \frac{\beta}{2A_T\left(1+B\right)}T \cdot A_T\left(A\mathcal{F}_3 + C\right) \\
\leq & \frac{8L\left(1+B\right)\mathcal{C}_3}{T^2} + \frac{4\mathcal{C}_3\mathcal{Q}}{T} + \frac{4\mathcal{C}_3\sqrt{\mathcal{Q}}}{\sqrt{T}} + \frac{\sqrt{\mathcal{Q}}}{2\sqrt{T}},
\end{aligned}
\tag{80}
$$

where the second inequality holds since Lemma 4.1 and the setting of $\beta$ in (10). $\qquad\square$

# G  NON-CONVEX OPTIMIZATION

In this section, we present Stochastic Accelerated Gradient Descent (stochastic AGD) (Algorithm 3) and its convergence analysis. Algorithm 3 could reduce to some famous algorithms, such as SGD, and was well studied in (Ghadimi & Lan, 2016; Kavis et al., 2022; Yu et al., 2025). SNAG (Algorithm 2) can be viewed a special case of Algorithm 3. To apply our theoretical analysis from the convex case to the non-convex case, we adopt a different step size setting.

---

**Algorithm 3** Stochastic Accelerated Gradient Descent (stochastic AGD)

---

**Require:** Horizon $T$, $\boldsymbol{x}_0^{ag} = \boldsymbol{x}_0 \in \mathbb{R}^d$, step sizes $\{\beta_t\}_{t\in[T]}, \{\lambda_t\}_{t\in[T]}$.
1: **for** $t = 1, \cdots, T$ **do**
2: $\quad \boldsymbol{x}_t^{md} = (1 - \alpha_t)\,\boldsymbol{x}_{t-1}^{ag} + \alpha_t\boldsymbol{x}_{t-1}$;
3: $\quad$ Set $\boldsymbol{g}_t = \nabla f_{\mathbf{z}}\left(\boldsymbol{x}_t^{md}; \mathbf{z}_t\right)$;
4: $\quad \boldsymbol{x}_t = \boldsymbol{x}_{t-1} - \lambda_t\boldsymbol{g}_t$;
5: $\quad \boldsymbol{x}_t^{ag} = \boldsymbol{x}_t^{md} - \beta_t\boldsymbol{g}_t$.

---

We have the following results for the above algorithm.

**Theorem 4.** *Let $T > 0$ and $f$ be an $(L_0, L_1, L_2)$-smooth function. Under Assumptions 1-3, consider Algorithm 3 with $\alpha_t = \frac{2}{t+1}$, $\lambda_t = \eta$ and $\beta_t = \eta\alpha_t + \lambda_t, \forall t \in [T]$. Let*

$$
\eta = \min\left\{\frac{1}{(L_1+L_2)\mathcal{Y}}, \frac{1}{8\mathcal{Y}_1\left(B\log\frac{Te}{\delta}+1\right)}, \frac{1}{4\sqrt{A\mathcal{Y}_1 T\log\frac{Te}{\delta}}\mathcal{F}_4}, \frac{1}{4\sqrt{C\mathcal{Y}_1 T\log\frac{Te}{\delta}}}, \frac{1}{6\mathcal{P}_c^2\log\frac{Te}{\delta}}\right\},
$$

*where*

$$
\mathcal{Y} = \sqrt{A\log\frac{Te}{\delta}\mathcal{F}_4} + \left(\sqrt{B\log\frac{Te}{\delta}}+1\right)\sqrt{g\left(\mathcal{F}_4\right)} + \sqrt{C\log\frac{Te}{\delta}},
\tag{81}
$$

$$
\mathcal{Y}_1 = L_0 + L_1\left(g\left(\mathcal{K}\right)\right)^{\frac{p}{2}} + L_2\mathcal{K}^q,
$$

$$
\mathcal{P}_c = \sqrt{A\mathcal{F}_4 + Bg\left(\mathcal{F}_4\right) + C},
\tag{82}
$$

$$
\mathcal{K} = \mathcal{F}_4 + \frac{1}{L_1+L_2}\sqrt{g(\mathcal{F}_4)} + \frac{L_0 + L_1\left(g(\mathcal{F}_4)\right)^{\frac{p}{2}} + L_2\mathcal{F}_4^q}{2\left(L_1+L_2\right)^2},
$$

$$
\mathcal{F}_4 = \triangle_1^{md} + 1 + \frac{1}{L_1+L_2}\sqrt{g(1+\triangle_1^{md})} + \frac{L_0 + L_1\left(g(1+\triangle_1^{md})\right)^{\frac{p}{2}} + L_2\left(1+\triangle_1^{md}\right)^q}{2\left(L_1+L_2\right)^2},
$$

*and $g$ is the function given by (24). Then with probability at least $1 - 2\delta$,*

$$
\begin{aligned}
\frac{1}{T}\sum_{l=1}^T\left\|\nabla f(\boldsymbol{x}_l^{md})\right\|^2 \leq & \frac{2\left(1+\triangle_1^{md}\right)}{T}\left(L_1+L_2\right)\mathcal{Y} + \frac{16\left(1+\triangle_1^{md}\right)}{T}\mathcal{Y}_1\left(B\log\frac{Te}{\delta}+1\right) \\
& + \frac{8\left(1+\triangle_1^{md}\right)}{\sqrt{T}}\left(\sqrt{A\mathcal{F}_4} + \sqrt{C}\right)\sqrt{\mathcal{Y}_1\log\frac{Te}{\delta}} \\
& + \frac{12\left(1+\triangle_1^{md}\right)}{T}\mathcal{P}_c^2\log\frac{Te}{\delta}.
\end{aligned}
\tag{83}
$$

The upper rate from (83) is of order $\tilde{O}(1/T + \sqrt{(A+C)/T})$, which matches that in (Ghadimi & Lan, 2016) for stochastic AGD with bounded variances and also the lower rate in (Arjevani et al., 2023) of finding stationary points in non-convex smooth stochastic optimizations with bounded variances when $C > 0$.

Under the $(L_0, L_1)$-smoothness assumption, Yu et al. (2025) analyzed stochastic AGD for non-convex objective functions and they proved that the average of the squared norm converges at the rate of $\tilde{\mathcal{O}}\left(1/T + \sqrt{(A+C)/T}\right)$ with high probability. Here, we follow the analytical approach from (Yu et al., 2025) and make slight modifications to the proof methods to accommodate the more general smooth assumptions. To prove the theorem, we first provide several useful lemmas following from (Ghadimi & Lan, 2016; Kavis et al., 2022; Yu et al., 2025).

**Proposition G.1** (Proposition 5.2 in (Kavis et al., 2022)). *Denote $\alpha_t = \frac{2}{t+1}$ and $\Gamma_t = (1 - \alpha_t)\Gamma_{t-1}$ with $\Gamma_1 = 1$, $\forall t \in [T]$. We have that for all $t \in [T]$,*

$$\Gamma_t \sum_{k=1}^t \frac{\alpha_k}{\Gamma_k} = 1, \tag{84}$$

*and*

$$\left[\sum_{k=t}^T (1 - \alpha_k)\,\Gamma_k\right] \frac{\alpha_t}{\Gamma_t} \leq 2. \tag{85}$$

**Lemma G.1.** *Given $T \geq 1$ and $\delta \in (0,1)$, if Assumptions 2 and 3 hold, then with probability at least $1 - \delta$,*

$$\sum_{k=1}^l -\left\langle \nabla f(\boldsymbol{x}_k^{md}), \boldsymbol{\xi}_k \right\rangle \leq \frac{1}{4} \sum_{k=1}^l \frac{\mathcal{P}_k^2}{\mathcal{P}_c^2} \left\| \nabla f(\boldsymbol{x}_k^{md}) \right\|^2 + 3\mathcal{P}_c^2 \log \frac{T}{\delta}, \quad \forall l \in [T], \tag{86}$$

*where*

$$\mathcal{P}_k = \sqrt{A\triangle_k^{md} + Bg\left(\triangle_k^{md}\right) + C}, \tag{87}$$

*and $\mathcal{P}_c$ is given by (82).*

*Proof.* Let $Z_k = -\left\langle \nabla f(\boldsymbol{x}_k^{md}), \boldsymbol{\xi}_k \right\rangle$. Note that $\nabla f(\boldsymbol{x}_k^{md})$ is a random variable dependent on $\mathbf{z}_1, \cdots, \mathbf{z}_{k-1}$ and $\boldsymbol{\xi}_k$ is dependent on $\mathbf{z}_1, \cdots, \mathbf{z}_k$. Therefore, it is apparent that $Z_k$ is a martingale difference sequence since

$$\mathbb{E}\left[-\left\langle \nabla f(\boldsymbol{x}_k^{md}), \boldsymbol{\xi}_t \right\rangle | \mathbf{z}_1, \cdots, \mathbf{z}_{k-1}\right] = -\left\langle \nabla f(\boldsymbol{x}_k^{md}), \mathbb{E}_k[\boldsymbol{\xi}_k] \right\rangle = 0.$$

Also by Assumption 3 and applying Cauchy-Schwarz inequality, we obtain that

$$\mathbb{E}_k\left[\exp\left(\frac{Z_k^2}{\left\|\nabla f(\boldsymbol{x}_k^{md})\right\|^2 \left(A\triangle_k^{md} + B\left\|\nabla f(\boldsymbol{x}_k^{md})\right\|^2 + C\right)}\right)\right]$$

$$\leq \mathbb{E}_k\left[\exp\left(\frac{\left\|\nabla f(\boldsymbol{x}_k^{md})\right\|^2 \left\|\boldsymbol{\xi}_k\right\|^2}{\left\|\nabla f(\boldsymbol{x}_k^{md})\right\|^2 \left(A\triangle_k^{md} + B\left\|\nabla f(\boldsymbol{x}_k^{md})\right\|^2 + C\right)}\right)\right] \leq \mathrm{e}.$$

Therefore, given any $l \in [T]$, applying Lemma C.4, we have that for any $\lambda > 0$, with probability at least $1 - \delta$,

$$\sum_{k=1}^l Z_t \leq \frac{3\lambda}{4} \sum_{k=1}^l \left\|\nabla f(\boldsymbol{x}_k^{md})\right\|^2 \left(A\triangle_k^{md} + B\left\|\nabla f(\boldsymbol{x}_k^{md})\right\|^2 + C\right) + \frac{1}{\lambda}\log\frac{1}{\delta}$$

$$\leq \frac{3\lambda}{4} \sum_{k=1}^l \left\|\nabla f(\boldsymbol{x}_k^{md})\right\|^2 \mathcal{P}_k^2 + \frac{1}{\lambda}\log\frac{1}{\delta}$$

where $\mathcal{P}_k$ is defined in (87). For any fixed $\lambda$, we can re-scale over $\delta$ and have that with probability at least $1 - \delta$, for all $l \in [T]$,

$$\sum_{k=1}^{l} - \left\langle \nabla f(\boldsymbol{x}_k^{md}), \boldsymbol{\xi}_k \right\rangle \leq \frac{3\lambda}{4} \sum_{t=1}^{l} \left\| \nabla f(\boldsymbol{x}_k^{md}) \right\|^2 \mathcal{P}_k^2 + \frac{1}{\lambda} \log \frac{T}{\delta}. \tag{88}$$

Let $\lambda = \frac{1}{3\mathcal{P}_c^2}$, and we obtain the desired result. $\qquad\square$

**Proposition G.2.** *Let $\{\boldsymbol{x}_t\}_{t \in [T]}$ and $\{\boldsymbol{x}_t^{md}\}_{t \in [T]}$ be generated by Algorithm 3. We have*

$$\boldsymbol{x}_t^{md} - \boldsymbol{x}_{t-1} = (1 - \alpha_t)\, \Gamma_{t-1} \sum_{k=1}^{t-1} \frac{\alpha_k}{\Gamma_k} \frac{(\lambda_k - \beta_k)}{\alpha_k} \boldsymbol{g}_k, \tag{89}$$

*and*

$$\left\| \boldsymbol{x}_t^{md} - \boldsymbol{x}_{t-1} \right\|^2 \leq (1 - \alpha_t)\, \Gamma_t \sum_{k=1}^{t-1} \frac{\alpha_k}{\Gamma_k} \frac{(\lambda_k - \beta_k)^2}{\alpha_k^2} \left\| \boldsymbol{g}_k \right\|^2. \tag{90}$$

*Proof.* From Algorithm 3, we have

$$\boldsymbol{x}_k^{ag} - \boldsymbol{x}_k = \boldsymbol{x}_k^{md} - \beta_k \boldsymbol{g}_k - \boldsymbol{x}_{k-1} + \lambda_k \boldsymbol{g}_k = (1 - \alpha_k)\left( \boldsymbol{x}_{k-1}^{ag} - \boldsymbol{x}_{k-1} \right) + (\lambda_k - \beta_k)\, \boldsymbol{g}_k.$$

Since $\boldsymbol{x}_0^{ag} = \boldsymbol{x}_0$, we obtain that

$$\boldsymbol{x}_k^{ag} - \boldsymbol{x}_k = \sum_{i=1}^{k} \left( \prod_{j=i+1}^{k} (1 - \alpha_j) \right) (\lambda_i - \beta_i)\, \boldsymbol{g}_i = \Gamma_k \sum_{i=1}^{k} \frac{1}{\Gamma_i} (\lambda_i - \beta_i)\, \boldsymbol{g}_i.$$

Taking the norm function on both sides and applying the triangle inequality, we have

$$\left\| \boldsymbol{x}_k^{ag} - \boldsymbol{x}_k \right\| \leq \Gamma_k \sum_{i=1}^{k} \frac{1}{\Gamma_i} |\lambda_i - \beta_i| \cdot \left\| \boldsymbol{g}_i \right\| = \Gamma_k \sum_{i=1}^{k} \frac{\alpha_i}{\Gamma_i} \frac{|\lambda_i - \beta_i|}{\alpha_i} \cdot \left\| \boldsymbol{g}_i \right\|. \tag{91}$$

By the iteration step in Algorithm 3, we have

$$\boldsymbol{x}_k^{md} - \boldsymbol{x}_{k-1} = (1 - \alpha_k)\left( \boldsymbol{x}_{k-1}^{ag} - \boldsymbol{x}_{k-1} \right).$$

Combining with (91), we obtain that

$$\left\| \boldsymbol{x}_k^{md} - \boldsymbol{x}_{k-1} \right\| = (1 - \alpha_k) \left\| \boldsymbol{x}_{k-1}^{ag} - \boldsymbol{x}_{k-1} \right\| \leq (1 - \alpha_k)\, \Gamma_{k-1} \sum_{i=1}^{k-1} \frac{\alpha_i}{\Gamma_i} \frac{|\lambda_i - \beta_i|}{\alpha_i} \cdot \left\| \boldsymbol{g}_i \right\|.$$

Similarly, by the convexity of norm square and (84),

$$\left\| \boldsymbol{x}_k^{md} - \boldsymbol{x}_{k-1} \right\|^2 \leq (1 - \alpha_k)^2\, \Gamma_{k-1} \sum_{i=1}^{k-1} \frac{\alpha_i}{\Gamma_i} \frac{(\lambda_i - \beta_i)^2}{\alpha_k^2} \left\| \boldsymbol{g}_i \right\|^2 = (1 - \alpha_k)\, \Gamma_k \sum_{i=1}^{k-1} \frac{\alpha_i}{\Gamma_i} \frac{(\lambda_i - \beta_i)^2}{\alpha_k^2} \left\| \boldsymbol{g}_i \right\|^2.$$

$\qquad\square$

**Lemma G.2.** *Let $\{a_t\}_{t \in [n]}$ be a sequence of non-negative real numbers. We have*

$$\sqrt{\sum_{i=1}^{n} a_i} \leq \sum_{i=1}^{n} \sqrt{a_i}.$$

In the following analysis, denote $\triangle_t = f(\boldsymbol{x}_t) - f^*$ for simplicity.

**Proposition G.3.** *Under the conditions and notations of Theorem 4, $\triangle_t^{md} \leq \mathcal{F}_4, \forall t \in [T]$, hold with probability at least $1 - \delta$.*

*Proof.* We assume that (47) and (86) always happen and then deduce $\triangle_t^{md} \leq \mathcal{F}_4$ for all $t \in [T]$. Since (47) and (86) happen with probability at least $1 - \delta$ separately, $\triangle_t^{md} \leq \mathcal{F}_4, \forall t \in [T]$, holds with probability at least $1 - 2\delta$. It is obvious that $f(\boldsymbol{x}_1^{md}) - f^* \leq \mathcal{F}_4$. Therefore, by Corollary 1 we have $\mathcal{P}_1 \leq \mathcal{P}_c$. Suppose that for some $t \in [T]$,

$$f(\boldsymbol{x}_l^{md}) - f^* \leq \mathcal{F}_4, \quad \forall l \in [t].$$

By the triangle inequality of the norm function, we have that for all $l \in [t]$,

$$\|\boldsymbol{g}_l\| \leq \|\boldsymbol{g}_l - \nabla f(\boldsymbol{x}_l^{md})\| + \|\nabla f(\boldsymbol{x}_l^{md})\|$$

$$\leq \sqrt{\left(A\triangle_l^{md} + B\left\|\nabla f(\boldsymbol{x}_l^{md})\right\|^2 + C\right)\log\frac{Te}{\delta}} + \|\nabla f(\boldsymbol{x}_l^{md})\|$$

$$\leq \sqrt{A\log\frac{Te}{\delta}\triangle_l^{md}} + \left(\sqrt{B\log\frac{Te}{\delta}} + 1\right)\|\nabla f(\boldsymbol{x}_l^{md})\| + \sqrt{C\log\frac{Te}{\delta}}, \qquad (92)$$

where the second inequality follows from Lemma E.1 and the last inequality follows from Lemma G.2. Combining with Corollary 1 and the assumption that $\triangle_l^{md} \leq \mathcal{F}_4, \forall l \in [t]$, we have

$$\|\boldsymbol{g}_l\| \leq \mathcal{Y}, \quad \forall l \in [t]. \qquad (93)$$

By the iteration step of Algorithm 3, we have

$$\|\boldsymbol{x}_l - \boldsymbol{x}_{l-1}\| = \lambda_l\|\boldsymbol{g}_l\| = \eta\|\boldsymbol{g}_l\| \leq \eta\mathcal{Y} \leq \min\{1/L_1, 1/L_2\}, \quad \forall l \in [t],$$

where the last inequality follows from the restriction of $\eta$. Thus, we could apply Lemma C.2 and obtain that

$$f(\boldsymbol{x}_l) - f(\boldsymbol{x}_{l-1})$$

$$\leq \langle\nabla f(\boldsymbol{x}_{l-1}), \boldsymbol{x}_l - \boldsymbol{x}_{l-1}\rangle + \frac{L_0 + L_1\|\nabla f(\boldsymbol{x}_{l-1})\|^p + L_2\triangle_{l-1}^q}{2}\|\boldsymbol{x}_l - \boldsymbol{x}_{l-1}\|^2$$

$$= -\eta\langle\nabla f(\boldsymbol{x}_{l-1}), \boldsymbol{g}_l\rangle + \frac{L_0 + L_1\|\nabla f(\boldsymbol{x}_{l-1})\|^p + L_2\triangle_{l-1}^q}{2}\eta^2\|\boldsymbol{g}_l\|^2$$

$$= -\eta\langle\nabla f(\boldsymbol{x}_l^{md}) + \nabla f(\boldsymbol{x}_{l-1}) - \nabla f(\boldsymbol{x}_l^{md}), \nabla f(\boldsymbol{x}_l^{md}) + \boldsymbol{\xi}_l\rangle$$

$$\quad + \frac{L_0 + L_1\|\nabla f(\boldsymbol{x}_{l-1})\|^p + L_2\triangle_{l-1}^q}{2}\eta^2\|\boldsymbol{g}_l\|^2$$

$$= -\eta\left\|\nabla f(\boldsymbol{x}_l^{md})\right\|^2 - \eta\langle\nabla f(\boldsymbol{x}_l^{md}), \boldsymbol{\xi}_l\rangle - \eta\langle\nabla f(\boldsymbol{x}_{l-1}) - \nabla f(\boldsymbol{x}_l^{md}), \boldsymbol{g}_l\rangle$$

$$\quad + \frac{L_0 + L_1\|\nabla f(\boldsymbol{x}_{l-1})\|^p + L_2\triangle_{l-1}^q}{2}\eta^2\|\boldsymbol{g}_l\|^2$$

$$\leq -\eta\left\|\nabla f(\boldsymbol{x}_l^{md})\right\|^2 - \eta\langle\nabla f(\boldsymbol{x}_l^{md}), \boldsymbol{\xi}_l\rangle + \eta\left\|\nabla f(\boldsymbol{x}_{l-1}) - \nabla f(\boldsymbol{x}_l^{md})\right\|\|\boldsymbol{g}_l\|$$

$$\quad + \frac{L_0 + L_1\|\nabla f(\boldsymbol{x}_{l-1})\|^p + L_2\triangle_{l-1}^q}{2}\eta^2\|\boldsymbol{g}_l\|^2,$$

where the first equation follows from the update rule in Algorithm 3 and the last line follows from Cauchy-Schwarz inequality. Applying Lemma G.2 with $\frac{\beta_t - \lambda_t}{\alpha_t} = \lambda_t = \eta$, we have

$$\left\|\boldsymbol{x}_l^{md} - \boldsymbol{x}_{l-1}\right\| = (1 - \alpha_l)\Gamma_{l-1}\left\|\sum_{k=1}^{l-1}\frac{\alpha_k}{\Gamma_k}\frac{(\lambda_k - \beta_k)}{\alpha_k}\boldsymbol{g}_k\right\| \leq (1 - \alpha_l)\Gamma_{l-1}\sum_{k=1}^{l-1}\frac{\alpha_k}{\Gamma_k}\eta\|\boldsymbol{g}_k\|$$

$$\leq \eta\mathcal{Y}\Gamma_{t-1}\sum_{k=1}^{l-1}\frac{\alpha_k}{\Gamma_k} \leq \min\{1/L_1, 1/L_2\}, \qquad (94)$$

where the first inequality follows from the triangle inequality and the second inequality holds since (92). The last inequality follows from (84). Note that $\|\boldsymbol{g}_l\| \leq \mathcal{Y}$ for all $l \in [t]$ and $\left\|\boldsymbol{x}_l^{md} - \boldsymbol{x}_{l-1}\right\|$

depends on $\boldsymbol{g}_1, \cdots, \boldsymbol{g}_{l-1}$. Thus, (94) holds for all $l \in [t+1]$. Applying Definition 1, we have that

$$
\begin{aligned}
& f(\boldsymbol{x}_l) - f(\boldsymbol{x}_{l-1}) \\
\leq & - \eta \left\| \nabla f(\boldsymbol{x}_l^{md}) \right\|^2 - \eta \left\langle \nabla f(\boldsymbol{x}_l^{md}), \boldsymbol{\xi}_l \right\rangle + \eta \left( L_0 + L_1 \left\| \nabla f(\boldsymbol{x}_{l-1}) \right\|^p + L_2 \triangle_{l-1}^q \right) \left\| \boldsymbol{x}_{l-1} - \boldsymbol{x}_l^{md} \right\| \left\| \boldsymbol{g}_l \right\| \\
& + \frac{L_0 + L_1 \left\| \nabla f(\boldsymbol{x}_{l-1}) \right\|^p + L_2 \triangle_{l-1}^q}{2} \eta^2 \left\| \boldsymbol{g}_l \right\|^2 \\
\leq & - \eta \left\| \nabla f(\boldsymbol{x}_l^{md}) \right\|^2 - \eta \left\langle \nabla f(\boldsymbol{x}_l^{md}), \boldsymbol{\xi}_l \right\rangle + \frac{L_0 + L_1 \left\| \nabla f(\boldsymbol{x}_{l-1}) \right\|^p + L_2 \triangle_{l-1}^q}{2} \left\| \boldsymbol{x}_{l-1} - \boldsymbol{x}_l^{md} \right\|^2 \\
& + \left( L_0 + L_1 \left\| \nabla f(\boldsymbol{x}_{l-1}) \right\|^p + L_2 \triangle_{l-1}^q \right) \eta^2 \left\| \boldsymbol{g}_l \right\|^2 \\
\leq & - \eta \left\| \nabla f(\boldsymbol{x}_l^{md}) \right\|^2 - \eta \left\langle \nabla f(\boldsymbol{x}_l^{md}), \boldsymbol{\xi}_l \right\rangle \\
& + \frac{\eta^2}{2} \left( L_0 + L_1 \left\| \nabla f(\boldsymbol{x}_{l-1}) \right\|^p + L_2 \triangle_{l-1}^q \right) (1 - \alpha_l) \Gamma_l \sum_{k=1}^{l-1} \frac{\alpha_k}{\Gamma_k} \left\| \boldsymbol{g}_k \right\|^2 \\
& + \eta^2 \left( L_0 + L_1 \left\| \nabla f(\boldsymbol{x}_{l-1}) \right\|^p + L_2 \triangle_{l-1}^q \right) \left\| \boldsymbol{g}_l \right\|^2 , \tag{95}
\end{aligned}
$$

where the second inequality follows from the fact that $ab \leq \frac{a^2 + b^2}{2}$ and the last inequality follows from (90). Summing up the above inequality over $l \in [t]$, we obtain that

$$
\begin{aligned}
f(\boldsymbol{x}_t) - f(\boldsymbol{x}_0) \leq & \frac{\eta^2}{2} \sum_{l=1}^{t} \left[ \left( L_0 + L_1 \left\| \nabla f(\boldsymbol{x}_{l-1}) \right\|^p + L_2 \triangle_{l-1}^q \right) (1 - \alpha_l) \Gamma_l \sum_{k=1}^{l} \frac{\alpha_k}{\Gamma_k} \left\| \boldsymbol{g}_k \right\|^2 \right] \\
& + \eta^2 \sum_{l=1}^{t} \left( L_0 + L_1 \left\| \nabla f(\boldsymbol{x}_{l-1}) \right\|^p + L_2 \triangle_{l-1}^q \right) \left\| \boldsymbol{g}_l \right\|^2 \\
& - \eta \sum_{l=1}^{t} \left\| \nabla f(\boldsymbol{x}_l^{md}) \right\|^2 - \eta \sum_{l=1}^{t} \left\langle \nabla f(\boldsymbol{x}_l^{md}), \boldsymbol{\xi}_l \right\rangle \\
\leq & \frac{\eta^2}{2} \sum_{l=1}^{t} \left[ \sum_{k=l}^{t} \left( L_0 + L_1 \left\| \nabla f(\boldsymbol{x}_{k-1}) \right\|^p + L_2 \triangle_{k-1}^q \right) (1 - \alpha_k) \Gamma_k \right] \frac{\alpha_l}{\Gamma_l} \left\| \boldsymbol{g}_l \right\|^2 \\
& + \eta^2 \sum_{l=1}^{t} \left( L_0 + L_1 \left\| \nabla f(\boldsymbol{x}_{l-1}) \right\|^p + L_2 \triangle_{l-1}^q \right) \left\| \boldsymbol{g}_l \right\|^2 \\
& - \eta \sum_{l=1}^{t} \left\| \nabla f(\boldsymbol{x}_l^{md}) \right\|^2 - \eta \sum_{l=1}^{t} \left\langle \nabla f(\boldsymbol{x}_l^{md}), \boldsymbol{\xi}_l \right\rangle . \tag{96}
\end{aligned}
$$

By (94), we have that $\left\| \boldsymbol{x}_l^{md} - \boldsymbol{x}_{l-1} \right\| \leq \min \{ 1/L_1, 1/L_2 \}$ for all $l \in [t+1]$. Thus, applying Lemma C.2 again, we obtain that

$$
\begin{aligned}
f(\boldsymbol{x}_{l-1}) \leq & f(\boldsymbol{x}_l^{md}) + \left\langle \nabla f(\boldsymbol{x}_l^{md}), \boldsymbol{x}_{l-1} - \boldsymbol{x}_l^{md} \right\rangle \\
& + \frac{L_0 + L_1 \left\| \nabla f(\boldsymbol{x}_l^{md}) \right\|^p + L_2 \left( \triangle_l^{md} \right)^q}{2} \left\| \boldsymbol{x}_{l-1} - \boldsymbol{x}_l^{md} \right\|^2 \\
\leq & f(\boldsymbol{x}_l^{md}) + \left\| \nabla f(\boldsymbol{x}_l^{md}) \right\| \cdot \left\| \boldsymbol{x}_{l-1} - \boldsymbol{x}_l^{md} \right\| \\
& + \frac{L_0 + L_1 \left\| \nabla f(\boldsymbol{x}_t^{md}) \right\|^p + L_2 \left( \triangle_l^{md} \right)^q}{2} \left\| \boldsymbol{x}_{l-1} - \boldsymbol{x}_l^{md} \right\|^2 ,
\end{aligned}
$$

where the second inequality follows from Cauchy-Schwarz inequality. Subtracting $f^*$ from both sides and applying the assumption that $\triangle_l^{md} \leq \mathcal{F}_4, \forall l \in [t]$, we have

$$
f(\boldsymbol{x}_{l-1}) - f^* \leq \mathcal{F}_4 + \frac{1}{L_1 + L_2} \sqrt{g(\mathcal{F}_4)} + \frac{L_0 + L_1 \left( g(\mathcal{F}_4) \right)^{\frac{p}{2}} + L_2 \mathcal{F}_4^q}{2 \left( L_1 + L_2 \right)^2} = \mathcal{K}, \quad \forall l \in [t].
$$

Thus, by Corollary 1, we have $\|\nabla f(\boldsymbol{x}_l)\| \leq \sqrt{g(\mathcal{K})}$ for all $l \in [t-1]$. Combining with (96), we obtain that

$$
\begin{aligned}
f(\boldsymbol{x}_t) - f(\boldsymbol{x}_0) \leq & \frac{\eta^2}{2}\mathcal{Y}_1 \sum_{l=1}^{t} \left[ \sum_{k=l}^{t} (1-\alpha_k)\,\varGamma_k \right] \frac{\alpha_l}{\varGamma_l} \|\boldsymbol{g}_l\|^2 + \eta^2 \mathcal{Y}_1 \sum_{l=1}^{t} \|\boldsymbol{g}_l\|^2 \\
& - \eta \sum_{t=1}^{l} \left\| \nabla f(\boldsymbol{x}_k^{md}) \right\|^2 - \eta \sum_{t=1}^{l} \left\langle \nabla f(\boldsymbol{x}_k^{md}), \boldsymbol{\xi}_t \right\rangle \\
\leq & 2\eta^2 \mathcal{Y}_1 \sum_{l=1}^{t} \|\boldsymbol{g}_l\|^2 - \eta \sum_{l=1}^{t} \left\| \nabla f(\boldsymbol{x}_l^{md}) \right\|^2 - \eta \sum_{l=1}^{t} \left\langle \nabla f(\boldsymbol{x}_l^{md}), \boldsymbol{\xi}_l \right\rangle, \quad (97)
\end{aligned}
$$

where the second inequality follows from (85). Using the fact that $\|\boldsymbol{a} + \boldsymbol{b}\|^2 \leq 2\|\boldsymbol{a}\|^2 + 2\|\boldsymbol{b}\|^2$ and applying (47), we have that for all $l \in [t]$,

$$
\begin{aligned}
\|\boldsymbol{g}_l\|^2 \leq & 2\|\boldsymbol{\xi}_l\|^2 + 2\left\| \nabla f(\boldsymbol{x}_k^{md}) \right\|^2 \\
\leq & 2\left( A\triangle_l^{md} + B\left\| \nabla f(\boldsymbol{x}_l^{md}) \right\|^2 + C \right) \log \frac{Te}{\delta} + 2\left\| \nabla f(\boldsymbol{x}_l^{md}) \right\|^2 \\
= & 2\left( A\log\frac{Te}{\delta}\triangle_l^{md} + \left( B\log\frac{Te}{\delta} + 1 \right)\left\| \nabla f(\boldsymbol{x}_l^{md}) \right\|^2 + C\log\frac{Te}{\delta} \right).
\end{aligned}
$$

Combining with (97) and applying Lemma G.1 to the summation of the martingale difference sequence, we obtain that

$$
\begin{aligned}
f(\boldsymbol{x}_t) - f(\boldsymbol{x}_0) \leq & 4\eta^2 \mathcal{Y}_1 \left( B\log\frac{Te}{\delta} + 1 \right) \sum_{l=1}^{t} \left\| \nabla f(\boldsymbol{x}_l^{md}) \right\|^2 + 4\eta^2 \mathcal{Y}_1 A\log\frac{Te}{\delta} \sum_{l=1}^{t} \triangle_l^{md} \\
& + 4\eta^2 t \mathcal{Y}_1 C\log\frac{Te}{\delta} - \eta \sum_{l=1}^{t} \left\| \nabla f(\boldsymbol{x}_l^{md}) \right\|^2 + \frac{1}{4}\eta \sum_{l=1}^{t} \frac{\mathcal{P}_l^2}{\mathcal{P}_c^2}\left\| \nabla f(\boldsymbol{x}_l^{md}) \right\|^2 + 3\eta\mathcal{P}_c^2 \log\frac{T}{\delta} \\
\leq & -\frac{\eta}{2} \sum_{l=1}^{t} \left\| \nabla f(\boldsymbol{x}_l^{md}) \right\|^2 + \frac{1}{4} + \frac{1}{4} + \frac{1}{2}. \quad (98)
\end{aligned}
$$

Since $\boldsymbol{x}_1^{md} = (1-\alpha_1)\boldsymbol{x}_0^{ag} + \alpha_1\boldsymbol{x}_0$ and $\boldsymbol{x}_0^{ag} = \boldsymbol{x}_0$, we have $f(\boldsymbol{x}_0) = f(\boldsymbol{x}_1^{md})$. Thus,

$$
\triangle_t \leq \triangle_1^{md} + 1. \quad (99)
$$

Since (94) holds for all $l \in [t+1]$, we have that

$$
\left\| \boldsymbol{x}_{t+1}^{md} - \boldsymbol{x}_t \right\| \leq \min\{1/L_1, 1/L_2\}.
$$

Therefore, applying Lemma C.2 again, we obtain that

$$
\begin{aligned}
f(\boldsymbol{x}_{t+1}^{md}) \leq & f(\boldsymbol{x}_t) + \left\langle \nabla f(\boldsymbol{x}_t), \boldsymbol{x}_{t+1}^{md} - \boldsymbol{x}_t \right\rangle + \frac{L_0 + L_1\|\nabla f(\boldsymbol{x}_t)\|^p + L_2\triangle_t^q}{2}\left\| \boldsymbol{x}_{t+1}^{md} - \boldsymbol{x}_t \right\|^2 \\
\leq & f(\boldsymbol{x}_t) + \|\nabla f(\boldsymbol{x}_t)\| \cdot \left\| \boldsymbol{x}_{t+1}^{md} - \boldsymbol{x}_t \right\| + \frac{L_0 + L_1\|\nabla f(\boldsymbol{x}_t)\|^p + L_2\triangle_t^q}{2}\left\| \boldsymbol{x}_{t+1}^{md} - \boldsymbol{x}_t \right\|^2,
\end{aligned}
$$

where the second inequality follows from Cauchy-Schwarz inequality. Subtracting $f^*$ from both sides and combining with (99), we have

$$
\begin{aligned}
& \triangle_{t+1}^{md} \\
\leq & \triangle_1^{md} + 1 + \frac{1}{L_1 + L_2}\sqrt{g(1 + \triangle_1^{md})} + \frac{L_0 + L_1\left( g(1 + \triangle_1^{md}) \right)^{\frac{p}{2}} + L_2\left( 1 + \triangle_1^{md} \right)^q}{2\left( L_1 + L_2 \right)^2} \leq \mathcal{F}_4.
\end{aligned}
$$
$$
(100)
$$

Now we finish the induction and obtain the desired result.

$\square$

*Proof of Theorem 4.* From Proposition G.3, we have that with probability at least $1-2\delta$, $\triangle_t^{md} \leq \mathcal{F}_4$ for all $t \in [T]$. Thus, (98) holds when $t = T$, i.e.,

$$\frac{\eta}{2} \sum_{l=1}^{T} \left\| \nabla f(\boldsymbol{x}_l^{md}) \right\|^2 \leq 1 + \triangle_1^{md}. \tag{101}$$

Dividing $T\eta/2$ on both sides and combining with the constraints of $\eta$, we get the desired results. $\quad\square$

## H    OMITTED PROOF

*Proof of Lemma 4.1.* To start with, we will prove the first line by induction. It is obvious that the inequality holds for $B_0 = 0$. Suppose that for some $0 \leq t \leq T$, we have

$$\frac{1}{4}k^2 \leq B_k \leq k^2, \forall k \in [t].$$

Then, we have

$$B_{t+1} \leq t^2 + \frac{1}{2}\left(1 + \sqrt{4t^2 + 1}\right) \leq t^2 + \frac{1}{2}\left(1 + 2t + 1\right) \leq (t+1)^2,$$

and

$$B_{t+1} \geq \frac{1}{4}t^2 + \frac{1}{2}\left(1 + \sqrt{t^2 + 1}\right) \geq \frac{1}{4}\left(t+1\right)^2.$$

Therefore, we finish the proof for $\frac{1}{4}t^2 \leq B_t \leq t^2, \forall t \in [T]$. For the second conclusion in Lemma 4.1,

$$\begin{aligned}
(A_t - A_{t-1})^2 = (B_t - B_{t-1})^2 &= \frac{1}{4}\left(1 + 2\sqrt{4B_{t-1} + 1} + 4B_{t-1} + 1\right) \\
&= B_{t-1} + \frac{1}{2}\left(1 + \sqrt{4B_{t-1} + 1}\right) \\
&= B_t.
\end{aligned}$$

Since $B_t \geq \frac{1}{4}t^2, \forall t \in [T]$, we have $B_t \geq 0, \forall t \in [T]$. Therefore,

$$A_t - A_{t-1} = B_t - B_{t-1} = \frac{1}{2} + \frac{1}{2}\sqrt{4B_{t-1} + 1} \geq 1.$$

Now we finish the proof for all the inequalities. $\quad\square$

