# OpenReview forum: "Convergence Analysis of Nesterov's Accelerated Gradient Descent under Relaxed Assumptions"
_ICLR.cc/2026/Conference — Submitted to ICLR 2026_

### Official Review · Reviewer_rtaj · 2025-10-28

**Soundness:** 3
**Presentation:** 2
**Contribution:** 2
**Rating:** 6
**Confidence:** 4

**Summary:**

This paper provides an analysis of Nesterov's accelerated gradient descent method for stochastic first-order optimization with weakened assumptions on both the smoothness of the objective and the bounds on the noise in the gradient. In my opinion, the paper makes a nice theoretical contribution.

**Strengths:**

The problem addressed is interesting, especially since Nesterov's method is widely used in machine learning. The technical assumptions are explained fairly well and the relationship with the existing literature is well-documented. The technical arguments appear correct to me as far as I can tell.

**Weaknesses:**

There are a few typos and issues with the presentation which I have also detailed below. These issues can be easily addressed, I think.

The work is quite technical and could be perceived as an incremental improvement over existing analyses. In my opinion, the paper would benefit if the authors would better describe the novel technical ideas which distinguish their approach from existing analyses. For example, could this be explained in the introduction, in addition to stating the new results?

Numerical experiments are missing.

**Questions:**

I think there is a typo in equation (2) (should the RHS be multiplied by |x-y|?)

In points (c) and (d) in the introduction, what are A,B, and C. It appears to me that only B has been defined in equation (4). I see these parameters have been introduced later in Assumptions 3 and 4, but I think it would made more sense to state it earlier in the paper.

I'm curious what results can be obtained in the strongly convex case. Do you have any ideas?

---

> ### Author Response · Authors · 2025-11-25
> **Response to Reviewer rtaj**
>
> # Response to Reviewer rtaj
>
> We thank you for your valuable comments which have greatly improved our presentations.
> We revised accordingly in the revision version.
>
> > There are a few typos and issues with the presentation which I have also detailed below. These issues can be easily addressed, I think.
>
> We have revised the manuscript accordingly and have also corrected some other typos. We will perform a careful final proofread before submitting the final version.
>
> > The work is quite technical and could be perceived as an incremental improvement over existing analyses. In my opinion, the paper would benefit if the authors would better describe the novel technical ideas which distinguish their approach from existing analyses. For example, could this be explained in the introduction, in addition to stating the new results?
>
> We present the key novelties of our proof in the proof sketch (Section 6, especially Section 6.3). We agree that a brief summary in the introduction would be beneficial and will incorporate one into the final version based on your suggestion.
>
> > Numerical experiments are missing
>
> We added some simple numerical results which show some promising results of stochastic NAG comparing with SGD on phase retrieval  trainings. See Appendix I. We will consider to add more numerical results in the final version.
>
> > I think there is a typo in equation (2) (should the RHS be multiplied by $|x-y|$?
>
> This is a typo. We revised accordingly.
>
> > In points (c) and (d) in the introduction, what are A,B, and C. It appears to me that only B has been defined in equation (4). I see these parameters have been introduced later in Assumptions 3 and 4, but I think it would made more sense to state it earlier in the paper.
>
> We agree that clarifying these parameters earlier would improve the presentation. For the purpose of the discussions and to maintain consistent reference to Assumptions 3 and 4 throughout the rebuttal period, we have kept the formal definitions in their current location. In the final version, we will move Assumptions 3 and 4 into the introduction section.
>
> > I'm curious what results can be obtained in the strongly convex case. Do you have any ideas?
>
> That's a very interesting direction we hadn't fully considered. Off the top of our head, strong convexity should improve the convergence rate. It could be challenging to estimate the function values in the strongly convex case. We willl  look into this.

---

> ### Author Response · Authors · 2025-12-02
> **Summary: Major Weaknesses from Reviewer rtaj and Responses**
>
> **W1.  a few typos and issues with the presentation**\
> **R1.**  We corrected the typos accordingly
>
> **W2.  the paper would benefit if the authors would better describe the novel technical ideas**\
> **R2.**  We revised this presentation issue accordingly.
>
> **W3. Numerical experiments are missing.**\
> **R3.** We added some simple numerical results.

---

### Official Review · Reviewer_bvGq · 2025-10-29

**Soundness:** 3
**Presentation:** 2
**Contribution:** 3
**Rating:** 4
**Confidence:** 3

**Summary:**

The authors study the convergence speed of the Nesterov accelerated gradient (NAG) algorithm applied to convex functions, that are not assumed to be L-smooth as it is standard in many former analysis, but instead verify a generalized smoothness assumption. The authors also study the convergence of a stochastic version of this algorithm, under "relaxed-affine variance"-like assumptions.

**Strengths:**

The optimal $1/T^2$ rate of NAG for smooth convex functions is an important result in optimization. As recent works suggest generalized smoothness assumption fits some non convex optimization problems (e.g. neural network) better that the classical L smooth assumption, it is indeed interesting to see that NAG preserves its acceleration property over gradient descent. Also, the noise assumption on the gradient is more general that many others considered in other works (bounded variance, strong growth condition, etc.). In this regard, this work is a nice generalization of existing works in the deterministic and stochastic cases.

**Weaknesses:**

- The authors claim that their first contribution (contribution (a)) is to provide "a more general and realistic smoothness condition".
If it is obvious that it is more general, the fact that is it more realistic than former concept of generalized smoothness [Zhang et al.
2020] seems not justified in the paper. If this claim remains in the paper, I suggest to clarify why and in which context Definition 1 is more realistic.

- Under Assumption 4 with $A = C = 0$ (strong growth condition), it is known in the smooth convex setting that we can obtain a $O(1/T^2)$ rate, without the additional $O(1/\sqrt{T})$ term. In your result, specifically in Theorem 3, the bound does not reduces to $O(1/T^2)$ in this setting, making the result sub optimal in this specific case. Is it a limit of your analysis ? I think it is not a major concern, but I suggest to at least briefly discuss it, e.g. under Theorem 3.

- In the end of Section 4, "our smoothness assumption could be more general". I think it is important to clarify this statement, specifically what is meant by "could". As written, it is thus unclear how it extends existing works.

- On page 1: "Furthermore, this complexity bound is known to be optimal for large enough dimension d,
as shown by (Nemirovskij, Yudin, 1983), without further assumptions." I think the sentence could be misleading. Please precise it is optimal among gradient based algorithms.

- I believe there is a major typo in (2), where a $|| x-y ||$ factor is missing. This typo affects the further discussion.

- In section 1, "As usual, one typically focuses on the function value gap for convex objectives and the squared gradient norm for non-convex ones." I know the authors want to stay evasive as it is an introduction, but the second part of the sentence is a bit too evasive to me. There is a whole litterature about minimization of non-convex function such that the focus remains the function value gap, e.g. for Polyak Lojasiewicz functions [Karimi et al.], (strongly) quasar convex functions [Hinder et al.], (strongly) quasiconvex functions [Grad et al.]. I suggest to clarify a bit this statement.


Minor points and typos.

- P.5 219-220 ", while for the stochastic case in Section 4.2." seems grammatically not correct.

- On page 2: "Assumption 2 is commonly used in the analysis for stochastic optimizations.". I also suggest to mention that it is a relevant assumption to study many practical setting, which is even better than simply being commonly used in former analysis.

- In section 6 about "$\mathcal{F}_2$", please avoid to use notations in the main text without at least referring the equation which defines it.

References:

-Hamed Karimi, Julie Nutini, Mark Schmidt, "Linear Convergence of Gradient and Proximal-Gradient Methods Under the Polyak-Łojasiewicz Condition", 2016

-Oliver Hinder, Aaron Sidford, Nimit S. Sohoni, "Near-Optimal Methods for Minimizing Star-Convex Functions and Beyond", 2019

-Sorin-Mihai Grad,  Felipe Lara Raul T. Marcavillaca,  "Strongly quasiconvex functions: what we know (so far)", 2024

**Questions:**

- Last sentence of Section 2 "(Wang et al., 2023) gave a counter-example showing the necessity of prior knowledge on problem parameters for step sizes under (L0, L1 )-smoothness.". Do you mean that adaptive method to compute parameters are not suited to generalized smoothness assumptions ? The sentence seems a bit evasive to me.

- On page 2: "Obviously, Definition 1 covers a broader range of relaxed smoothness than the generalized smoothness". I think it is important to at least refer here to your Appendix A, currently not referred in your main text. Could you please give some intuition about which kind of functions verify your condition and will not verify the former condition of Zhang et al. (2020)? Is the function introduced in Appendix A such an example ?

- Could you please give a bit of intuition about how much Assumption 3 is stronger than Assumption 4 ? For example, could you provide an example of a setting where Assumption 4 is verified and not Assumption 3 ?

- Could you justify why you call Assumption 4 "Relaxed affine variance-noise", while it is introduced as "Expected smoothness" in (Khaled & Richtarik, 2023)?

- By curiosity, replacing Assumption 3 by Assumption 4 in Theorem 2, can you obtain a bound in expectation trivially with your current analysis ?


- The work [Hermant et al.] shows that a stochastic version of NAG for convex and L-smooth functions, under the strong growth condition ($A=C=0$ in your Assumption 4), achieves a $o(1/T^2)$ rate almost surely. Do you think such almost sure rate could be deduce in your setting ?

- As an opening question, [Guillet et al.] shows that the L-smooth assumption is, compared to other assumption, not "stable". Do you think generalized smoothness assumptions such as yours avoid this problem ?

References:

-Ahmed Khaled, Peter Richtárik, "Better Theory for SGD in the Nonconvex World", 2020

-Julien Hermant, Marien Renaud, Jean-François Aujol, Charles Dossal, Aude Rondepierre, "Gradient correlatio is a key ingredient to accelerate SGD with momentum", 2025

-Charles Guille-Escuret, Baptiste Goujaud, Manuela Girotti, Ioannis Mitliagkas, "A Study of Condition Numbers for First-Order Optimization", 2020

---

> ### Author Response · Authors · 2025-11-25
>
> # Response to Reviewer bvGp (1/3)
> We thank you for your valuable comments which have greatly improved our presentations. We revised accordingly in the revision version.
> ## Weakness
>  > The authors claim that their first contribution (contribution (a)) is to provide "a more general and realistic smoothness condition". If it is obvious that it is more general, the fact that is it more realistic than former concept of generalized smoothness [Zhang et al. 2020] seems not justified in the paper. If this claim remains in the paper, I suggest to clarify why and in which context Definition 1 is more realistic.
>
> We revised our statement to ''a more general smoothness condition" to more accurately reflect its scope. Our work is situated between two related notions: $(L_0,L_1,0)$-smoothness, which is empirically known [Zhang et al. 2020] for neural networks training and is theoretically proved for phase retrieval in the appendix, and $(L_0,0,L_2)$-smoothness, which is theoretically proven for specific shallow neural networks in the appendix. To bridge this divide, we propose the $(L_0,L_1,L_2)$-smoothness condition, which subsumes both and provides a unified framework for analysis.
>
> > Under Assumption 4 with  $A=C=0$ (strong growth condition), it is known in the smooth convex setting that we can obtain a rate $O(1/T^2)$, without the additional term $O(1/\sqrt{T})$. In your result, specifically in Theorem 3, the bound does not reduces to $O(1/T^2)$ in this setting, making the result sub optimal in this specific case. Is it a limit of your analysis? I think it is not a major concern, but I suggest to at least briefly discuss it, e.g. under Theorem 3.
>
> Thank you for your suggestion, and your raising interesting point. We added a comparison that _''Under Assumption 4 with $A=C=0$ and standard smoothness,  Hermant et al. [2025] showed almost sure convergence rates of $o\left((B+1)/T^2\right)$ for stochastic NAG in general convex optimization problems. In comparisons, the noise assumption in the above theorem is more general, but the rate has an extra suboptimal term $\mathcal{O}(\sqrt{B/T})$ with respect to $B$."_ with the related work under Theorem 3. We think it should be due to a limit of our analysis.
>
> > In the end of Section 4, "our smoothness assumption could be more general". I think it is important to clarify this statement, specifically what is meant by "could". As written, it is thus unclear how it extends existing works.
>
> We made the comparison clearer and changed the related statements in the revision as follows:
>
> _''which matches the rate for the deterministic NAG in [Li et al., 2024] under a generalized $(L_0,L_1,0)$-smoothness: $\|\nabla^2 f\left(x\right)\|\leq l\left(\|\nabla f\left(x\right)\|\right)$ with a sub-quadratic non-decreasing positive function $l$ up to logarithm factors. Note that Li et al. [2024] did not provide the analysis for stochastic NAG and we consider the  $(L_0,L_1,L_2)$-smoothness. ''_
>
> > On page 1: "Furthermore, this complexity bound is known to be optimal for large enough dimension d, as shown by (Nemirovskij, Yudin, 1983), without further assumptions." I think the sentence could be misleading. Please precise it is optimal among gradient based algorithms.
>
> We changed the statement as ''this complexity bound is known to be optimal among gradient based algorithms".
>
> > I believe there is a major typo in (2), where a factor $\|\|x-y\|\|$ is missing. This typo affects the further discussion.
>
> This is a typo, and we revised accordingly.
>
> > In section 1, "As usual, one typically focuses on the function value gap for convex objectives and the squared gradient norm for non-convex ones." I know the authors want to stay evasive as it is an introduction, but the second part of the sentence is a bit too evasive to me. There is a whole litterature about minimization of non-convex function such that the focus remains the function value gap, e.g. for Polyak Lojasiewicz functions [Karimi et al.], (strongly) quasar convex functions [Hinder et al.], (strongly) quasiconvex functions [Grad et al.]. I suggest to clarify a bit this statement.
>
> We revised accordingly. We changed the statement as _''As usual, one typically focuses on the function value gap forconvex objectives and the squared gradient norm for general non-convex ones"_ and we also added a footnote that _''An extensive literature on minimizing structured non-convex functions focuses on the function value gap. Examples include work on Polyak-Łojasiewicz functions [Karimi et al., 2016], (strongly) quasar-convex functions [Hinder et al., 2020] and (strongly) quasiconvex functions [Grad et al., 2025]. This is beyond the discussion of this paper."_ on Page 1.

---

> ### Author Response · Authors · 2025-11-25
>
> # Response to Reviewer bvGp (2/3)
> ## Minor points and typos
>  > P.5 219-220 ", while for the stochastic case in Section 4.2." seems grammatically not correct.
>
> We changed ''while'' as ''and'' in the revision.
>
> > On page 2: "Assumption 2 is commonly used in the analysis for stochastic optimizations.". I also suggest to mention that it is a relevant assumption to study many practical setting, which is even better than simply being commonly used in former analysis.
>
> We changed the statement as _''Assumption 2 is a relevant assumption for studying many practical settings and is also commonly used in the analysis of stochastic optimization.''_ in the revision.
>
> > In section 6 about "$\mathcal{F}_2$", please avoid to use notations in the main text without at least referring the equation which defines it.
>
> We added _''where $\mathcal{F}_2$ is a positive constant given explicitly in ...''_ in the revision. See also the other places for this similar issue.
>
> ## Questions
> > Last sentence of Section 2 "(Wang et al., 2023) gave a counter-example showing the necessity of prior knowledge on problem parameters for step sizes under $(L_0, L_1)$-smoothness.". Do you mean that adaptive method to compute parameters are not suited to generalized smoothness assumptions ? The sentence seems a bit evasive to me.
>
> Thank you so much. We changed the statement as _''Wang et al. [2023] gave a counter-example showing the necessity of prior knowledge on problem parameters for learning rates in AdaGrad under $\left(L_0,L_1\right)$-smoothness."_ in the revision.
>
> > On page 2: "Obviously, Definition 1 covers a broader range of relaxed smoothness than the generalized smoothness". I think it is important to at least refer here to your Appendix A, currently not referred in your main text. Could you please give some intuition about which kind of functions verify your condition and will not verify the former condition of Zhang et al. (2020)? Is the function introduced in Appendix A such an example?
>
> We added some comments that _''Particularly, it is situated between two related notions: $(L_0,L_1,0)$-smoothness, which is empirically verified but theoretically unproven [Zhang et al. 2020], and $(L_0,0,L_2)$-smoothness, which has been theoretically proven for specific shallow neural networks [Taheri and Thrampoulidis, 2023]	and phase retrieval problems in the appendix but may not empirically verified."_ after Definition 1 in the revision. We do not have any intuition for which kind of functions verify this condition, and we agree with that investigating more on this aspect is interesting.

---

> ### Author Response · Authors · 2025-11-25
>
> # Response to Reviewer bvGp (3/3)
> ## Questions
> > Could you please give a bit of intuition about how much Assumption 3 is stronger than Assumption 4 ? For example, could you provide an example of a setting where Assumption 4 is verified and not Assumption 3 ?
>
> We added the intuition comment _''as it controls all moments of the noise distribution, while Assumption 4 only controls its second moment (the variance)"_ after Assumption 4 in the revision. One typical example is the  $t(3)$ distribution gradient noise, which is known to has a well-defined, finite variance and it is not sub-Gaussian.
>
> > Could you justify why you call Assumption 4 "Relaxed affine variance-noise", while it is introduced as "Expected smoothness" in (Khaled \& Richtarik, 2023)?
>
> We chose "Relaxed Affine Variance" to emphasize the specific structure of the assumption: it models the variance of the stochastic gradient as being bounded by an affine (linear) function of the full gradient norm (as well as the function values), which is a relaxation of the simpler bounded variance assumption. Khaled and Richtarik [2023] essentially used the equivalent form of
> $\mathbb{E}_z [\| \nabla f(x;z)\|^2] \leq A \left(f(x)-f^*\right)+\tilde{B} \|\nabla f \left(x\right)\|^2+C$
> which can be closely related to the smoothness and interpolating regimes, and thus they named it expected smoothness. To prevent confusion, we added the sentence _''Assumption 4 was initially proposed by [Khaled and Richtarik, 2023] under the name expected smoothness. Its original, equivalent form is:_
>
> $\mathbb{E}_{z}\left[\|\nabla f_z(x;z)\|^2\right]\leq A\left(f(x)-f^*\right)+B'\|\nabla f\left(x\right)\|^2+C, \forall x\in\mathbb{R}^d$."
> after Assumption 4 in the revision.
>
> > By curiosity, replacing Assumption 3 by Assumption 4 in Theorem 2, can you obtain a bound in expectation trivially with your current analysis?
>
> Thank you for this insightful question. You are correct to point this out. Replacing Assumption 3 with Assumption 4 in Theorem 2 does not, in fact, lead trivially to a bound in expectation with our current analysis. One of  the technical reasons is that, under Assumption 4 and generalized smoothness, the second order term raised from the descent lemma involves higher order of function values which seems hard to controlled. We have added the sentence _"Our analysis for the above theorem, which relies on Assumption 3, does not apply under the weaker noise condition of Assumption 4 in the generalized smoothness."_ to the discussion under Theorem 2.
>
> > The work [Hermant et al.] shows that a stochastic version of NAG for convex and $L$-smooth functions, under the strong growth condition ( $A=C=0$ in your Assumption 4), achieves a  $o(1/T^2)$ rate almost surely. Do you think such almost sure rate could be deduce in your setting?
>
> This an interesting question and we do not have any clear answer. We will consider it in the final version. It seems that our analysis can not deduce this almost sure fast rate $o(1/T^2)$ under the strong growth condition even in the standard smoothness. We added some related comments _''Under Assumption 4 with $A=C=0$ and standard smoothness,  Hermant et al. [2025] showed almost sure convergence rates of $o\left((B+1)/T^2\right)$ for stochastic NAG in general convex optimization problems. In comparisons, the noise assumption in the above theorem is more general, but the rate has an extra suboptimal term $\mathcal{O}(\sqrt{B/T})$ with respect to $B$."_ under Theorem 3.
>
> > As an opening question, [Guillet et al.] shows that the $L$-smooth assumption is, compared to other assumption, not "stable". Do you think generalized smoothness assumptions such as yours avoid this problem ?
>
> Thank you for this valuable question. We agree that fully addressing it requires further research, and we see it as a compelling avenue for future work. We have added  related comments _''Via a notion of continuity, Guille-Escuret et al. [2021] demonstrated that the strong convexity and smoothness have a weakness resulting in a lack of robustness for tuning first-order algorithms, and they presented promising alternatives.''_ in the discussion section.

---

> > ### Comment · Reviewer_bvGq · 2025-11-26
> >
> > Thank you for the detailed response and for addressing my concerns, which were mostly about presentation and clarification. While I believe the motivation and implications of the specific relaxation $(L_0,L_1,L_2)$-smoothness could be further developed to fully highlight its relevance, I find the overall contribution valuable. I thus increase my score.
> > As a last suggestion (I will not lower my score if you don't follow it), please try to refer to your Appendix in the main text. In particular, I believe your non-convex convergence result of Appendix G is not mentionned in the main text, which seems somewhat unfortunate to me.

---

> > > ### Author Response · Authors · 2025-11-26
> > > **We truly appreciate your helpful comments!**
> > >
> > > We thank you again for your helpful comments, which means a lot for us. These improve the presentation and clarity. We acknowledge some of the presentations in the original version are not clear or precise. We also thank you for your encouraging comments and mentioning the existing almost sure convergence rate with better dependency on $B$, which inspires us to improve the bounds.   After the submission of the revision, we are also aware that we did not mention the new added contents (including the comparisons tables) in the main text. We will definitely added some statements mentioning these in the final version.  Thanks!

---

> ### Author Response · Authors · 2025-12-03
> **Summary: Weakness from Reviewer bvGq and Our Response, Discussions**
>
> ## Weakness from Reviewer bvGq and Our Response
> **W1. If this claim remains, I suggest to clarify...**\
> **W2. I suggest to at least briefly discuss it...**\
> **W3. it is important to clarify this statement**\
> **W4. Please precise it is optimal among gradient based algorithms.**\
> **W5.  there is a major typo**\
> **W6. I suggest to clarify a bit...**
>
> **R1-6**. We revised these presentation issues accordingly.
>
> ## Final comments (26 Nov 2025, 18:37) from Reviewer bvGq
> Thank you for the detailed response and for addressing my concerns, which were **mostly about presentation and clarification**.  **the overall contribution valuable. I thus increase my score.**

---

### Official Review · Reviewer_t1mr · 2025-10-29

**Soundness:** 3
**Presentation:** 3
**Contribution:** 3
**Rating:** 6
**Confidence:** 4

**Summary:**

This paper is concerned with the convergence analysis of Nesterov Accelerated Gradient Descent  (NAG) both in deterministic and stochastic settings, under a convexity assumption. The main contribution of the paper is to relax the classical assumptions made on the function to minimize. The classical assumtion is that the function is L-smooth (i.e. that its gradient is L-Lipshitz). Howerver, such an assumption is not satisfied in many machine learning problems. The author thus replace it with a more general (and also technical one).
Nevertheless, the authors are able to show that they get the same type of speed of convergence as in the classical case.

**Strengths:**

* Minimisation of large dimension functions is a bottlneck in machine learning at the moment. The main tools used to tackle it are first order methods, and Nesterov acceleration is therefore one of the most promising venue. The main drawback of NAG is its setting (assumtions on the function to minimize) that is too strong for machine learning. Relaxing this setting is therefore major issue at the moment within the optimization/machine learning community.

* Both detremintic case and stochastic cases are considered. The deterministic setting is easier to understand, and it helps understanding the stochastic one (which is the one used in practive for machine learning problems).

* Most of the obtained convergence results are optimal.

**Weaknesses:**

* Relaxing the L-smoothness assumption is indeed a good point. However, the main limitation when using NAG is the convexity assumption. This convexity assumption is of course not satisfied by machine learning problems. As a consequence, this limits the interest of the work, since even by relaxing the L-smoothness assumption, we are still far from the setting that happens in practice for machine learning problems.

* The proofs are technical and sometimes hard to follow. In the deterministic case, would it be possible to derive a continuous analysis (with some ODEs) ? see e.g. the paper by Su, Boyd and Candes
https://arxiv.org/abs/1503.01243

* The authors have made a very nice bibliography work. However, they could also refer to an ICLR 2025 paper :
Gradient Correlation is a key ingredient to accelerate SGD with momentum, by Hermant et al
https://arxiv.org/abs/2410.07870

**Questions:**

* There are several updates rule for NAG (see e.g. the last reference of the previous section).
Are the results of the paper also true for those other update rules, or are the proofs specific to the choice made in the paper ?

* The authors quickly explain the difference of their proof with respect to the classical assumption of L-smoothness in section 6.3
I followed the proof in the appendix, but I could not get whether the new proof can be split into 2 parts (or if they are mixed):
a first part to get some control of the gradient thanks to the new assumption, and then a second part which would basically be the classical proof once the estimate of the gradient has been obtained.
If so, this would give a framework to relax even further the L-smoothness assumption.

* I have some difficulties understanding the class of functions that satisfy the new smoothness assumption (which is much more involved than the classical L-smoothness assumption). I found the example in the appendix a good idea to understand further what it means. However, are there some more intuitive general properties that are satisfied by such functions ?
This would help understanding how much the set of functions has been extended thanks to the relaxation of the smoothness property.

* Any hint on how to get rid of the convexity assumption ?

---

> ### Author Response · Authors · 2025-11-25
> **Response to Reviewer t1mr (1/2)**
>
> # Response to Reviewer t1mr (1/2)
> We thank you for your valuable comments which have greatly improved our presentations. We revised accordingly in the revision version.
>
> >  Relaxing the $L$-smoothness assumption is indeed a good point. However, the main limitation when using NAG is the convexity assumption. This convexity assumption is of course not satisfied by machine learning problems. As a consequence, this limits the interest of the work, since even by relaxing the $L$-smoothness assumption, we are still far from the setting that happens in practice for machine learning problems.
>
> We __drop the convexity assumption__ by adding the convergence analysis with respect to the stationary points under the new introduced generalized $(L_0,L_1,L_2)$-smoothness assumption for the non-convex stochastic optimizations __in Appendix G__.
>
> Our new introduced $(L_0,L_1,L_2)$-smoothness is situated between two related notions: $(L_0,L_1,0)$-smoothness, which is empirically known [Zhang et al., 2020] for neural networks training (but without theoretically proven) and is theoretically proved for phase retrieval in the appendix, and $(L_0,0,L_2)$-smoothness, which is theoretically proven for specific shallow neural networks in the appendix.
>
> We also added two numerical experiments on neural networks training and phase retrieval on synthesis datasets __in Appendix I__, showing that stochastic NAG converges faster in these two problem compared to SGD. We do not know why this happens, and we thus start with the simple convex case and our theoretical results indeed show faster convergence rate of  stochastic NAG over SGD, under the $(L_0,L_1,L_2)$-smoothness assumption.
>
> Motivated by our numerical experiments, we think that neural networks and phase retrieval problems may have some nice properties such as quasi-convexity, but examining these properties are too difficult for us and would be left as a future work. In fact, there are other kinds of work showing that for phase retrieval problems with structured random sampling, the object function has been shown to be (quasi)-convex provided the sample size is large enough.
>
> As highlighted in both the introduction and the main theorem section, proposing the new $(L_0,L_1,L_2)$-smoothness condition is only one of our contributions. Our second contribution is establishing the fast convergence rate for stochastic NAG under $(L_0,L_1,0)$-smoothness. This has been an open problem since [Li et al., 2024], who provided a fast rate only for deterministic NAG and a standard rate for stochastic gradient descent. We believe a proof for stochastic NAG under this condition presents challenges; in particular, the analysis for deterministic NAG by [Li et al., 2024] does not appear to be trivially extendable.
>
> Our third contribution is to provide fast convergence rates for stochastic NAG in standard smooth stochastic optimization under the more general expected relaxed affine-variance noise [Khaled and Richt´arik, 2023], which has been shown to encompass many practical noise settings, and such a result is new for stochastic NAG.
>
> To highlight our theoretical contributions, we added two tables __in Appendix A__.
>
> > The proofs are technical and sometimes hard to follow. In the deterministic case, would it be possible to derive a continuous analysis (with some ODEs) ? see e.g. the paper by Su, Boyd and Candes https://arxiv.org/abs/1503.01243
>
> We have tried to revised Section 6 to improve its clarity and flow.
>
> Thank you for mentioning this interesting paper. We believe that their proof approach is quite different from ours in some aspects and we leave it as a future work. We also added some related comments ''Su et al. (2016) introduced a second-order ODE and accompanying tools for characterizing Nesterov’s accelerated gradient method." in the revision.
>
> > The authors have made a very nice bibliography work. However, they could also refer to an ICLR 2025 paper : Gradient Correlation is a key ingredient to accelerate SGD with momentum, by Hermant et al https://arxiv.org/abs/2410.07870
>
> Thank you for sharing this interesting paper. We added some related comment ''[Hermant et al., 2025] showed the expected convergence rate of $\mathcal{O}\left(\left(B+1\right)/T^2\right)$ and almost-sure rate of $o\left(\left(B+1\right)/T^2\right)$ for ACDM in general convex optimization problems, and they derived fast convergence rates for ACDM in strongly convex optimization problems." on Page 4. We also made some comparisons.

---

> > ### Comment · Reviewer_t1mr · 2025-11-27
> >
> > I would like to thank the authors for their detailed answer. I have the feeling that most of my concerns have been addressed.
> > I really like the new convergence result in the non convex case.
> > This paper would be a nice contribution to ICLR 2026.
> > This is probably out of the scope of this paper, but it would be nice to explain why these new regularity assumptions are a way to get rid for the convexity assumption.
> > I will raise my score accordingly.

---

> > > ### Author Response · Authors · 2025-11-28
> > > **We truly appreciate your helpful comments!**
> > >
> > > We thank you again for your valuable time on the review and participation in the discussion. We thank you very much for your encouraging comments, particularly the suggestion on the potential future research topics. We will try to add more comments and explore more per the new assumptions in the final revision. Thanks!

---

> ### Author Response · Authors · 2025-11-25
> **Response to Reviewer t1mr (2/2)**
>
> # Response to Reviewer t1mr (2/2)
>
> > There are several updates rule for NAG (see e.g. the last reference of the previous section). Are the results of the paper also true for those other update rules, or are the proofs specific to the choice made in the paper?
>
> Thanks a lot for this insightful question. Algorithm 2 with $\beta=1$ In [Hermant et al., 2025] is the same as Algorithm 1 in our paper.
>
>
> > The authors quickly explain the difference of their proof with respect to the classical assumption of L-smoothness in section 6.3 I followed the proof in the appendix, but I could not get whether the new proof can be split into 2 parts (or if they are mixed): a first part to get some control of the gradient thanks to the new assumption, and then a second part which would basically be the classical proof once the estimate of the gradient has been obtained. If so, this would give a framework to relax even further the L-smoothness assumption.
>
> The noise variance is upper bounded by the function value gap and the gradient norm, and the latter one could be controlled by the former one. Thus, to bound the norm of the noise, we first obtain the expected bound of $f(x_t^{md})-f^*$ for all $t\in[T]$.
>
> This approach does not work for generalized smooth functions with Assumption 4. One of the technical reasons is that, under Assumption 4 and generalized smoothness, the second order term raised from the descent lemma involves higher order of the function values which seems hard to control.
>
> > I have some difficulties understanding the class of functions that satisfy the new smoothness assumption (which is much more involved than the classical $L$-smoothness assumption). I found the example in the appendix a good idea to understand further what it means. However, are there some more intuitive general properties that are satisfied by such functions ? This would help understanding how much the set of functions has been extended thanks to the relaxation of the smoothness property.
>
> Thanks for your valuable question. We believe that it is meaningful to analyze other properties related to the novel smoothness assumptions and we view it as an interesting future work.
>
> > Any hint on how to get rid of the convexity assumption?
>
> We __drop the convexity assumption__ by adding the convergence analysis with respect to the stationary points under the new introduced generalized $(L_0,L_1,L_2)$-smoothness assumption for the non-convex stochastic optimizations __in Appendix G__.

---

> ### Author Response · Authors · 2025-12-03
> **Summary: Weakness from Reviewer t1mr and Our Response, Discussions**
>
> ## Weakness from Reviewer t1mr and Our Response
> **W1. The limitation is the convexity assumption**\
> **R1**.  Refer to [**R1 for Reviewer Rmjo**].
>
> **W2. The proofs are technical and sometimes hard to follow**\
> **A2.** We revised the proof (Section E) to improve the presentation.
>
> **W3. refer to an ICLR 2025 paper**\
> **A3.** We made some comments and comparisons with this paper. (Lines 168-170; Appendix A)
>
> ## Final Comments (28 Nov 2025, 01:38) from Reviewer t1mr
>
>  **most of my concerns have been addressed. I like the result in the non convex case. This paper would be a nice contribution. I will raise my score.**

---

### Official Review · Reviewer_Rmjo · 2025-10-31

**Soundness:** 3
**Presentation:** 3
**Contribution:** 2
**Rating:** 2
**Confidence:** 4

**Summary:**

This paper present a new convergence proof for the Nesterov accelerated algorithm (NAG) for convex function in both deterministic and stochastic setting under relaxed assumption. In particular, a new relaxed assumption of the Lipschitz smoothness of the fonction is introduced and acceleration is proved under this new assumption. The work focuses on the convergence of NAG.

**Strengths:**

- A new convergence result of NAG in both deterministic and stochastic setting is proved.
- The proofs seem correct.

**Weaknesses:**

Major Weaknesses:
- The assumption of this work are far from real-world machine learning applications. First, the fonction needs to be convex which is not verified for neural network training. Then, the relaxed assumption introduced in this work is verified by a very particular neural network detailed in Appendix A. However, it is not clear why this "relaxed assumption" is interesting for real-world applications. This "relaxed assumption" of smoothness is only motivated by the fact that it is "relaxed".
- No experiments or real-world applications of the convergence results are provided reducing strongly the impact of the work.
- The paper is not pedagogical. In particular, I appreciate the effort to give insides on the proof of Theorem 2 in Section 2. However, Section 6 is very technical and unclear for me without looking at the detailed proof in Appendix. I suggest to remove it of the main paper.

Minor Weaknesses:
- In the introduction [e.g. in Contributions (c), (d) or in line 166], you mention $A$, $B$, $C$ the constants involves in Assumption 4 but without introduced it before. I suggest to move Assumption 4 around equation (4) in order to state notations before using them.
- Some notations are confusing. In Algorithm 1, you use $x_t$, $x_t^{md}$ and $x_t^{ag}$ without introducing what denote "md" and "ag". I will suggest to use different letters for each of these intermediary sequence, e.g. $x,y,z$. Moreover, I suggest to state explicitly that $x_t^{ag}$ is the output of Algorithm 1.

Typos/imprecisons:
- Definition 1: because $\|x-y\| \le \min\{\frac{1}{L_1}, \frac{1}{L_2}\}$, you need that $L_1, L_2 > 0$ not only $L_1, L_2 \ge 0$. Or could you precise that you mean by $1/0 = +\infty$.
- Line 351 : What is $\mathcal{F}_2$ ? Could you add the definition in section 6 ?
- Equation (19): It seems that $\tilde{d} = d\times m$, could you state it for clarity ? The weights $a_j$ seem to not be trainable (it is the setting of [1]), could you state it clearly ?
- Line 593. In your notation, according to [1], it should be $w_i$ instead of $x_i$ in the definition of $R$.
- Lemma A.2, $F^\star$ has not been introduced before in Section A.
- Line 632-635: It is not clear for me why the constraint $L_2\log{L_2} \ge h \log{H}$ implies $L_2 \ge H \exp{\frac{h}{L_2}}$. Could you detail this point ?
- Could you provide a reference for the proof of Lemma B.1 ?
- In the proof of Lemma B.3, by applying the Young inequality, you assume that $p$ and $q$ are strictly positive which is not the case in your definition of $(L_0, L_1, L_2)$-smoothness. Could you precise this case when $p = 0$ or $q = 0$ ?
- Line 802: it is not evident for me that $\beta \sqrt{g(\mathcal{F}_1)}$ is smaller that $\min\{1/L_1, 1/L_2\}$, could you justify this statement ?
- Line 846: I suggest to recalled the definition of $\lambda_l$ for clarity
- Line 248: The point 2. of Lemma 4.1 could be precise into $B_t < A_t$ since the step-size is strictly positive. It will simplify the computation in line 848.
- Line 852: I suggest to recall the definition of $\beta$ for clarity.
- Lemma D.2: The definition of $\xi_k$ is not provided.
- Line 1102: I will detail that you use the definition of $\beta$ and in particular the inequality with $\mathcal{G}_{1,1}$.
- Line 1161: It should have a $\frac{1}{2}$ factor in front of the term in $\|\xi_l\|^2$
- Line 1166: I suggest to recall the setting of $\lambda_l$ for clarity as it is changing through theorems.
- Line 1178: The definition of $\beta$ related to $\mathcal{G}_{1,3}$ is used implicitly, could you make it explicit ?
- Line 1093-1095: The introduction paragraph is a bit confusing. I suggest to reformulate with an explicit use of Lemma D.1
- Equation (66), what is the definition of $t$ ? I think it should be $T$ instead.
- Line 1214: It is not very precise. Equation (59) is used but not Lemma D.5. You should state clearly that equation (51) implies (59), which is used there.
- Lemma D.7 in line 1292: You use Equation (66) (of Lemma D.6) to demonstrate this lemma. However, Equation (66) is derived under the hypothesis that $f(x_{k}^{md}) - f^\star \le \mathcal{F}_2$ which is the inequality that you aim to demonstrate in Lemma D.7. I suspect a circular argument. Can you make explicit in each Lemma (D.5, D.6, D.7) which are the assumptions and re-write the paragraph in line 1292 to clarify the arguments ?

[1]  Hossein Taheri and Christos Thrampoulidis. Fast convergence in learning two-layer neural networks with separable data. In Proceedings of the AAAI Conference on Artificial Intelligence, volume 37, pp.9944–9952, 2023.

**Questions:**

- Could you comment how this paper is relaxed with the following works ?
Y. Carmon, J.C. Duchi, O. Hinder, and A. Sidford, “Lower bounds for finding stationary points,” Mathematical Programming, vol. 184, no. 1-2, pp. 71–120, 2020.
J. Hermant, M. Renaud, J.F. Aujol, C. Dossal, A. Rondepierre, "Gradient correlation is a key factor to accelerate SGD with momentum", ICLR 2025
Y. Hong, J. Lin "On Convergence of Adam for Stochastic Optimization under Relaxed Assumptions", Neurips 2024
- In section A, an example of function (L_0, 0, L_2)-smooth is given. What is the practical interest of the relaxed $L_1$, do you have concrete example of function which are (L_0, L_1, L_2)-smooth and not (0, L_1, L_2)-smooth or (L_0, 0, L_2)-smooth or (L_0, L_1, 0)-smooth ?
- In which real-world application, the relaxed smoothness assumption appears ? What is the motivation for looking at this assumption ?

---

> ### Author Response · Authors · 2025-11-25
> **Response to Reviewer Rmjo (1/5)**
>
> # Response to Reviewer Rmjo (1/5)
> We thank you for your valuable comments which have greatly improved our presentations. We revised accordingly in the revision version.
>
> > The assumption of this work are far from real-world machine learning applications. First, the function needs to be convex which is not verified for neural network training. Then, the relaxed assumption introduced in this work is verified by a very particular neural network detailed in Appendix A. However, it is not clear why this ''relaxed assumption'' is interesting for real-world applications. This ''relaxed assumption'' of smoothness is only motivated by the fact that it is ''relaxed''.
>
> Thank you for your question.  We made some corresponding changes in the revision and we would like to clarify as follows.
>
> We __drop the convexity assumption__ by adding the convergence analysis with respect to the stationary points under the new introduced generalized $(L_0,L_1,L_2)$-smoothness assumption for the non-convex stochastic optimizations __in Appendix G__.
>
>  Our new introduced $(L_0,L_1,L_2)$-smoothness is situated between two related notions: $(L_0,L_1,0)$-smoothness, which is empirically known [Zhang et al., 2020] for neural networks training (but without theoretically proven) and is theoretically proved for phase retrieval in the appendix, and $(L_0,0,L_2)$-smoothness, which is theoretically proven for specific shallow neural networks in the appendix. We propose the $(L_0,L_1,L_2)$-smoothness condition, which subsumes both and provides a unified framework for analysis.
>
>  We also added two numerical experiments on neural networks training and phase retrieval on synthesis datasets __in Appendix I__, showing that stochastic NAG converges faster in these two problem compared to SGD.
>   We do not know why this happens, and we thus start with
>  the simple convex case and our theoretical results indeed show faster convergence rate of  stochastic NAG over SGD, under the $(L_0,L_1,L_2)$-smoothness assumption.
>
>  Motivated by our numerical experiments, we think that neural networks and phase retrieval problems may have some nice properties such as quasi-convexity, but examining these properties are too difficult for us and would be left as a future work.
>  In fact, there are other kinds of work showing that for phase retrieval problems with structured random sampling, the object function has been shown to be (quasi)-convex provided the sample size is large enough.
>
>  As highlighted in both the introduction and the main theorem section, proposing the new $(L_0,L_1,L_2)$-smoothness condition is only one of our contributions. Our second contribution is establishing the fast convergence rate for stochastic NAG under $(L_0,L_1,0)$-smoothness. This has been an open problem since [Li et al., 2024], who provided a fast rate only for deterministic NAG and a standard rate for stochastic gradient descent. We believe a proof for stochastic NAG under this condition presents challenges; in particular, the analysis for deterministic NAG by [Li et al., 2024] does not appear to be trivially extendable.
>
> Our third contribution is to provide fast convergence rates for stochastic NAG in standard smooth stochastic optimization under the more general expected relaxed affine-variance noise [Khaled and Richt´arik, 2023], which has been shown to encompass many practical noise settings, and such a result is new for stochastic NAG.
>
> To highlight our theoretical contributions, we added two tables __in Appendix A__.

---

> ### Author Response · Authors · 2025-11-25
> **# Response to Reviewer Rmjo (2/5)**
>
> # Response to Reviewer Rmjo (2/5)
>
> > No experiments or real-world applications of the convergence results are provided reducing strongly the impact of the work.
>
> Thank you for your comments. We ran the stochastic NAG algorithm on the phase retrieval problem and a two-layer neural network; the results are provided __in Appendix I__.
>
> > The paper is not pedagogical. In particular, I appreciate the effort to give insides on the proof of Theorem 2 in Section 2. However, Section 6 is very technical and unclear for me without looking at the detailed proof in Appendix. I suggest to remove it of the main paper.
>
> Thank you for your suggestion on the presentation. We have tried to revised Section 6 to improve its clarity and flow. Our intention was to provide a proof sketch in the main text and highlight the novelties of our proof. We have adopted a writing style that is standard for this conference. We believe it clearly presents our work, though we thank the reviewer for the suggestion.
>
> > In the introduction [e.g. in Contributions (c), (d) or in line 166], you mention  the constants involves in Assumption 4 but without introduced it before. I suggest to move Assumption 4 around equation (4) in order to state notations before using them.
>
> We agree that clarifying these parameters earlier would improve the presentation. For the purpose of the discussions and to maintain consistent reference to Assumptions 3 and 4 throughout the rebuttal period, we have kept the formal definitions in their current location. In the final version, we will move Assumptions 3 and 4 into the introduction section.
>
> > Some notations are confusing. In Algorithm 1, you use $x_t$, $x_t^{md}$ and $x_t^{ag}$ without introducing what denote ''md'' and ''ag''. I will suggest to use different letters for each of these intermediary sequence, e.g. $x,y,z$. Moreover, I suggest to state explicitly that $x_t^{ag}$ is the output of Algorithm 1.
>
> We have added the introduction to them. In each theorem, we emphasize that $x_t^{ag}$ is the output of the algorithm.

---

> ### Author Response · Authors · 2025-11-25
> **Response to Reviewer Rmjo (3/5)**
>
> # Response to Reviewer Rmjo (3/5)
>
> > Definition 1: because $\|\|x-y\|\| \leq \min \{1/L_1,1/L_2\}$, you need that $L_1,L_2>0$ not only $L_1, L_2\geq 0$. Or could you  precise that you mean by $1/0=+\infty$.
>
> We added a footnote ''For the sake of rigor, we define $1/0=+\infty$ throughout the paper." on Page 2.
>
> > Line 351: What is $\mathcal{F}_2$? Could you add the definition in Section 6?
>
> $\mathcal{F}_2$ is a positive constant from the appendix. We added ''$\mathcal{F}_2$ is a positive constant given explicitly in (51)." in the revision.
>
> > Equation(19): It seems that $\tilde{d}=d \times m$, could you state it for clarity? The weights $a_j$ seem to not be trainable (it is the setting of [1]), could you state it clearly?
>
> You are absolutely correct that $\tilde{d}=d\times m$ and $a_j$ is fixed. We have added the description about this issue in the revision.
>
> > Line 593. In your notation, according to [1], it should be $w_i$ instead of $x_i$ in the definition of $R$.
>
> You are correct and we revised it accordingly.
>
> > Lemma A.2, $F^*$ has not been introduced before in Section A.
>
> We have added the introduction to $F^*$.
>
> > Line 632-635: It is not clear for me why the constraint $L_2\log L_2\geq h \log H$ implies $L_2\geq H \exp(h/L_2)$. Could you detail this point ?
>
> We changed ''$L_2 \log L_2\geq h \log H$" as ''$L_2=\max \{h, H\text{e}\}$".
>
> > Could you provide a reference for the proof of Lemma B.1?
>
> We added ''Refer to [Attia and Koren, 2023] for a proof".
>
> > In the proof of Lemma B.3, by applying the Young inequality, you assume that $p$ and $q$ are strictly positive which is not the case in your definition of $(L_0,L_1,L_2)$-smoothness. Could you precise this case when $p=0$ or $q=0$?
>
> Thanks for your careful reading. We apply Young's inequality to the term $2L_1\|\|\nabla f(x_t)\|\|^p \triangle_t$, which is independent of $q$. In the revised version, we first assume $0<p<2$ and then apply Young's inequality. Since we denote $0^0=1$, (29) still holds when $p=0$.
>
>
> > Line 802: it is not evident for me that $\beta\sqrt{g(\mathcal{F}_1)}$ is smaller than $\min\{1/L_1,1/L_2\}$, could you justify this statement?
>
> Thanks a lot. We have $\beta\sqrt{g(\mathcal{F}_1)}\leq \min\{1/L_1,1/L_2\}$ since $\beta=\frac{1}{\mathcal{L}_1}$, where $$\mathcal{L}_1=2\left(L_0+L_1\left(\left(g(\mathcal{F}_1)\right)^{\frac{1}{2}}+\left(g(\mathcal{F}_1)\right)^\frac{p}{2}\right)+L_2\left(\left(g(\mathcal{F}_1)\right)^{\frac{1}{2}}+\mathcal{F}_1^q\right)+8\mathcal{C}_1^2\left(L_1+L_2\right)^4 \right).$$
> To make it easier to follow, we have modified the presentation correspondingly.
>
> > Line 846: I suggest to recalled the definition of $\lambda_l$ for clarity.
>
> Thank you for your valuable suggestion. We have modified such issue throughout our paper.

---

> ### Author Response · Authors · 2025-11-25
> **Response to Reviewer Rmjo (4/5)**
>
> # Response to Reviewer Rmjo (4/5)
>
> > Line 248: The point 2. of Lemma 4.1 could be precise into $B_t<A_t$ since the step-size is strictly positive. It will simplify the computation in line 848.
>
> Thanks a lot. We have revised it accordingly.
>
> > Line 852: I suggest to recall the definition of $\beta$ for clarity.
>
> Thanks for your good advice. We have addressed this kind of issue.
>
> > Lemma D.2: The definition of $\xi_k$ is not provided.
>
> We have added the definition of $\xi_k$ at the beginning of Section 6.
>
> > Line 1102: I will detail that you use the definition of $\beta$ and in particular the inequality with $\mathcal{G}_{1,1}$.
>
> Yes. We have emphasized it in the revised version.
>
> > Line 1161: It should have a $\frac{1}{2}$ factor in front of the term in $\|\|\xi_l\|\|$.
>
> We believe the constant should be $1$ instead of $\frac{1}{2}$, which is derived from (63), (64), (65) in the revised paper. The factor might be $1/2$ with a more dedicated computation. We will look into it.
>
> > Line 1166: I suggest to recall the setting of $\lambda_l$ for clarity as it is changing through theorems.
>
> We revised it accordingly.
>
> > Line 1178: The definition of $\beta$ related to $\mathcal{G}_{1,3}$ is used implicitly, could you make it explicit?
>
> We revised accordingly.
>
> > Line 1093-1095: The introduction paragraph is a bit confusing. I suggest to reformulate with an explicit use of Lemma D.1.
>
> We revised accordingly.
> In the revision, we assume (53) alway holds for all $t\in[T]$ in the subsequent proof after.
>
> > Equation (66), what is the definition of $t$? I think it should be $T$ instead.
>
> The iteration $t$ here is defined in Lemma E.5, i.e, ''Suppose that $f(x_l^{md})-f^*\leq\mathcal{F}_2, \forall l\in[t]$.'' Lemma E.6 is based on Lemma E.5 and Equation (66) (Equation (67) in the revised version) holds for $0\leq l\leq t$ only, instead of $0\leq l \leq T$. We modified the conditions and results of Lemma E.5 in the revision.
>
> > Line 1214: It is not very precise. Equation (59) is used but not Lemma D.5. You should state clearly that equation (51) implies (59), which is used there.
>
> We revised accordingly and also Lemma E.5. See Line 1326 in the revision.
>
> > Lemma D.7 in line 1292: You use Equation (66) (of Lemma D.6) to demonstrate this lemma. However, Equation (66) is derived under the hypothesis that $f(x_k^{md})-f^*\leq \mathcal{F}_2$ which is the inequality that you aim to demonstrate in Lemma D.7. I suspect a circular argument. Can you make explicit in each Lemma (D.5, D.6, D.7) which are the assumptions and re-write the paragraph in line 1292 to clarify the arguments?
>
> In the revision, Section D becomes Section E and Lemmas D.5, D.6, D.7 are replaced by Lemmas E.5, E.6, E.7 respectively. And we now made explicit in each lemma the assumptions and we modified the proof slightly in this section. (74) in Lemma E.7 (D.7 in original version)  holds if Equations (53), (54), and (58) hold. Basically we assume inequities (53), (54), and (58) hold, and we prove the desired error bounds using an induction argument. Note that we can prove that  (53), (54), and (58) hold with high probability using the noise assumptions.
>
> In the proof for Lemma E.7, we apply induction an argument. We first assume that for some $t\in[T]$, $f(x_l^{md})-f^* \leq \mathcal{F}_2$ holds for all $l\in[t]$.
>
> Then, we prove $f(x_{t+1}^{md})-f^* \leq \mathcal{F}_2$ using Lemma E.5 and Lemma E.6, which are based on the hypothesis that $f(x_l^{md})-f^*\leq \mathcal{F}_2, \forall l\in[t]$.
>
> We also revised the proof sketch in Section 6
>
> Thank you so much for reading the proof and your questions which have made the proof logic more clearer and improved the presentation.
> We will proofread the manuscript throughly and carefully in the final version.

---

> ### Author Response · Authors · 2025-11-25
> **Response to Reviewer Rmjo (5/5)**
>
> # Response to Reviewer Rmjo (5/5)
>
> > Could you comment how this paper is relaxed with the following works ? Y. Carmon, J.C. Duchi, O. Hinder, and A. Sidford, ''Lower bounds for finding stationary points'', Mathematical Programming, vol. 184, no. 1-2, pp. 71–120, 2020. J. Hermant, M. Renaud, J.F. Aujol, C. Dossal, A. Rondepierre, ''Gradient correlation is a key factor to accelerate SGD with momentum'', ICLR 2025 Y. Hong, J. Lin ''On Convergence of Adam for Stochastic Optimization under Relaxed Assumptions'', Neurips 2024
>
> Both [Arjevani et al., 2023] and [Hermant et al., 2025] focused on smooth ($L_1,L_2=0$ in Definition 1) optimization while the former one provided the lower bound for first-order algorithms with bounded variance ($A,B=0$ in Assumption 4) and the latter studied the variant of stochastic NAG with strong growth noise assumption ($A,C=0$ in Assumption 4).
> Hong and Lin [2024] analyzed Adam under the affine-variance noise assumption of its sub-Gaussian form and $(L_0,L_1)$-smoothness of its modified version, i.e., there exist $L_0,L_1\geq 0$, for any $x,y\in\mathbb{R}^d$, such that $\|\|x-y\|\|\leq 1/L_1$,
> $$
>   \|\|\nabla f(x)-\nabla f(y)\|\|\leq \left(L_0+L_1\|\|\nabla f(x)\|\|^p\right)\|\|x-y\|\|,
> $$
> where $p$ is a non-negative constant. However, $(L_0,L_1,L_2)$-smooth or $(L_0,0,L_2)$-smooth functions are not included.
>
> We also added some related comments in the revision.
>
> > In section A, an example of function $(L_0, 0, L_2)$-smooth is given. What is the practical interest of the relaxed $L_1$, do you have concrete example of function which are $(L_0, L_1, L_2)$-smooth and not $(0, L_1, L_2)$-smooth or $(L_0, 0, L_2)$-smooth or $(L_0, L_1, 0)$-smooth?
>
> We do not have any intuition for which kind of functions are $(L_0, L_1, L_2)$-smooth but not $(0, L_1, L_2)$-smooth or $(L_0, 0, L_2)$-smooth or $(L_0, L_1, 0)$-smooth, and we agree with that investigating more on this aspect is interesting.
> However, we believe the novel $(L_0, 0, L_2)$-smoothness condition is meaningful since it subsumes both $(L_0,L_1,0)$-smoothness and $(L_0,0,L_2)$-smoothness.
>
> > In which real-world application, the relaxed smoothness assumption appears? What is the motivation for looking at this assumption?
>
> Thanks for your question.  Our new introduced $(L_0,L_1,L_2)$-smoothness is situated between two related notions: $(L_0,L_1,0)$-smoothness, which is empirically known [Zhang et al., 2020] for neural networks training (but without theoretically proven) and is theoretically proved for phase retrieval in the appendix, and $(L_0,0,L_2)$-smoothness, which is theoretically proven for specific shallow neural networks in the appendix. We propose the $(L_0,L_1,L_2)$-smoothness condition, which subsumes both and provides a unified framework for analysis.
> We do not have any intuition for which real-world application verifies this condition, and we agree with that investigating more on this aspect is interesting.

---

> > ### Comment · Reviewer_Rmjo · 2025-11-25
> >
> > I thanks the authors for their detailed answer and taking into account my comments.
> >
> > In particular, I confirm that the technical issues that I pointed have been corrected. Therefore, I am confident with the fact that the results presented in this paper are correct.
> >
> > About my major weakness:
> > - I truly appreciate for adding the example of phase retrieval. It gives a real-world application to this work. Thanks ! Understanding more the introduced relaxed smoothness assumption is an important line of research to understand the scope of this work.
> > - I also appreciate the preliminary numerical experiments. It gives more examples of applications of your work. However, I believe that the experimental part could be detailed.
> > - I see that effort have been made to make the paper more pedagogical. I still believe that Section 6 should be remove (for instance putting the experimental part instead). Moreover, I read that the comparison with the literature is more complete and the tables of Appendix A make the paper more clear.
> >
> > Thus, I consider that the authors take into account my main weaknesses. So I raise my score accordingly. I think this paper will be valuable for the conference.

---

> ### Author Response · Authors · 2025-11-25
> **We truly appreciate your helpful and encouraging comments**
>
> We are delighted that we have addressed your major concerns! We truly appreciate your helpful comments! We learn a lot from your detailed comments and feel very delightful during the discussion. Your encouraging comments will inspire us to make further progress. In fact, we are working on improving our bounds from $1/T^2 + \sqrt{(A+B+C)/T}$ to $1/T^2 + \sqrt{(A+C)/T}$, motivated by the interesting references you shared (also from Reviewer bvGq ). Hopefully we can incorporate these result in the final version.
>
> > I also appreciate the preliminary numerical experiments. It gives more examples of applications of your work. However, I believe that the experimental part could be detailed.
>
> We will try to add more details and more experiments in the final version.
>
> > I see that effort have been made to make the paper more pedagogical. I still believe that Section 6 should be remove (for instance putting the experimental part instead). Moreover, I read that the comparison with the literature is more complete and the tables of Appendix A make the paper more clear.
>
> We agree with the suggestion to replace the proof sketch with the experimental section (and comparison tables). We acknowledge that the current proof sketch is not so helpful, a concern also noted by Reviewer rtaj and Reviewer t1mr.
> We will incorporate your suggestions into the final version. Thanks!

---

> ### Author Response · Authors · 2025-12-03
> **Summary: Major Weakness from Reviewer Rmjo and Response, Discussions**
>
> ## Major Weakness from Reviewer Rmjo and Our Response
> **W1. The function needs to be convex. The relaxed assumption introduced is verified by a particular neural network**\
> **R1.**
> - We **drop the convexity** by adding convergence in the non-convex case (Appendix G).
> - This relaxed (smooth) assumption unifies $(L_0,L_1,0)$-smoothness [Zhang et al., 2020] for neural networks (empirically) and phase retrieval [Chen et al., 2023] (theoretically), and $(L_0,0,L_2)$-smoothness [Taheri & Thrampoulidis, 2023] for neural networks (theoretically).
> - Besides, our contributions include (see Appendix A for comparisions):
>    - fast rate for stochastic NAG under $(L_0,L_1,0)$-smoothness, open since [Li et al., 2024] who provided a fast rate only for deterministic NAG and a standard rate for SGD,
>    - fast rates for stochastic NAG in standard smoothness under the more general expected relaxed affine-variance noise [Khaled and Richt´arik, 2023].
>
> **W2. No experiments are provided reducing the impact.** \
> **R2.** We added two simple experiments (Section 6) to complement.
>
> **W3.Typos/imprecisons; re-write the paragraph in line 1292 to clarify**\
> **R3.** We revised accordingly. We modified slightly the proof accordingly.
>
>
> ## Final Comments (26 Nov 2025, 00:24) from Reviewer Rmjo
>
>  **The technical issues have been corrected. The authors take into account my main weaknesses. I raise my score. This paper will be valuable.**

---

### Author Response · Authors · 2025-11-25
**General Response and Major Changes**

# General Response and Major Changes
We thank all the reviewers for their valuable comments which have greatly improved our presentations.
We revised accordingly in the revision version.
Those major changes were marked with blue.  Particularly,  we
* added the convergence analysis with respect to the stationary points under the new introduced generalized \$(L_0,L_1,L_2)$\-smoothness assumption
for the non-convex stochastic optimizations in Appendix G;
* added two numerical experiments on neural networks training and phase retrieval on synthesis datasets in Appendix I, showing that stochastic NAG converges faster in these two problem compared to SGD;
* added two tables in Appendix A, to highlight our theoretical contributions.




**Reference**:

-Yossi Arjevani, Yair Carmon, John Duchi, Dylan J Foster, Nathan Srebro, and Blake
Woodworth. Lower bounds for non-convex stochastic optimization. Mathematical
Programming, 199(1):165–214, 2023.

-Amit Attia and Tomer Koren. SGD with AdaGrad stepsizes: Full adaptivity with high
probability to unknown parameters, unbounded gradients and affine variance. In International
Conference on Machine Learning, 2023.

-Sorin-Mihai Grad, Felipe Lara, and Raúl T Marcavillaca. Strongly quasiconvex functions:
what we know (so far). Journal of Optimization Theory and Applications, 205(2):38,
2025.

-Charles Guille-Escuret, Manuela Girotti, Baptiste Goujaud, and Ioannis Mitliagkas. A
study of condition numbers for first-order optimization. In International Conference
on Artificial Intelligence and Statistics, pages 1261–1269. PMLR, 2021.

-Julien Hermant, Marien Renaud, Jean-François Aujol, Charles Dossal, and Aude Rondepierre.
Gradient correlation is a key factor to accelerate SGD with momentum. In
International Conference on Learning Representations, 2025.

-Oliver Hinder, Aaron Sidford, and Nimit Sohoni. Near-optimal methods for minimizing
star-convex functions and beyond. In Conference on Learning Theory, pages 1894–1938.
PMLR, 2020.

-Yusu Hong and Junhong Lin. On convergence of Adam for stochastic optimization under
relaxed assumptions. Advances in Neural Information Processing Systems, 37:10827–
10877, 2024.

-Hamed Karimi, Julie Nutini, and Mark Schmidt. Linear convergence of gradient and
proximal-gradient methods under the polyak-łojasiewicz condition. In Joint European
Conference on Machine Learning and Knowledge Discovery in Databases, pages 795–
811. Springer, 2016.

-Ahmed Khaled and Peter Richtárik. Better theory for SGD in the nonconvex world.
Transactions on Machine Learning Research, 2023.

-Hao Chuan Li, Jian Qian, Yi Tian, Alexander Rakhlin, and Ali Jadbabaie. Convex
and non-convex optimization under generalized smoothness. In Advances in Neural
Information Processing Systems, 2024.

-Hossein Taheri and Christos Thrampoulidis. Fast convergence in learning two-layer neural
networks with separable data. In Proceedings of the AAAI Conference on Artificial
Intelligence, volume 37, pages 9944–9952, 2023.

-Bo Han Wang, Hui Shuai Zhang, Zhi Ming Ma, and Wei Chen. Convergence of AdaGrad
for non-convex objectives: Simple proofs and relaxed assumptions. In Conference on
Learning Theory, 2023.

-Jing Zhao Zhang, Tian Xing He, Suvrit Sra, and Ali Jadbabaie. Why gradient clipping
accelerates training: A theoretical justification for adaptivity. In International
Conference on Learning Representations, 2020.

---

> ### Author Response · Authors · 2025-12-03
> **New changes in the latest version (December 3)**
>
> * replace the proof sketch in Section 6 with numerical experiments (per suggested by Reviewer Rmjo on 25 Nov 2025)
> * modify the analysis of Theorem 3 to improve the sub optimal term from $\mathcal{O}(\sqrt{B/T})$ to $\mathcal{O}(B/T^2)$

---

### Author Response · Authors · 2025-12-03
**Final remarks**

We sincerely thank the reviewers' valuable and expert comments, which have greatly improved the paper. We also extend our thanks to the ACs for handling the extra workload, especially during this special year.

We **revised** according to valuable suggestions from reviewers. The major changes were marked with blue. Particularly, we
* added convergence analysis w.r.t. the stationary points under the new generalized $(L_0,L_1,L_2)$-smoothness assumption for the non-convex stochastic optimizations in Appendix G; (**Revision Part 1**)
* added two numerical experiments on neural networks training and phase retrieval on synthesis datasets in Section 6, showing that stochastic NAG converges faster in these two problem compared to SGD; (**Revision Part 2**)
* added two tables in Appendix A, to highlight our theoretical contributions. (**Revision Part 3**)
* clarified the motivation of proposing the novel smoothness condition and the contributions of our paper more clearly. (**Revision Part 4**)

We **highlight our main contributions** include:
- fast rate for stochastic NAG under $(L_0,L_1,0)$-smoothness and convexity, open since [Li et al., 2024] who provided a fast rate only for deterministic NAG and a standard rate for SGD,
- fast rates for stochastic NAG in standard convex smoothness under the more general expected relaxed affine-variance noise [Khaled and Richt´arik, 2023],
- introducing the new relaxed (smooth) assumption unifies $(L_0,L_1,0)$-smoothness [Zhang et al., 2020] for neural networks (empirically) and phase retrieval [Chen et al., 2023] (theoretically), and $(L_0,0,L_2)$-smoothness [Taheri & Thrampoulidis, 2023] for neural networks (theoretically), providing a unify analysis (in both convex and nonconvex cases)

We believe **we have addressed the reviewers' concerns**:
* Reviewer Rmjo (Weakness1) and Reviewer t1mr (Weakness1) raised concerns about the convexity assumption. We added nonconvex convergence (Revision Part 1) and highlighted our other theoretical contributions (Revision Parts 3-4).
* Reviewer Rmjo (Weakness2) and Reviewer rtaj (Weakness 3) said missing experiments. We added two simple numerical results. (Revision Part 2).
* Reviewer Rmjo (Weakness3) suggested ``re-write the paragraph in line 1292 to clarify". We modified slightly the proof accordingly to fix this technical clarity.
* Reviewer Rmjo (Weakness3), Reviewer t1mr (Weakness2-3), Reviewer bvGq (Weakness1-6), and Reviewer rtaj (Weakness 1-2) raised concerns about presentation issues. We revised accordingly.

According to **the final comments from Reviewer** Rmjo (25 Nov), Reviewer t1mr (28 Nov), and Reviewer bvGp (26 Nov), all of them thought their concerns have been addressed and said they will raise their scores. (In fact, they raised their scores before reverting: Rmjo, from 2 to 6; t1mr, from 6 to 8; bvGp, from 4 to 6.)

---

### Meta-Review · Area_Chair_PDZr · 2026-01-07

**Summary:**

This paper presents the convergence analyses for NAG method with a general smoothness condition. The authors provided rates for convex settings, and later in the rebuttal, rates for nonconvex settings. While some of the reviewers commented that they increased their scores, some weaknesses of the paper remain.

**Reviewer Concerns:**

There are several major weaknesses in this paper:
- The novelties and significance of the contributions. One reviewer stated that the work could be perceived as an incremental improvement over existing analyses. The authors' answer pointed to the proof sketch and section 6.3, then deleted it in the revision. Also, the proposed assumption is a generalization of two previous assumptions. Hence the novelty is limited, and this point is still unresolved.
- The limitations of the convex setting, which the authors addressed by adding a nonconvex analysis. While the nonconvex analysis plays an important role in the contribution of this paper, it is only stated in the appendix, and the authors introduced this part in the rebuttal. Given the short rebuttal timeframe and that the content is in the appendix, this addition is a major revision and should require another round of reviewing.
- The analysis is limited as it does not recover the optimal rate of NAG in the strong growth setting.

**Reviewer Scores:**

While some reviewers stated that they may have raised the score, their comments were very close to the incident; I am not able to see how their score changed. In addition, their comments are about the nonconvex analysis, which is part of the appendix, and requires a new thorough review. Given the short timeframe of the discussion that overlaps with the incident, which did not allow a full evaluation of the new nonconvex analysis, I do not take into account the comments of the reviewers that said they changed their reviews because of the nonconvex analysis.

---

### Decision · Program_Chairs · 2026-01-26

Reject